

# Bias correcting precipitation forecasts to improve the skill of seasonal streamflow forecasts

Louise Crochemore[1], Maria-Helena Ramos[1], and Florian Pappenberger[2,3]

[1]Irstea, Hydrosystems and Bioprocesses Research Unit, 1 rue Pierre Gilles de Gennes, F- 92 761, Antony, France.
[2]ECMWF, European Centre for Medium-Range Weather Forecasts, Shinfield Park, Reading, RG2 9AX, UK.
[3]School of Geographical Sciences, University of Bristol, University Road, Bristol, BS8 1SS, UK.

*Correspondence to:* Louise Crochemore (louise.crochemore@irstea.fr)

**Abstract.** Meteorological centres make sustained efforts to provide seasonal forecasts that are increasingly skilful, which has the potential to benefit streamflow forecasting. Seasonal streamflow forecasts can help to take anticipatory measures for a range of applications, such as water supply or hydropower reservoir operation and drought risk management. This study assesses the skill of seasonal precipitation and streamflow forecasts in France to provide insights into the way bias correcting precipitation

forecasts can improve the skill of streamflow forecasts at extended lead times. We apply eight variants of bias correction approaches to the precipitation forecasts prior to generating the streamflow forecasts. The approaches are based on the linear scaling and the distribution mapping methods. A daily hydrological model is applied at the catchment scale to transform precipitation into streamflow. We then evaluate the skill of raw (without bias correction) and bias corrected precipitation and streamflow ensemble forecasts in sixteen catchments in France. The skill of the ensemble forecasts is assessed in reliability,

sharpness, accuracy, and overall performance. A reference prediction system, based on historical observed precipitation and catchment initial conditions at the time of forecast (i.e., ESP method), is used as benchmark in the computation of the skill. The results show that, in most catchments, raw seasonal precipitation and streamflow forecasts are often more skilful than the conventional ESP method in terms of sharpness. However, they are not significantly better in terms of reliability. Forecast skill is generally improved when applying bias correction. Two bias correction methods show the best performance for the studied

catchments, each method being more successful in improving specific attributes of the forecasts: the simple linear scaling of monthly values contribute mainly to increasing forecast sharpness and accuracy, while the empirical distribution mapping of daily values is successful in improving forecast reliability.

## 1 Introduction

Numerous activities with economic, environmental and political stakes benefit from knowing and anticipating future streamflow

conditions at different lead times. While flood forecasting requires forecasts up to several hours or days ahead, other areas such as water supply reservoir operations or drought risk management need forecasts for the months or season ahead. Regardless of the considered lead time, streamflow forecasting systems are frequently updated to take the latest useful information content into account (e.g. last observed discharges, soil moisture or snow cover) and developed to make use of numerical weather model outputs to extend the range of skilful predictions.





Seasonal forecasts have shown to perfectly fall within a context of proactive risk management, for example, for drought management (e.g. Wilhite et al., 2000; Dutra et al., 2014; Mwangi et al., 2014; Wetterhall et al., 2015). Extended-range forecasting systems can be valuable tools to help decision-makers in planning long-term strategies for water storage (Crochemore et al., 2016) and to support adaptation to climate change (Winsemius et al., 2014). Nevertheless, several users still remain doubtful

whether seasonal forecasts can be trustworthy or skilful enough to enhance decision-making in an operational context (Rayner et al., 2005). Lemos et al. (2002) list the performance of seasonal forecasts, the misuse of seasonal forecasts by end-users and the lack of consideration of end-users' needs in the development of products as major obstacles to the widespread of seasonal forecasting in North-East Brazil. It is therefore crucial to assess the potential of available seasonal forecasting products and communicate on the assets and shortcomings of the different approaches that can benefit the water sector (Hartmann et al.,

10  2002).

Seasonal forecasting methods in hydrology can be broadly divided into two categories: statistical methods which use a statistical relationship between a predictor and a predictand, and dynamical methods which use seasonal meteorological forecasts as input to a hydrological model. More recently, mixed approaches have been investigated in the attempt to take advantage of initial land surface conditions, seasonal predictions of atmospheric variables and the predictability information contained

in large-scale climate features (see Robertson et al., 2013; Yuan et al., 2015, and references therein). Ensemble Streamflow Prediction (ESP; Day, 1985) is a dynamical method that is widely used to forecast low flows and reservoir inflows at long lead times (Faber and Stedinger, 2001; Nicolle et al., 2014; Demirel et al., 2015). It consists in using historical weather data as input to a hydrological model whose states were initialized for the time of the forecast. The ESP method is also used along with the Reverse-ESP method to determine the relative impacts of meteorological forcings and hydrological initial conditions on

the skill of streamflow predictions (Wood and Lettenmaier, 2008; Shukla et al., 2013; Yossef et al., 2013). An alternative dynamical method consists in using seasonal forecasts from regional climate models (RCMs) (Wood et al., 2005). This approach yields better results when seasonal predictability is enhanced by meteorological forcings rather than by initial conditions. Climate model outputs may also be more suitable to capture the specific climate conditions at the time of the forecast, whereas ESP-based methods will be limited to the range of past observations and challenged by climate non-stationarity.

The use of climate model outputs in hydrology has however some methodological implications. Outputs are produced for grid scales that are usually too coarse for streamflow forecasting at the catchment scale. This can lead to errors in capturing forecast uncertainty and introduce significant biases. Post-processing (including bias correction techniques and downscaling procedures) is usually a necessary first step prior to using climate model outputs to model streamflow. A range of methods has been proposed in the literature and the best method usually depends on the modelling chain being investigated and the studied

area, with levels of performance that may vary with the forecast horizon or the targeted application (Christensen et al., 2008; Gudmundsson et al., 2012).

Bias correction is usually an integral part of post-processing techniques applied to forecasting systems. Weather forecasting has performed bias correction of numerical model outputs through model output statistics (MOS) for decades. In hydrologic ensemble prediction systems, post-processing has become more and more popular in the last decade, particularly for medium-

range ensemble forecasting (e.g. Weerts et al., 2011; Zalachori et al., 2012; Verkade et al., 2013; Madadgar et al., 2014; Roulin





and Vannitsem, 2015). In seasonal forecasting, two popular bias correction methods are linear scaling and distribution mapping. Linear scaling corrects the mean of the forecasts based on the difference between observed and forecast means, whereas distribution mapping matches the statistical distribution of forecasts to the distribution of observations. These approaches, which can also be applied to improve the performance of ESP forecasts (Wood and Schaake, 2008), focus on increasing
forecast skill and reliability, by reducing errors in the forecast mean and improving forecast spread.

Studies comparing different bias correction methods in seasonal hydrological forecasting are still rare in the literature. However, we can find studies reviewing and comparing methods to bias correct RCM outputs and quantify climate change impacts, although their efficiency in this context is still a topic of discussion (Ehret et al., 2012; Muerth et al., 2013; Teutschbein and Seibert, 2013). Teutschbein and Seibert (2012) compared six methods, among which linear scaling and parametric distribution
mapping, to bias correct RCM simulations of precipitation and temperature in Sweden. The authors recommended using the distribution mapping method for current climate conditions. They also highlighted the need to assume that bias correction procedures are stationary to correct future climate projections and evaluate changes in flow regimes. In Norway, Gudmundsson et al. (2012) proposed a comparison of eleven methods to bias correct RCM precipitation. The methods derived from distribution transformations (e.g. distribution mapping based on fitted theoretical distributions), parametric transformations such as
linear scaling, and nonparametric transformations such as distribution mapping based on empirical distributions. Their study highlighted the differences between the bias corrections and the necessity to test methods prior to their application. The authors recommended using nonparametric methods since these methods were the most effective to reduce the bias and did not require any approximations of the empirical distributions.

The European Centre for Medium-range Weather Forecasts (ECMWF) produces seasonal forecasts from GCM simula-
tions (Molteni et al., 2011). Weisheimer and Palmer (2014) evaluated the reliability of the precipitation forecasts issued by ECMWF System 4 on a scale ranging from "dangerous" to "perfect". Over the world, precipitation forecasts often fell within the "marginally useful" category. In France, they were ranked as "marginally useful" during wet winters and summers, "not useful" in dry winters, and "dangerous" in dry summers. Kim et al. (2012) also evaluated the skill of System 4 precipitation and temperature at the global scale. Despite good overall performances, they identified systematic biases, e.g. a warm bias in the
North Atlantic. Several studies have proposed to bias correct ECMWF System 4 precipitation forecasts in different contexts. Di Giuseppe et al. (2013) applied a spatially-based precipitation bias correction to improve malaria forecasts. Trambauer et al. (2015) applied a linear scaling method to forecast hydrological droughts in Southern Africa. In the same context, Wetterhall et al. (2015) applied a quantile mapping method to daily precipitation values, and showed that bias correction was able to improve the skill of the system to forecast dry spells.
This paper aims to further investigate the potential of bias corrected ECMWF System 4 forecasts to improve streamflow forecasts at extended lead times. By comparing several variants of linear scaling and distribution mapping methods, the study provides insights into the way bias correcting seasonal precipitation forecasts can contribute to the skill of seasonal streamflow predictions. Forecasts are evaluated over the 1981-2010 period in 16 catchments in France. Section 2 presents the catchment set, the forecast and observed data, as well as the hydrological model used. Section 3 presents the bias correction methods





investigated, as well as the calibration and evaluation frameworks adopted. Results are presented in Sections 4 to 6 for the quality of the raw (uncorrected) and the bias corrected forecasts. In Section 7, conclusions and limitations are discussed.

## 2 Data and hydrological model

### 2.1 Seasonal forecasts and observed data

This study is based on daily seasonal precipitation forecasts from ECMWF System 4. System 4 provides a 51-member forecast ensemble for the next seven months at a TL255 (about 0.7°) spatial resolution (Molteni et al., 2011). ECMWF retrospectively produced forecasts for the period running from 1981 to 2010. These are composed of 51 ensemble members for February, May, August and November, and 15 members for the other months. For the purpose of this study, the 1981-2010 forecasts were aggregated at the catchment scale (i.e., areal precipitations were computed for each catchment). Only the first 90 days of the

forecast horizon were considered.

    Observed precipitation data used for the calibration and evaluation of the bias correction methods come from the SAFRAN reanalysis of Météo-France (Quintana-Seguí et al., 2008; Vidal et al., 2010). Daily values are available at an 8x8 km grid resolution covering France. They were also aggregated at the catchment scale. Mean areal potential evapotranspiration was computed for each catchment based on observed temperatures from the SAFRAN reanalysis (Oudin et al., 2005). Daily streamflow data

at the outlet of each catchment come from the French national archive (*Banque Hydro*).

### 2.2 Studied catchments and hydrological model

The catchment set was selected from the database in Nicolle et al. (2014). It comprises 16 catchments spread over France with a dominant pluvial regime. Catchments show an average solid fraction of precipitation below 10% and are thus not influenced by snow. Their main characteristics are shown in Table 1, and their location in Fig. 1.

We applied the conceptual, reservoir-based GR6J hydrological model (Pushpalatha et al., 2011) at the daily time step. The model is composed of three reservoirs (one for the production function and two for the routing function), and one unit hydrograph to account for flow delays. The model inputs are daily precipitation and potential evapotranspiration at the catchment scale. The model output is the daily streamflow at the catchment outlet. Interannual potential evapotranspiration was used to focus solely on the influence of precipitation inputs on streamflow forecasts. The model was calibrated in each catchment with

the Kling-Gupta Efficiency (Gupta et al., 2009) applied to root-squared flows. When the model is applied to forecast streamflow, the last observed streamflow at the time of forecast is used to update the levels of the routing reservoirs before issuing the forecasts.



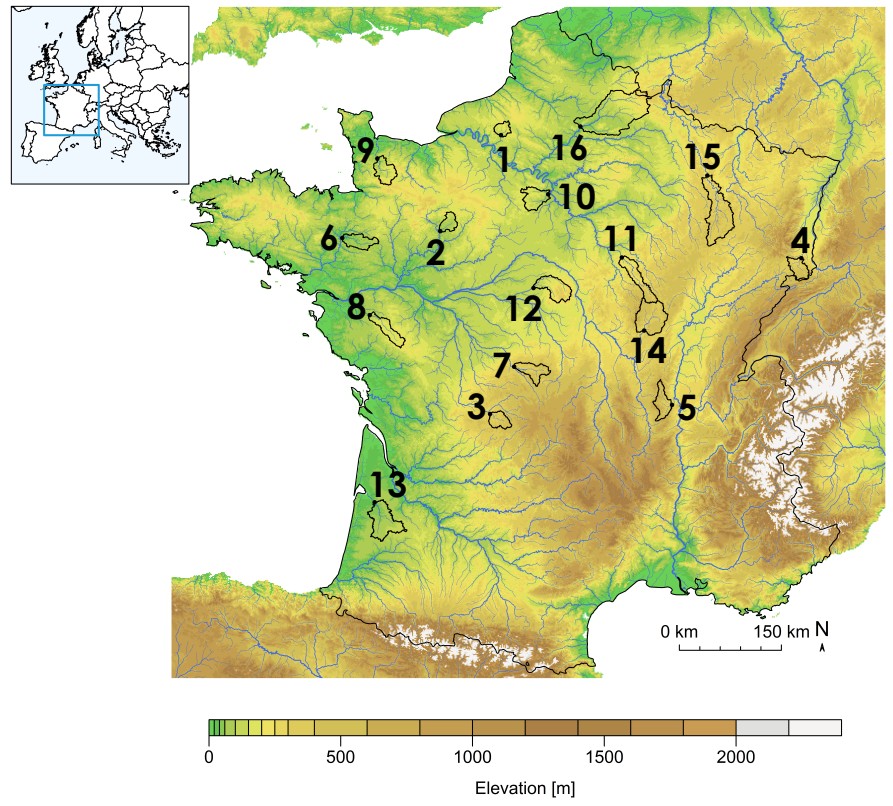

**Figure 1.** Location of the 16 studied catchments in France identified by their number (see Table 1 for details).

## 3 Methods

### 3.1 Overview of the calibration-evaluation approach

Bias correction methods were calibrated and evaluated in each catchment over the 1981-2010 period. The one-year-leave-out cross-validation method was applied to calibrate and evaluate the methods over independent periods. Given a target application year within the study period, all available years but the target year are used in the calibration process. Results of the calibration are then applied to the target application year and bias corrected forecasts are evaluated against observations.

In the calibration step, we considered two approaches. The simplest calibration uses all days of the years within the calibration dataset. An alternative approach consists in calibrating the bias correction methods for each calendar month. Additionally, since we are dealing with forecasts issued up to 90 days ahead, and since forecast performance varies with lead time, calibration also takes the lead time into account. In this study, lead times were grouped from 1 to 30 days, 31 to 60 days and 61 to 90 days ahead. The calibrated bias correction factors are then applied to the daily values of the ensemble precipitation forecasts in the target application year. The hydrological model is forced by precipitation forecasts and streamflow ensemble forecasts




are obtained. The modelling chain is applied to raw and bias corrected precipitation forecasts. Precipitation and streamflow forecasts are evaluated with deterministic and probabilistic scores commonly used in ensemble forecasting.

## 3.2    Bias correction methods

We applied the linear scaling (LS) and the distribution mapping (DM) methods to the raw System 4 precipitation forecasts. The DM method was applied following three variants: considering the empirical distribution of monthly values (EDM), a fitted gamma distribution of monthly values (GDM), and the empirical distribution of daily values (EDMD). Each method was applied on a monthly (-m) or a yearly (-y) basis (Table 2).

### 3.2.1    Linear scaling of precipitations

The LS method consists in correcting the monthly mean values of the forecasts to match the monthly mean values of the observations. A scaling factor (or bias) is calculated considering the ratio between the observed and the forecast (ensemble mean) values. A scaling factor higher (lower) than 1 indicates that the mean ensemble forecast underpredicts (overpredicts) the mean observed value. A value of 1 indicates no bias in the forecasts. The scaling factor obtained through calibration is then applied as a multiplicative factor to correct raw daily precipitation forecasts.

### 3.2.2    Distribution mapping of precipitations

The DM method consists in correcting the precipitation forecasts so that their statistical distribution matches that of the observations. There are several ways to match forecast and observed distributions or quantiles, and existing techniques mainly differ on how the forecast and observed cumulative distribution functions (CDF) are considered. In some techniques, a parametric distribution is fitted to the forecast and observed datasets, while in others the empirical distributions and linear interpolations between data points or estimated quantiles are considered. In any case, observed and forecast CDFs must be determined from long data series.

In this study, the calibration of the DM method was first carried out considering empirical (EDM) and gamma-fitted (GDM) distributions of observed and forecast (ensemble mean) precipitation values averaged monthly. A third variant considered directly the empirical distribution of the daily values of the ensemble members (EDMD). These variants are listed in Table 2. After calibration, bias correction is applied to the daily precipitation forecasts of each application period. In the case of EDM and GDM, all daily values are corrected based on the correction suited to their monthly average. In the case of EDMD, each daily precipitation value of each forecast member is corrected individually.

## 3.3    Evaluation framework

For each catchment, daily forecasts are issued once every month, up to 90 days ahead, during the 1981-2010 period. The quality of the forecasts was evaluated at the weekly time step (i.e., daily forecasts and observations are averaged over the week). Scores were computed as a function of lead time and for the winter (December-January-February), the spring (March-





April-May), the summer (June-July-August) and the autumn (September-October-November) seasons. Four criteria were used to assess reliability, sharpness, accuracy and overall performance of the ensemble forecasts (Gneiting et al., 2007; Eslamian, 2015; Musy et al., 2015).

### 3.3.1 Evaluation criteria

Reliability is a forecast attribute that refers to the statistical consistency between observed frequencies and forecast probabilities. In this study, it is evaluated with the Probability Integral Transform (PIT) diagram (Gneiting et al., 2007; Laio and Tamea, 2007). The PIT diagram is the cumulative distribution of the positions of the observation within the cumulative forecast distribution. A reliable forecast has a PIT diagram superposed with the 1:1 diagonal. If the PIT diagram shows a curve systematically above (below) the diagonal, the observed values are too frequently located in the lower (upper) parts of the forecast

distribution, suggesting a systematic bias of the forecasts towards overprediction (underprediction). If the points in the diagram are too concentrated in the vicinity of the end points (0 and 1), forecasts are too narrow and observations fall more frequently than expected on the tails of the forecast distribution. On the contrary, too many points concentrated in the midrange indicate a forecast distribution that is too wide. In order to numerically compare results among catchments, we also computed the area between the curve of the PIT diagram and the 1:1 diagonal, as proposed by Renard et al. (2010). The smaller this area is, the

more reliable the ensemble.

     Sharpness is a property of the forecasts only. It refers to the concentration of the predictive distribution and indicates how spread the members of an ensemble forecast are. In this study, sharpness was evaluated with the 90% interquantile range (IQR; Gneiting et al., 2007), i.e. the difference between the $95^{th}$ and the $5^{th}$ percentiles of the forecast distribution. The final IQR score is the average of the interquantile range at each time step of the evaluation period. The narrower the IQR is, the sharper

the ensemble.

     The accuracy of the forecasts is assessed with the mean absolute error (MAE). The MAE computes the average (over the evaluation period) of the absolute difference between the forecast ensemble mean and the observed value. Smaller MAE values correspond to more accurate forecasts.

     Last, the Continuous Rank Probability Score (CRPS) evaluates the overall performance of the forecasts. It is defined as

the integral of the squared distance between the cumulative distribution of the forecast members and a step function for the observation (Hersbach, 2000). The CRPS score is the average of this integral computed at each time step of the evaluation period. The lower the CRPS is, the better the overall performance of the forecasts.

### 3.3.2 Skill scores

Forecast skill is evaluated by comparing the performance of a given forecast system with the performance of a reference

forecast. The skill score is computed for a given lead time *i*.

$$SkillScore_i = 1 - \frac{Score_i^{Syst}}{Score_i^{Ref}} \qquad (1)$$



When the skill score is superior (inferior) to zero, the forecast system is more (less) skilful than the reference. When the skill score is equal to zero, the system and the reference have equivalent skill.

The skill scores were computed for the probabilistic scores presented in the previous section (noted PITSS, IQRSS and CRPSS hereafter). The reference used to evaluate precipitation forecasts is based on past observations and is representative of the catchment climatology: for a given day and year, it is the ensemble of precipitation values observed on that same Gregorian day in other years of the observation period. The reference used to evaluate streamflow forecasts is the Ensemble Streamflow Prediction (ESP), which corresponds to the streamflow ensemble obtained when the reference precipitation ensemble is used as input to the hydrological model. Pappenberger et al. (2015) highlight the importance of the reference chosen to compute skill scores and list a number of options for streamflow forecasting. The ESP is a commonly used method in seasonal forecasting. It allows applying the same hydrological modelling setup to both the precipitation forecasts and the reference precipitation ensemble. Therefore, differences in performance are mainly due to differences between the precipitation inputs to the model. One would expect that precipitation and streamflow forecasts perform better than precipitation climatology or ESP, at least in the first lead times. At longer lead times, natural variability should end up being a sound forecast. In our study, we also used an ensemble based on past streamflow observations (on the same day as the given forecast day) to evaluate performance. This allows to use as reference an ensemble that does not use any precipitation forecasts or hydrological model.

Finally, several studies have shown that the ensemble size induces a bias when computing skill scores with ensembles of different sizes. This bias usually leads to an underestimation of the skill of the forecast system when the system has fewer members than the reference. Ferro et al. (2008) provide a synthesis of previous studies on the influence of ensemble size on probability scores and propose a correction factor to remove the bias in the computation of CRPS skill scores. This correction was applied to compute the CRPSS in this study. Since the ensemble size of System 4 precipitation forecasts varies with the month, we used the ensemble size averaged over one year.

### 3.3.3 Gain in lead time from bias correcting seasonal forecasts

Skill scores can be computed to indicate the gain in performance brought by bias correction methods. To that effect, we use the raw (uncorrected) forecasts as reference in the computation of the skill scores. An indicator of forecast performance can be derived from the evolution of these skill scores: the lead time up to which bias corrected seasonal forecasts have more skill than raw forecasts. Nicolle et al. (2014) defined an indicator named UFL (Useful Forecasting Lead time) as "the lead time beyond which model performance is not at least 20% better than benchmark performance". Here, we considered the lead time beyond which the seven-day moving average of the skill score becomes negative. UFL values were then grouped in four categories: (1) None: no improvement over the forecast reference, (2) <30: gain up to 30 days, (3) <60: gain greater than 30 days and up to 60 days and (4) >60: gain greater than 60 days.



## 4 Quality of the raw seasonal forecasts

### 4.1 Performance of raw precipitation forecasts

Figure 2 presents the evolution of IQRSS and CRPSS with lead time, for winter (DJF) and summer (JJA). Each line corresponds to a catchment. Skill in sharpness and overall performance is very similar in winter and in summer (as well as in spring and

autumn, not shown). Precipitation forecasts are overall sharper than historical precipitations in the large majority of catchments and up to long lead times. Some exceptions appear for lead times longer than three weeks, and especially in winter (wetter season in the majority of catchments). In terms of overall performance, precipitation forecasts clearly have skill up to two to three weeks ahead for 7-day averaged areal precipitation. At longer lead times, they are equivalent or perform slightly worse than historical precipitations.

Figure 3 shows the PIT diagrams for lead times of 30 and 90 days, for winter and summer. Grey lines represent the reliability of historical precipitations and coloured lines represent the reliability of System 4 precipitation forecasts in each catchment. Dotted lines represent deviations of +0.1 and -0.1 from the bisector. The two seasons yield very similar results (also observed in spring and autumn, not shown). In all catchments and for both lead times, historical precipitations are reliable, as expected. Seasonal precipitation forecasts also show some reliability, but tend to overpredict precipitations in both seasons and at both

lead times. The concentration of points in the zero end points in most of the curves of the System 4 forecasts shows that low values of the observations are too often falling in the lower tail of the forecast distribution. This effect tends to decrease with increasing lead time. This is an indication that forecasts are too narrow and overpredict the lowest observations. It can also translate a difficulty of the system to forecast null precipitation.

### 4.2 Performance of raw streamflow forecasts

Streamflow forecasts are generated by using raw seasonal precipitation forecasts as input to the hydrological model. Forecast skill is evaluated using the ESP method as reference (Fig. 4). Differences in forecast skill between the winter and summer seasons are more noticeable when evaluating streamflow forecasts rather than precipitation forecasts. Streamflow forecasts generated from raw precipitation forecasts are sharper than ESP up to twelve weeks ahead in most catchments (IQRSS above zero in Fig. 4). Approximately, only four catchments stand out in both seasons with lower skill than ESP (six in spring and one

in autumn, not shown). However, even in these catchments, sharpness can be improved using seasonal precipitation forecasts for lead times up to three weeks in winter (as well as in spring and autumn, not shown). Concerning overall performance (CRPSS in Fig. 4), skill can be observed for lead times up to four weeks in some catchments. In winter, as well as in spring and autumn (not shown), this is observed in the majority of catchments, while in summer, this concerns only a couple of catchments. At longer lead times, ESP and streamflow forecasts generated from raw precipitation forecasts are equivalent in

most catchments for the winter season. In summer, as well as in spring and autumn (not shown), the difference in skill at longer lead times is more pronounced and most catchments have a clearly negative skill in terms of overall forecast performance.

PIT diagrams are shown for each catchment, for the winter and summer seasons, and for lead times of 30 and 90 days (Fig. 5). In winter and spring (not shown), ESP forecasts and seasonal streamflow forecasts generated from raw precipita-



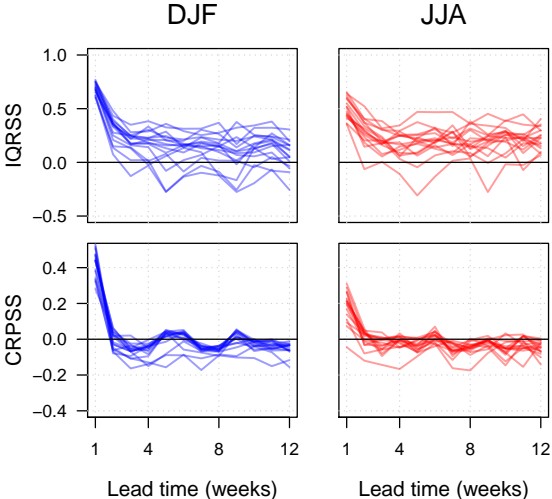

**Figure 2.** Skill of raw weekly precipitation forecasts as a function of the lead time for all catchments and all seasons. The skill is computed based on the IQR (top) and the CRPS (bottom) and the reference is historical precipitations. Each column corresponds to a target season. Each line represents the skill score in a catchment for forecast horizons within the target season.

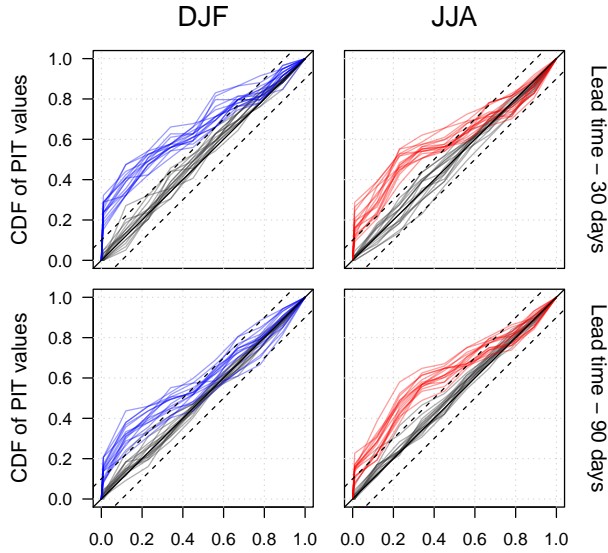

**Figure 3.** PIT diagram of raw precipitation forecasts (coloured lines) and historical precipitations (grey lines) for lead times of 30 days (top) and 90 days (bottom). Each column corresponds to a target season. Each line represents the PIT diagram in a catchment for forecast horizons within the target season.





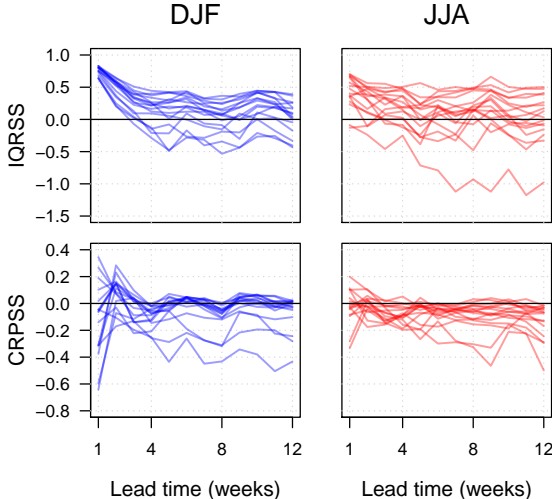

**Figure 4.** Skill of weekly streamflow forecasts from raw precipitation forecasts as a function of the lead time for all catchments and all seasons. The skill is computed based on the IQR (top) and the CRPS (bottom) and the reference is Ensemble Streamflow Prediction. Each column corresponds to a target season. Each line represents the skill score in a catchment for forecast horizons within the target season.

tion forecasts show good reliability, although the curves above the diagonal indicate that forecasts are slightly overpredicting streamflow. Streamflow forecasts for the autumn season (not shown) also show good reliability, but with a tendency to underpredict streamflow. In summer (Fig. 5, right), streamflow forecasts from both, ESP forecasts and forecasts generated from raw seasonal precipitation forecasts, show problems in forecast reliability. PIT curves clearly indicate a concentration of points at

5 the end points of the diagram and, consequently, narrow ensemble forecasts. In most catchments, 20% to 60% of observed values fall in the lowest interval of the forecast distribution or below it, i.e., outside the forecast range. Although reliability is slightly improved with lead time, streamflow ensemble forecasts remain under-dispersive at 90 days of lead time. This could be the result of at least two factors acting alone or jointly: a difficulty of the hydrological model to reach the lowest streamflow values in the simulations of the recession periods, and the influence of not considering uncertainties in the hydrological initial

conditions at the time of forecasting.

### 4.3 Summary of the quality of raw seasonal forecasts

Skill in the overall performance of System 4 raw precipitation forecasts, at the catchment scale and over a reference forecast based on past observed precipitations, was observed up to two to three weeks in the studied catchments. When looking at streamflow forecasts generated from the input of raw seasonal forecasts to a hydrological model, skill over the traditional ESP

method was observed up to four weeks, but only in few catchments. The asset of System 4 raw precipitation forecasts and related streamflow forecasts over historical precipitations and ESP, respectively, resides mainly in their sharpness. However, the evaluation of forecast quality shows also that forecasts are often too narrow and suffer from underprediction or overpre-





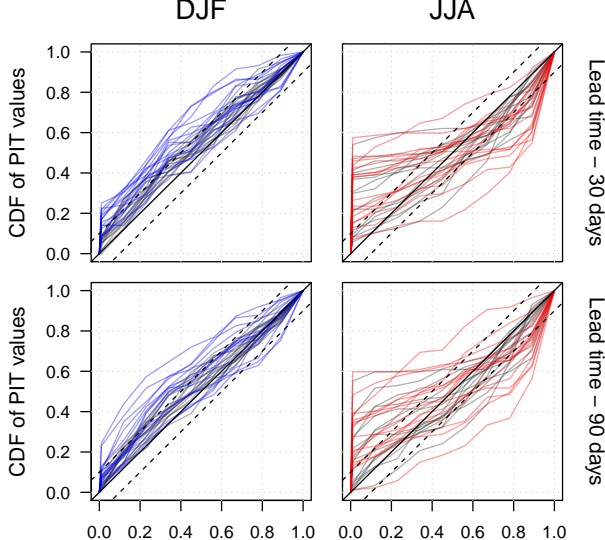

**Figure 5.** PIT diagram of streamflow forecasts from raw precipitation forecasts (coloured lines) and Ensemble Streamflow Prediction (grey lines) for lead times of 30 days (top) and 90 days (bottom). Each column corresponds to a target season. Each line represents the PIT diagram in a catchment for forecast horizons within the target season.

diction. Improving forecast reliability, while maintaining forecast sharpness is clearly a challenge. In the following section, we investigate the presence of biases in System 4 precipitation forecasts and the impact of bias correction on seasonal precipitation and streamflow forecasts.

# 5 Bias correction of seasonal precipitation forecasts

5  ## 5.1 Overview of the effectiveness of the bias correction methods

Forecast bias, i.e. the ratio between the mean observation and the average forecast ensemble mean, was computed for each catchment over the 1981-2010 period. The bias was computed for each calendar month, but also considering the whole year. Figure 6 shows the biases expressed as deviations from 1 (i.e., $1 - Bias$), before and after applying the bias correction methods. It illustrates the results obtained in four catchments at the 2-month lead time (i.e., considering the forecasts issued for day 31 

10  to day 60 in the forecast range). The effectiveness of each bias correction method can be easily seen from the coloured charts: unbiased forecasts have a deviation equal to 0 (white colour); positive deviations (red colour) and negative deviations (blue colour) indicate overprediction and underprediction, respectively. A deviation equal to 0.75 (-3) can be interpreted as the mean forecast being four times larger (smaller) than the mean observation. Overall, when computing the deviations for all monthly





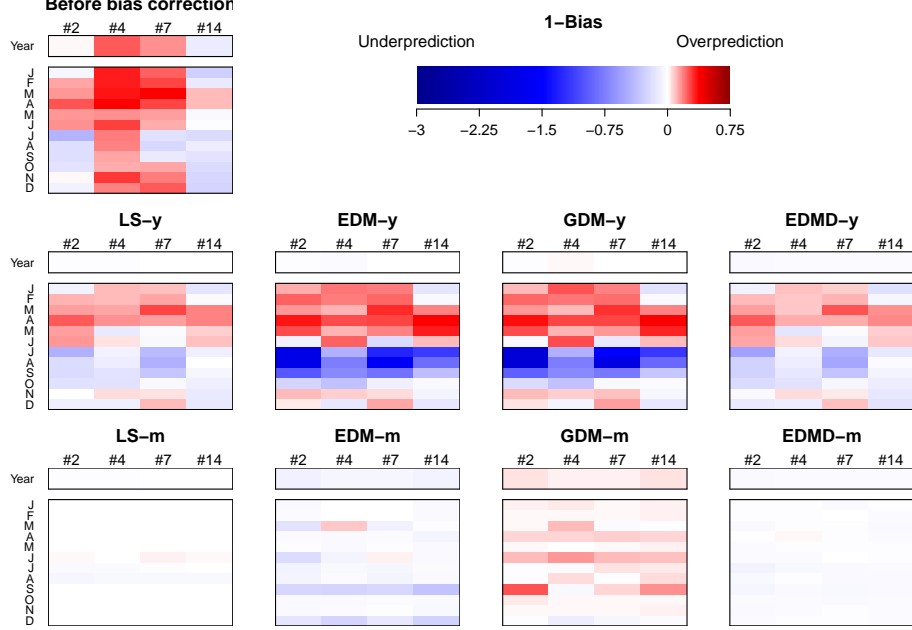

**Figure 6.** Deviation of the precipitation bias from 1, for catchments 2, 4, 7 and 14, over the 1981-2010 period. The deviation is shown for the whole year (top line) and for each calendar month. The bias is only shown for lead times between 31 and 60 days. Blue-shaded areas (negative values) represent a tendency of underpredicting precipitations and red-shaded areas (positive values) a tendency of overpredicting precipitations. The top left graph represents the bias of raw precipitation forecasts, and each of the other graphs represents the bias after applying one of the bias correction methods.

lead times of the forecast range, we observed that the biases vary more with the calendar month of the forecast horizon than with lead time. For this reason, we only show the 2-month lead time.

In general, seasonal forecasts tend to overpredict precipitations over the year in most catchments. Overprediction tends to occur near the end of the winter (rainy) season and throughout the spring season. Conversely, precipitations tend to be

5 underpredicted from the end of the summer (dry) season and until the beginning, and sometimes throughout, the autumn season. The four selected catchments illustrate the variety of conditions we encountered in the bias correction analysis. In catchment 2, precipitations could be considered unbiased when carrying the analysis over the year. However, this result hides monthly underpredicting and overpredicting biases which compensate over the year. In this catchment, forecasts tend to overpredict from February to June and underpredict from July to October. The yearly result may also be a reflection of the lack of important

10 biases in the months of December and January, which are, climatologically, the rainiest months in this catchment. This type of variation in bias was also observed in catchments 6, 11, 12 and 13. In catchment 4, precipitation forecasts are strongly overpredicting observations in all calendar months and thus over the year. This catchment stands out because in no other catchment do we observe a similarly strong and systematic bias. This catchment is the one located at the most eastern part of France. Its main river (l'Ill) is a tributary of the Rhine river. It has its sources in the Jura mountains and receives several



tributaries from the Vosges mountains. In catchment 7, precipitations are overpredicted over the year, with the strongest positive deviations concentrated during the rainy season, basically from November to April. The same behaviour is found in catchments 5, 10 and 15. Interestingly, catchments with a clear overprediction, i.e. catchments following the patterns depicted in Fig. 6 for catchments 4 and 7, correspond to the catchments in which System 4 raw precipitation and streamflow forecasts showed

low skill in sharpness and/or overall performance. Last, catchment 14 is representative of catchments 1, 3, 8, 9 and 16 in the database. Forecasts slightly underpredict precipitations over the year, with a tendency to underpredict precipitations in all seasons but the spring season, whose precipitations are slightly overpredicted.

Figure 6 also presents the remaining biases after the application of the eight bias correction methods to the raw precipitation forecasts. We present the results over the whole year and for each month. The same four selected catchments illustrate the

results for the 2-month lead time. All correction methods are effective to correct biases of precipitation forecasts over the year. However, this is not observed in the bias correction for each calendar month. Results for the methods calibrated on a yearly basis (LS-y, EDM-y, GDM-y, EDMD-y) show that the absence of bias over the year is mainly achieved through an effect of compensation between over and underprediction among the calendar months. Particularly EDM-y and GDM-y methods show a strong pattern of monthly biases, even after bias correction, towards overprediction of precipitations in winter and spring, and

underprediction in summer and autumn.

By construction, monthly calibrated methods perform much better when looking at monthly biases. LS-m and EDMD-m are particularly effective in all catchments. Forecasts corrected with EDM-m tend to slightly underpredict precipitations, while forecasts corrected with GDM-m tend to overpredict precipitations. This may be an effect of the application of distribution mapping based on monthly values. Distribution mapping requires that the time structure of forecast and observed precipitation

are coherent, so that upper forecast values are shifted towards upper observed values and conversely. However, raw monthly forecast means from System 4 do not always reproduce the time structure of monthly observations and often fail to reach extreme monthly values. Therefore, correction factors obtained with a distribution mapping based on monthly values show poorer performance, and the method can wrongly increase or decrease daily precipitation values.

### 5.2 Comparison of bias correction factors for LS and EDMD methods

The LS and EDMD methods showed more effectiveness in reducing bias in the precipitation forecasts. In order to better understand how the two methods compare, we plotted in Fig. 7 their correction factors for catchment 7 over the 1981-2010 period for the 2-month lead time. Black lines represent correction factors from LS. Each day, one correction factor is applied to all members of the ensemble forecast at the 2-month lead time. Grey-shaded areas represent the range of correction factors applied with EDMD, and darker grey lines represent the median correction factor. For EDMD, each precipitation value has a

specific correction factor depending on its probability of occurrence. Therefore, for a given day and lead time, the number of correction factors is equal to the number of ensemble members.

LS-y provides relatively constant bias correction factors over the study period. Since, on average, precipitations in catchment 7 are overpredicted by System 4 forecasts, this correction factor is smaller than 1. The bias correction factors are obtained with the one-year-leave-out calibration framework. It is interesting to note that removing one year within the 30 years of the





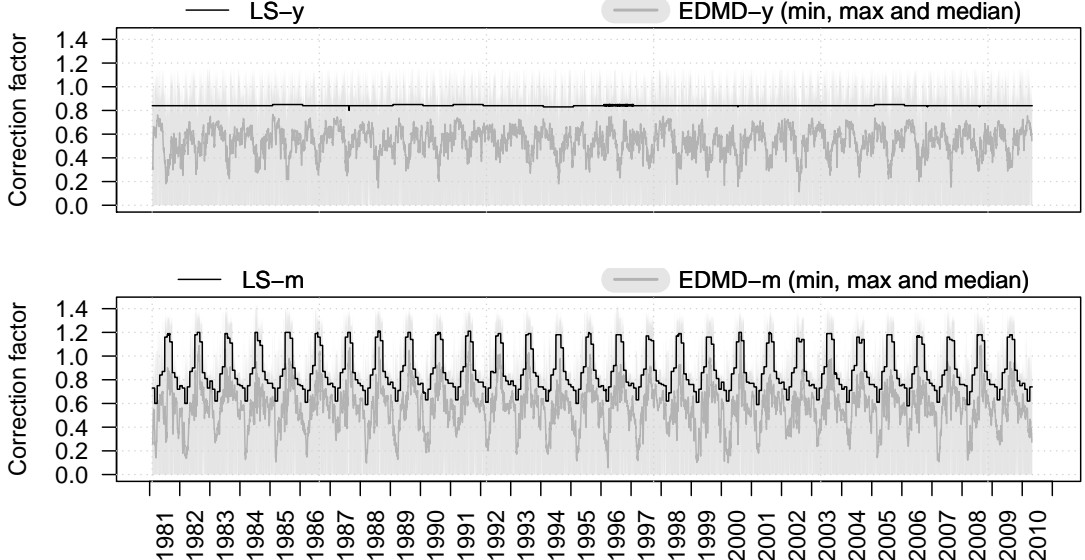

**Figure 7.** Bias correction factors applied to each day of the record period with the LS and EDMD methods. Correction factors are only shown in the case of catchment 7 and for the second month lead of the precipitation forecasts. The top graph presents correction factors obtained with LS and EDMD calibrated over the whole year, and the bottom graph presents correction factors obtained with LS and EDMD calibrated monthly.

calibration period has little impact over the calibrated correction factors, even for an extreme dry year such as 1989 in this catchment. With EDMD-y, correction factors vary for each day of the study period. These factors remain smaller or close to 1. Their median values are smaller than the LS-y correction factors and the maximum values are slightly greater than the LS factors. When calibrated monthly, correction factors obtained with LS-m depict a variation, ranging from 0.6 to 1.2. They

5 present a recurring pattern over the year, which follows what was shown in Fig. 6, i.e., that precipitations in catchment 7 are, on average, overpredicted during the winter and spring seasons, leading to correction factors smaller than 1, and underpredicted from July to September, leading to bias correction factors greater than 1. This pattern in the factors indicates that the LS method might be further simplified to provide correction factors that would solely vary with the calendar month, regardless of the year, or in the case of LS-y, be constant over the target period. Correction factors computed with EDMD-m present a

10 similar pattern to the one observed with LS-m, but their range is more variable, with values between 0 and 1.4. This method is particularly interesting because, as opposed to LS, it also corrects the frequency of precipitation days, given the null values of some correction factors.



### 5.3 Impact of bias correction on the useful forecasting lead time

The four criteria used to evaluate reliability, accuracy, sharpness and overall performance were applied to the precipitation forecasts bias corrected with each of the eight bias correction methods. They were also applied to the seasonal streamflow forecasts generated from inputting the different bias corrected precipitation forecasts to the hydrological model. Skill scores

were computed with the raw seasonal precipitation forecasts as reference forecast for precipitation, and with the (raw) streamflow forecasts generated from raw precipitation forecasts as reference forecast for streamflow. For each variable (precipitation and streamflow), each criterion, each bias correction method, each catchment and each season, we obtained the corresponding UFL (Useful Forecasting Lead time). We then evaluated the proportion of catchments falling in each UFL group (as defined in Section 3.3.3). Results are shown in Fig. 8 and Fig. 9, for precipitation and streamflow forecasts, respectively.

In Fig. 8, the two bias correction methods that stand out regarding overall performance (CRPS), in all seasons, are LS and EDMD. This is in accordance with our previous results on the efficiency of each method to correct biases. When looking more closely at improvements in the PIT criterion, as measured by the UFL, EDMD clearly stands out from the other methods. The proportion of catchments with skill improvement over raw forecasts is almost always 100%, and skill is often extended up to 60 days and more. The other methods are quite equivalent to each other, although LS performs slightly better, with greater

improvements in larger proportions of catchments, especially in winter and spring, for reliability (PIT), accuracy (MAE) and overall performance (CRPS). In terms of sharpness (IQR), the best performing method varies with the season. Precipitation forecasts in spring (MAM) are sharper when corrected with methods calibrated monthly, while forecasts in summer and autumn are sharper with methods calibrated yearly. To effectively address the tendency to overestimate spring precipitations, the multiplicative correction factor of a monthly calibrated bias correction for the spring season will be smaller than 1, and much

smaller than the correction factor obtained with a yearly calibrated correction. Therefore, the spring interquartile range will be further reduced by the method calibrated monthly than by the method calibrated yearly. This reasoning only applies to LS, EDM and GDM since EDMD corrects each ensemble member independently.

Figure 9 shows the results for the streamflow forecasts. LS and EDMD methods are able to extend the lead time of bias corrected predictions further than other methods, and for a higher proportion of catchments in the large majority of seasons

and criteria. Again, EDMD methods yield the best improvements in reliability. LS yields results slightly better than EDMD in sharpness and accuracy. EDM and GDM clearly have lower performance, except in some cases in sharpness and for spring and summer.

### 5.4 Summary of the comparison of bias correction methods

In general, LS and EDMD bias correction methods show good performance for precipitation forecasts, although in a distinct

way. While EDMD clearly improves forecast reliability, LS shows better performance in improving sharpness. In terms of streamflow forecasts, LS and EDMD are the methods that offer the best performance. Again, EDMD may be preferred if focus is placed on forecast reliability, while LS may be preferred if sharpness and accuracy are the criteria one is looking to improve. Since streamflow forecasts generated from raw System 4 precipitation forecasts are already, in most of the studied catchments,





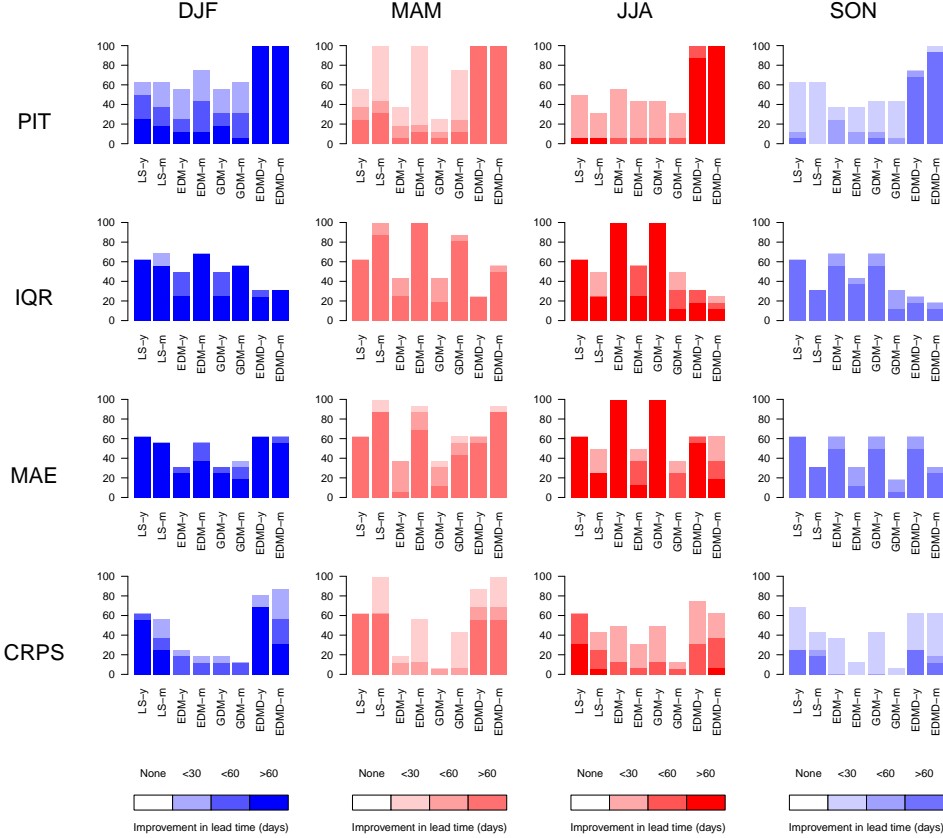

**Figure 8.** Number of catchments (%) in each UFL value category, i.e. number of catchments in which bias corrections increase the lead time up to which seasonal precipitation forecasts have skill in regards to raw seasonal precipitation forecasts. Each row corresponds to an evaluation criterion and each column corresponds to a season. Colour shades indicate the UFL category, i.e. the lead time up to which precipitation forecasts are improved.

sharper than the ESP reference, but lack reliability (as shown in Fig. 4 and Fig. 5), it seems appropriate to give priority to a correction method that improves reliability, while providing good overall performance. Therefore, in the following, we will only consider the monthly calibrated version of EDMD (EDMD-m) to further investigate the skill of bias corrected seasonal forecasts in the 16 selected French catchments. The monthly version is chosen to ensure that monthly biases are removed and that the correction will perform relatively equally in all seasons, while avoiding the "mis-estimation" of forecast skill (Hamill and Juras, 2006).



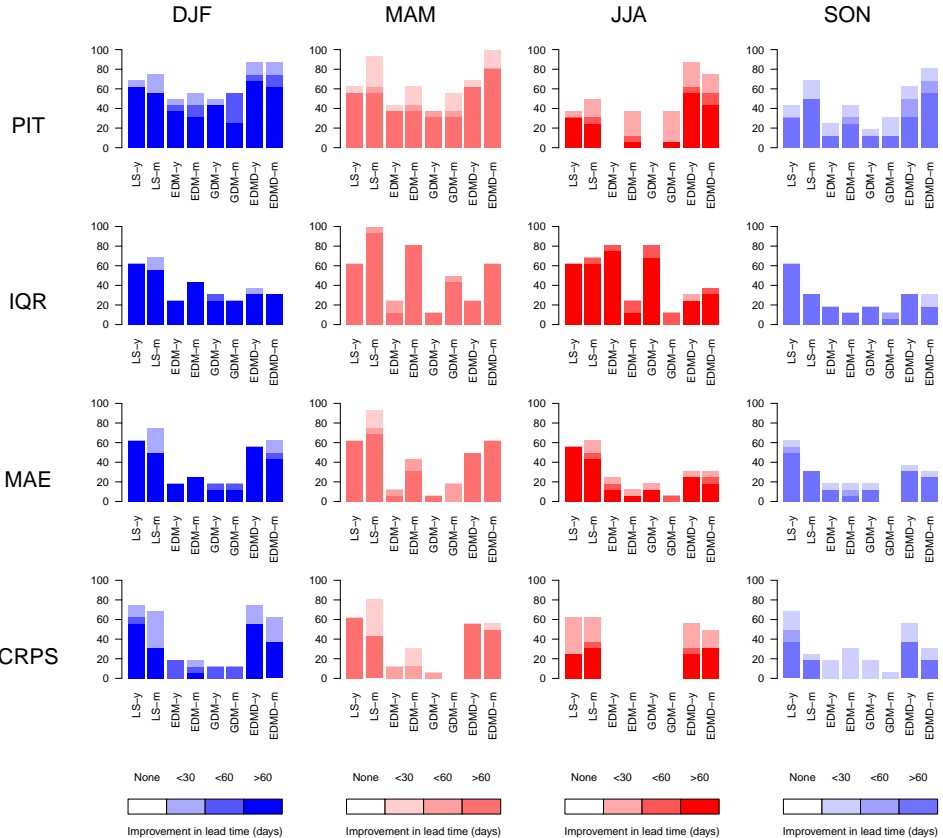

**Figure 9.** Number of catchments (%) in each UFL value category, i.e. number of catchments in which bias corrections increase the lead time up to which seasonal streamflow forecasts have skill in regards to seasonal streamflow forecasts generated from raw seasonal precipitation forecasts. Each row corresponds to an evaluation criterion and each column corresponds to a season. Colour shades indicate the UFL category, i.e. the lead time up to which streamflow forecasts are improved.

## 6  Skill scores of bias corrected seasonal forecasts

### 6.1  Performance of bias corrected precipitation forecasts

Figure 10 (for sharpness and overall performance) and Fig. 11 (for reliability) present the skill of seasonal precipitation forecasts bias corrected with EDMD-m. Skill scores are computed with historical precipitation as the reference. In order to better evaluate the impact of bias correction on forecast skill, the y-axes in Fig. 10 are the same as in Fig. 2. The comparison of these two figures shows that bias correcting the raw System 4 forecasts reduces the differences in skill between catchments. After bias correction, catchments present very similar evolutions of the skill with the lead time. We can also infer that, after bias correction, in some catchments, the values of IQR and CRPS are lower than before bias correction. Nevertheless, bias





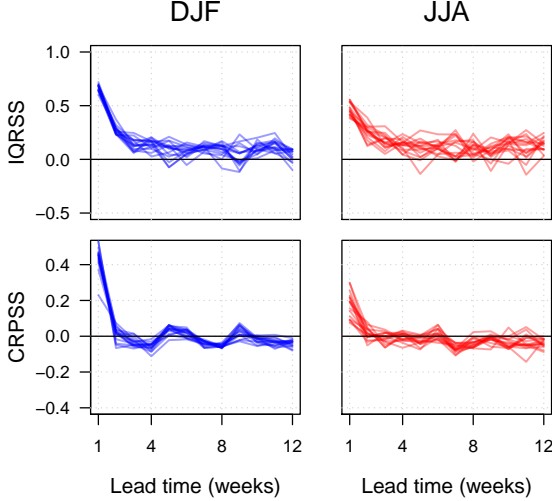

**Figure 10.** Skill of weekly precipitation forecasts corrected with EDMD-m as a function of the lead time for all catchments and all seasons. The skill is computed based on the IQR (top) and the CRPS (bottom) and the reference is historical precipitations. Each column corresponds to a target season. Each line represents the skill score in a catchment for forecast horizons within the target season.

corrected forecasts remain sharper than the reference (i.e., skill scores are always greater than zero). In the catchments where the raw forecasts performed worse than historical precipitations (i.e., skill scores lower than zero in Fig. 2), bias corrected forecasts become sharper and gain skill in regards to the reference. Forecast skill in overall performance (CRPSS) is observed up to two to three weeks ahead, after which forecasts attain skill equal to that of the reference forecast. Skill is improved in catchments that performed worse than the reference prior to bias correction (i.e., skill scores lower than zero in Fig. 2). Figure 10 illustrates these findings for winter (DJF) and summer (JJA), but results are similar for spring and autumn (not shown).

Figure 11 shows that the most remarkable improvement in performance due to bias correction is achieved in reliability. While precipitation forecasts had a tendency to overpredict prior to bias correction, bias corrected precipitations are reliable in all catchments. Figure 11 shows the results for winter and summer, and for lead times of 30 and 90 days, but conclusions are similar in the other seasons and lead times (not shown). Even though a slight tendency to overpredict precipitations remains in winter for short lead times, the improvements are noticeable. The EDMD-m bias correction was able to address the concentration of points in the zero end point observed in Fig. 3 for the raw forecasts.

## 6.2 Performance of bias corrected streamflow forecasts

The quality of the streamflow forecasts generated from the precipitation forecasts corrected with EDMD-m is investigated in Fig. 12 and Fig. 13 (IQRSS and CRPSS) and in Fig. 14 (PIT diagrams). These figures can be compared to Fig. 4 and Fig. 5 which were obtained from the analysis of streamflow forecasts generated from raw precipitation forecasts. As seen with precipitation forecasts, bias correction also reduces the differences in streamflow forecast skill between catchments and seasons





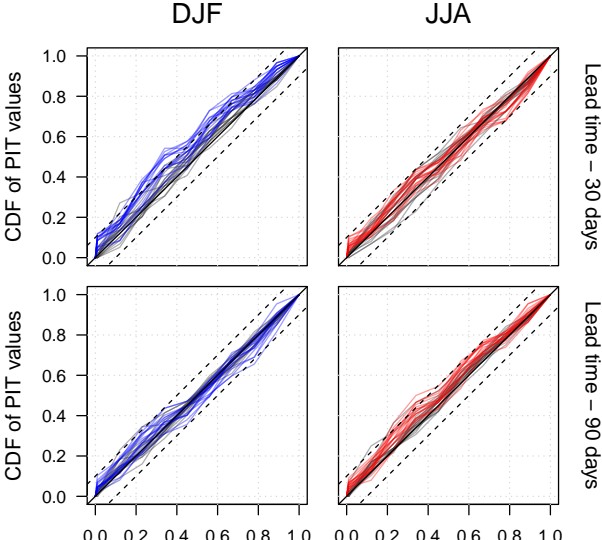

**Figure 11.** PIT diagram of precipitation forecasts corrected with EDMD-m (coloured lines) and historical precipitations (grey lines) for lead times of 30 days (top) and 90 days (bottom). Each column corresponds to a target season. Each line represents the PIT diagram in a catchment for forecast horizons within the target season.

(Fig. 12). Again, this translates into a loss in skill in catchments with the sharpest ensemble forecasts before bias correction, but also in a gain in skill in catchments where raw streamflow forecasts had negative skill. Overall, after bias correction, streamflow forecasts are sharper than ESP in all catchments and seasons (only the winter and summer seasons are shown but results are similar for the spring and autumn seasons). In terms of overall performance (CRPSS), the skill of streamflow forecasts was largely improved, especially in catchments that had very low skill prior to bias correction (i.e., CRPSS values well below zero in Fig. 4). In winter, autumn and spring, skill over the ESP reference is observed up to four weeks ahead in several catchments (even up to five weeks ahead in spring and autumn), while in summer, it is observed up to two to three weeks. At longer lead times, streamflow forecasts show an overall performance equivalent or slightly lower than the performance of the ESP method. Some studies use past streamflow observations (referred to as streamflow climatology) as the reference forecast to assess the skill of streamflow forecasts (e.g. Trambauer et al., 2015; Wetterhall et al., 2015). Figure 13 shows the skill in overall performance and sharpness when streamflow climatology is used as reference to calculate the skill of EDMD-m bias corrected forecasts. As expected, streamflow forecasts generated from bias corrected precipitation forecasts are sharper and present better overall performance than streamflow climatology, even for lead times of up to twelve weeks in some catchments. In one catchment (catchment 1), skill scores are systematically higher than the scores of the other catchments. In this catchment, streamflow climatology is very wide, with interannual variability of the same order of magnitude as interseasonal variability.

The PIT diagrams in Fig. 14 show that the reliability of streamflow forecasts is also improved after bias correcting precipitation forecasts. In winter (DJF) and spring (not shown), streamflow forecasts are now reliable and equivalent to ESP, although



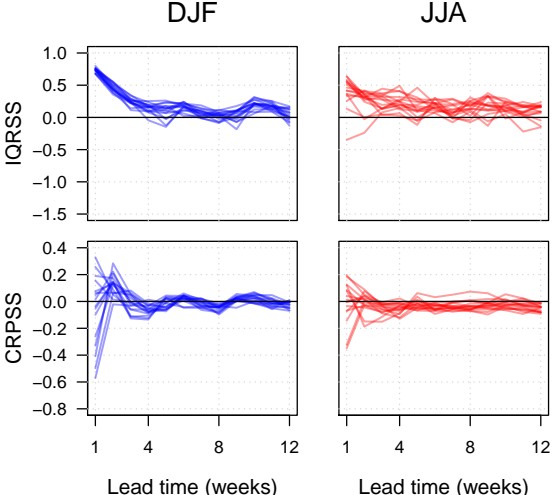

**Figure 12.** Skill of streamflow forecasts obtained from precipitation forecasts corrected with EDMD-m as a function of the lead time for all catchments and all seasons. The skill is computed based on the IQR (top) and the CRPS (bottom) and the reference is Ensemble Streamflow Prediction. Each column corresponds to a target season. Each line represents the skill score in a catchment for forecast horizons within the target season.

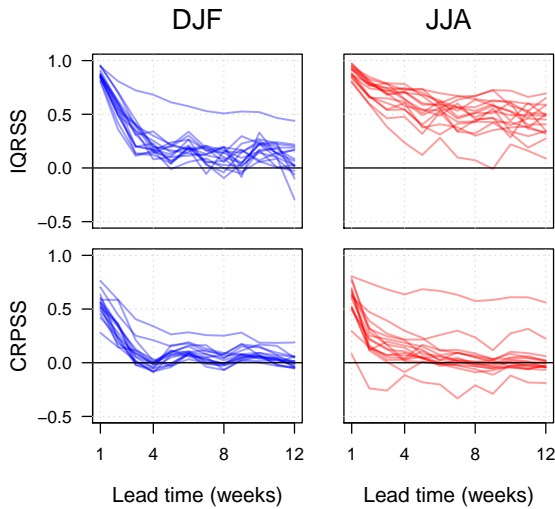

**Figure 13.** Skill of EDMD-m debiased streamflow forecasts as a function of the lead time for all catchments and all seasons. The skill is computed based on the IQR (top) and based on the CRPS (right) and the reference is historical streamflow. Each column corresponds to the target season of forecast lead times. Each plotted line represents the performance of a catchment.

forecasts still show a slight tendency to overpredict streamflows. In autumn (not shown), streamflow forecasts are also reliable in most catchments, but with a tendency to underpredict streamflows. Summer (JJA) streamflow forecasts are also more reliable





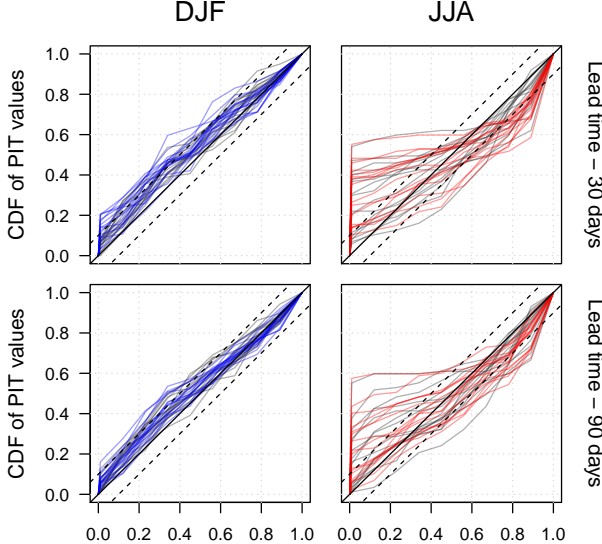

**Figure 14.** PIT diagram of streamflow forecasts obtained from precipitation forecasts bias corrected with EDMD-m (coloured lines) and Ensemble Streamflow Prediction (grey lines) for lead times of 30 days (top) and 90 days (bottom). Each column corresponds to a target season. Each line represents the PIT diagram in a catchment for forecast horizons within the target season.

than they were prior to bias correction, but they still depict poor reliability and show that there is room for improvements. As shown by other studies in ensemble forecasting (Zalachori et al., 2012; Verkade et al., 2013; Roulin and Vannitsem, 2015), a simple bias correction of meteorological inputs is obviously not enough to achieve streamflow forecast reliability. In our case, the difficulties of the hydrological model in reaching lower streamflow values remain. This highlights the need for taking into account other sources of hydrological modelling uncertainties and including additional post-processing, targeting directly streamflow forecasts.

### 6.3 How improvements in precipitation forecasts propagate to streamflow forecasts?

We have seen that the use of reliable precipitation forecasts as input to a hydrological model does not automatically generate reliable streamflow forecasts. In order to further understand how improvements in precipitation forecasts propagate to streamflow forecasts, we compared the skill scores of EDMD-m bias corrected precipitation forecasts with the skill scores of the streamflow forecasts generated from these bias corrected precipitations. We focused the analysis on the four catchments previously selected as representative of the database, i.e. catchments 2, 4, 7 and 14.

Figure 15 presents the results for the CRPSS, IQRSS and the PITSS (PIT area) in these four catchments. The reference forecast for the computation of the skill scores of the bias corrected forecasts is the raw forecast. The skill thus represents a measure of the improvement due to bias correction. Skill scores were averaged over lead times of 10 days to 90 days.





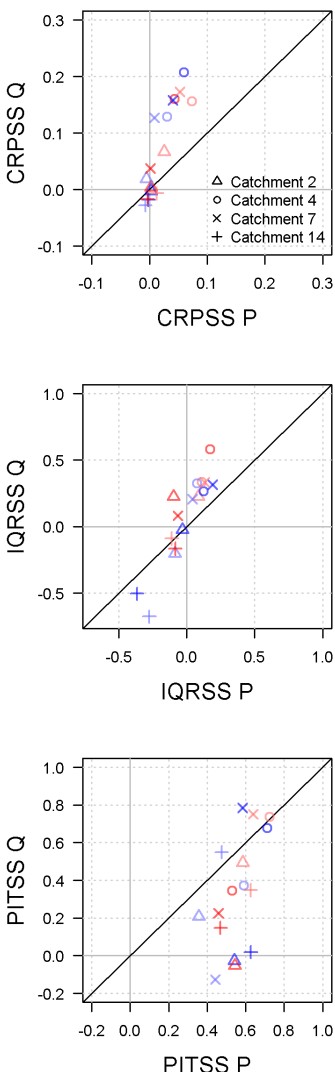

**Figure 15.** Skill scores of streamflow forecasts after correction with EDMD-m against skill scores of precipitation forecasts after correction with EDMD-m. The skill score of forecasts corrected with EDMD-m is computed in regards to raw forecasts. It is then averaged over lead times 10 to 90 days to obtain a single value. Results are shown for all four seasons in four selected catchments (Catchments 2, 4, 7 and 14). Skill scores were obtained based on the CRPS (top), the IQR (middle) and the PIT diagram area (bottom). The 1:1 diagonal corresponds to an equivalent performance increase in precipitation and streamflow.

In overall performance (CRPSS), bias correcting precipitation forecasts either led to a gain in skill in both precipitation and streamflow forecasts, as in catchments 4 and 7 and in some seasons in catchment 2, or to a skill equivalent to the skill prior to bias correction, as in catchment 14. Since catchments 4 and 7 were the ones with the most biased forecasts (cf. Fig. 6), there was





more room for improvement in these catchments. Catchment 14 had the smallest bias of the four catchments. Bias correction had thus little impact on precipitation forecasts, and therefore also on streamflow forecasts. Interestingly, the improvement achieved in streamflow is always superior to the improvement achieved in precipitation, or equivalent when there was no gain in skill. It seems therefore that a small improvement in the overall performance of precipitation inputs (as measured by the

CRPS) can translate in a greater improvement in streamflow forecasts.

If we look at the skill in sharpness (IQRSS) and in reliability (PITSS) of the ensemble forecasts, we observe different behaviours. In sharpness, a loss in skill was observed in catchments 2 and 14, while a gain was observed in catchments 4 and 7. When a gain was achieved, the gain is superior in streamflow forecasts than in precipitation forecasts. If we look at reliability, skill was always improved by bias correcting the precipitation forecasts, with skill scores always superior to 0.3. The gain in

streamflow is mainly positive, but not always, as in the case of precipitation forecasts. Although the majority of skill scores are superior to 0.1, some values are below the zero skill score line. The gain in reliability from the application of bias correction to raw precipitation forecasts is, in general, superior in precipitation forecasts than it is in streamflow forecasts.

Based on our results, we can say that in catchments with small biases, here represented by catchments 2 and 14, overall performance was mainly stable from precipitation to streamflow forecasts. However, in these catchments, a gain in reliability

was generally associated with a loss in sharpness. In catchments with greater biases, here represented by catchments 4 and 7, overall performance, sharpness and reliability were improved for both precipitation and streamflow forecasts by simply bias correcting the precipitation forecasts.

## 6.4  Example of forecast hydrographs in a selected catchment

Figure 16 presents the hydrographs of the forecasts obtained from historical streamflow (HistQ), ESP, and seasonal forecasts

bias corrected with LS-m and EDMD-m, from April 2004 to April 2007 in catchment 7. We show forecasts for lead times from 31 days to 60 days, i.e., forecasts issued in the previous month. Ensemble forecasts are represented by the median forecasts and two prediction intervals: the 25%-75% interval containing 50% of the ensemble members (dark grey zone), and the 5%-95% interval with 90% of the ensemble members (light grey zone). Observed streamflow is also shown. In this catchment, seasonal forecasts had a strong bias and bias correction methods performed well.

The hydrograph for historical streamflow represents the interannual variability in streamflow in the catchment, except that the forecast year is excluded for cross-validation. It relies on past observations of streamflow and does not include seasonal meteorological forecasts. We can see that the observations fall within the forecast ranges in most cases, which indicates, as expected with climatology, good forecast reliability. However, the forecast lacks sharpness during low-flow periods. Accuracy of the median forecast is, in general, good, although too high and low peak flows are not well reproduced.

The forecasts obtained with the ESP method use past observations of precipitation as input to the hydrological model rather than seasonal meteorological forecasts. They show visible improvements in sharpness during low flow periods, while reliability seems preserved. Accuracy of the median forecasts seems equal or lower than observed with historical streamflow.

The hydrographs representing the streamflow forecasts obtained from bias corrected System 4 precipitation seasonal forecasts show forecasts that are sometimes even sharper than ESP forecasts, as seen, for instance, for the rising limb in 2005.





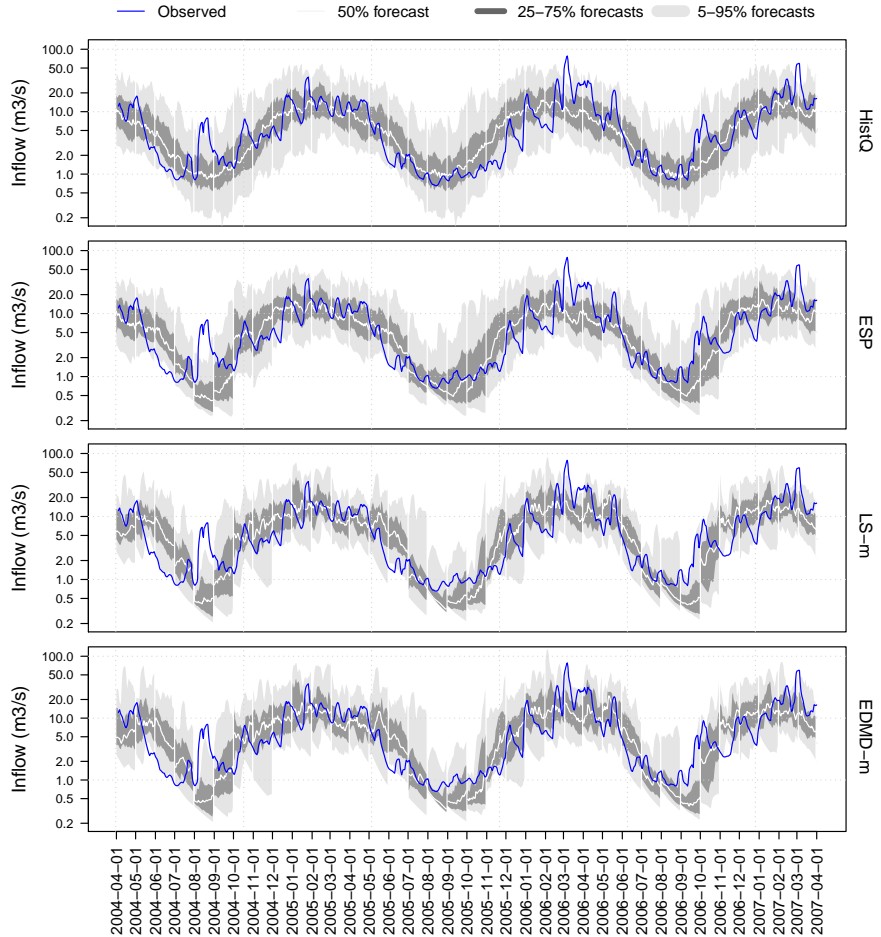

**Figure 16.** Hydrographs obtained with historical streamflow, ESP, seasonal forecasts corrected with LS-m and seasonal forecasts corrected with EDMD-m in catchment 7 from 1 April 2004 until 1 April 2007. The vertical axis is logarithmic. The blue line represents the observed streamflow. The grey shaded areas present the forecasts issued in the previous month, i.e. 31 to 60 days prior to the observations.

Overall, the observed streamflow falls within the forecast ranges. In some situations, as in the peak event in August 2004, prediction intervals of bias corrected seasonal forecasts, particularly in the EDMD-m case, are closer to observations than ESP forecasts. In general, visual differences in quality between seasonal streamflow forecasts obtained from precipitation forecasts corrected with LS-m and EDMD-m are hardly noticeable. Although EDMD-m forecasts seem to present slightly larger predic-

5    tion intervals, which could result in better reliability but lower sharpness comparatively to LS-m, the accuracy of their median forecasts is practically identical. The visual inspection of these graphs for all catchments indicates similar results. Although our analyses and evaluation criteria have indicated the EDMD-m as the preferred method for the studied catchments, LS-m



also yields good improvements in precipitation and streamflow forecasts. Since this method is easier to implement, it can be an alternative to the application of EDMD-m in operational forecasting systems.

## 7 Discussion and conclusions

We assessed the quality of ECMWF System 4 precipitation forecasts for seasonal streamflow forecasting in 16 catchments

in France. We evaluated areal precipitation forecasts over the catchments and streamflow forecasts generated from inputting precipitation forecasts to a lumped hydrological model. Results show that, in most catchments, raw (uncorrected) System 4 precipitation forecasts are sharper than precipitation climatology (i.e., ensemble forecasts built from past observed precipitations) in all seasons. However, raw precipitation forecasts show poor reliability and a tendency to overpredict precipitations. Likewise, streamflow forecasts generated from raw System 4 precipitations are sharper, but far less reliable than forecasts based

on the ESP approach (i.e., ensemble forecasts obtained from running the hydrological model with current initial conditions and past observed precipitations). Yet, in overall performance, raw precipitation forecasts yield improvements up to two weeks in all catchments over precipitation climatology, and streamflow forecasts yield improvements up to three to four weeks over ESP in some catchments. In general, improving forecast reliability, while maintaining (or not diminishing too much) forecast sharpness, was clearly a challenge for bias correction methods.

An in-depth analysis of the biases of System 4 seasonal precipitation forecasts showed strong monthly biases sometimes hidden at the scale of the year, depending on the catchment. Bias correction methods calibrated over the whole year were therefore less efficient when evaluating forecasts over calendar months. In the majority of catchments, the empirical distribution mapping of daily values (EDMD) or the simple linear scaling method (LS) applied to raw System 4 precipitation forecasts showed more effectiveness in correcting the yearly but also the monthly biases. These methods also gave the highest increase

in overall performance for streamflow forecasting. Empirical distribution mapping of daily values calibrated for each calendar month (EDMD-m) was particularly efficient to increase reliability of precipitation and streamflow forecasts, while linear scaling (LS-m) led to higher improvements in sharpness and accuracy.

The EDMD-m bias correction method was further investigated to better understand its impact on the skill of bias corrected seasonal forecasts in the studied catchments. Overall, the application of bias correction reduced the differences in forecast

performance between seasons and catchments for precipitation and streamflow forecasts. Also, bias correction ensured that precipitation and streamflow forecasts were at least equivalent in performance to the historical precipitations and streamflow forecasts based on historical precipitations, respectively, up to three months ahead. In catchments with greater biases, overall performance, sharpness and reliability were improved for both precipitation and streamflow forecasts by simply bias correcting the precipitation forecasts. Overall performance was mainly stable in catchments with small biases. However, in these catch-

ments, a gain in reliability was generally associated with a loss in sharpness. The evaluation of forecasts after bias correction, for the purposes of operational applications on water and risk management, may therefore involve a trade-off between sharpness and reliability. Furthermore, while precipitation forecast reliability is improved with bias correction, the evaluation of streamflow forecast reliability shows that there is still room for improvement. Notably, bias correction of precipitation inputs




was not enough to achieve good reliability in summer streamflow forecasts. This highlighted the need for adding a step of streamflow post-processing to the forecasting system.

This study compared eight simple bias correction methods to correct precipitation seasonal forecasts and investigated how one of them impacts the skill of streamflow forecasts. The catchments studied were not influenced by snowmelt flows and thus
only precipitation was considered in the bias correction procedures. In other contexts, it may be interesting to also include bias correction of temperature forecasts, with appropriate methods to consider space-time interdependencies of the meteorological variables. The explicit consideration of temperature forecasts could also benefit the skill of low flow forecasts in summer, when evapotranspiration can play a crucial role.

Several other approaches for post-processing and bias correction exist, for instance, based on MOS techniques, space-time
disaggregation schemes or Bayesian Model Averaging (Gneiting et al., 2005; Raftery et al., 2005; Liu et al., 2013; Hemri et al., 2014). These could be investigated to contribute to the comprehensive comparison of options for bias correcting precipitation and temperature forecasts prior to seasonal streamflow forecasting.

Last, other forecasting methods selecting historical precipitations based on climate indicators have been investigated in the literature for seasonal hydrological forecasting in regions where strong correlations have been observed, e.g. in the United
States or in Australia. In France, weak correlations have often shown that climate indicators may not be adapted to forecast precipitations at the seasonal scale. However, the use of indicators derived from seasonal forecasts could potentially improve the selection of past precipitation scenarios, which might enhance the skill of ESP methods to forecast streamflow.

*Acknowledgements.* This work was partly funded by the Interreg IVB NWE programme of the European Union, project DROP (Benefit of governance in DROught adaptation). The first author acknowledges Dr. Christopher A. T. Ferro for his insights on probabilistic scores.





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





**Table 1.** Number, name, surface, and mean annual precipitation, potential evapotranspiration and streamflow for the studied catchments.

| # | River | Gauging station | Surface (km$^2$) | Mean annual precipitation (mm/yr) | Mean annual evapotran- spiration (mm/yr) | Mean annual flow (mm/yr) |
|---|---|---|---|---|---|---|
| 1 | Andelle | Vascoeuil | 377 | 952 | 628 | 332 |
| 2 | Orne Saosnoise | Montbizot [Moulin Neuf Cidrerie] | 501 | 735 | 696 | 163 |
| 3 | Briance | Condat-sur-Vienne [Chambon Veyrinas] | 605 | 1100 | 706 | 427 |
| 4 | Ill | Didenheim | 668 | 956 | 664 | 309 |
| 5 | Azergues | Lozanne | 798 | 931 | 689 | 296 |
| 6 | Seiche | Bruz [Carcé] | 809 | 732 | 696 | 181 |
| 7 | Petite Creuse | Fresselines [Puy Rageaud] | 853 | 899 | 680 | 316 |
| 8 | Sèvre Nantaise | Tiffauges [la Moulinette] | 872 | 898 | 712 | 331 |
| 9 | Vire | Saint-Lô [Moulin des Rondelles] | 882 | 958 | 629 | 448 |
| 10 | Orge | Morsang-sur-Orge | 934 | 658 | 680 | 131 |
| 11 | Serein | Chablis | 1119 | 842 | 675 | 220 |
| 12 | Sauldres | Salbris [Valaudran] | 1220 | 803 | 684 | 240 |
| 13 | Eyre | Salle | 1678 | 1025 | 785 | 323 |
| 14 | Arroux | Etang-sur-Arroux [Pont du Tacot] | 1792 | 981 | 655 | 390 |
| 15 | Meuse | Saint-Mihiel | 2543 | 948 | 639 | 372 |
| 16 | Oise | Sempigny | 4320 | 805 | 639 | 250 |

**Table 2.** Bias corrections applied: corresponding abbreviations, method used for calibration and description.

| Abbreviation | Calibration based on | Description |
|---|---|---|
| LS-y | the whole year | Linear scaling of monthly values |
| LS-m | calendar months | |
| EDM-y | the whole year | Empirical distribution mapping of monthly values |
| EDM-m | calendar months | |
| GDM-y | the whole year | Gamma distribution mapping of monthly values |
| GDM-m | calendar months | |
| EDMD-y | the whole year | Empirical distribution mapping of daily values |
| EDMD-m | calendar months | |