# Peer review of "Bias correcting precipitation forecasts to improve the skill of seasonal streamflow forecasts"

_Hydrology and Earth System Sciences, 2016_

## Referee Comment (RC1) · M. Zappa (Referee) · 23 Mar 2016

See attached document.

Needed revisions are in my opinion between minor and major.

I finally opted for major, since I want to re-evaluate the PET point.

Best regards Massimiliano Zappa

WSL, Birmensdorf

Please also note the supplement to this comment:

[Figure]

http://www.hydrol-earth-syst-sci-discuss.net/hess-2016-78/hess-2016-78-RC1-supplement.pdf

---

## Referee Comment (RC2) · M. Zappa (Referee) · 23 Mar 2016

**Bias correcting precipitation forecasts to improve the skill of seasonal streamflow forecasts**

**Louise Crochemore, Maria-Helena Ramos, and Florian Pappenberger**

The authors are presenting a generally well-conceived study focussed on possible improvement of seasonal streamflow forecasts by applying bias correction to the forcing precipitation forecasts. I found the manuscript well written, with appealing and adequate artwork and well discussed results. I found that the whole manuscript well supports the results the authors are enumerating. The scores the chose are adequate and the combination of scores allows drawing conclusions that are not blended by the scores that show maximal improvement with respect to the defined references.

It is not clear which PET forcing is used in forecasts mode. If the authors use the observation-based reanalysis of PET in combination with the precipitation forecasts, then the appeal of this study would be quite reduced, in my "operational-minded" opinion.

Furthermore, the discussion and conclusions section is not adequately considering previous studies.

Issues to be addressed (Page(s) – Line(s)):

General comments:

While the Introduction is well balanced and gives useful insight on previous work on the topic and also references supporting the envisaged methodology, I found that the final paragraphs should possibly include more information on the novelty of the present manuscript. Also in the methodological section some more referencing is needed. See minor comments for this.

4 – 4-15: We learn here about the meteorological forcing. It is clear to me how you use precipitation, but as a forecast and as SAFRAN product. Concerning Potential Evapotranspiration (PET), only SAFRAN is declared. I'd like you to declare which PET is used in retrospective forecasts forced by the ECMWF products. If it is from ECMWF, you should state why you are not post-processing it. If you use SAFRAN, you should be able to assess how much uncertainty are you neglecting by using the best observed estimates of PET instead of using a forecasted value (which you need to do as soon as you will deploy the system in real-time). In our experience, for basins not affected by snow-melt, the post-processing of relative-humidity data (an important proxy the evaporation demand by the atmosphere) helps improving the estimation of hydrological droughts (Jörg-Hess et al., 2015).

4-25: I just reviewed another paper on seasonal forecasting where authors did not show any score concerning their calibration/validation and I amended it. Same here. I am happy with a table as supplementary material.

24 – 18: I like this evaluation very much, just, I miss some quantification supporting the description based on visual inspection you are giving. Be pragmatic.

25 -3: The discussion section is here quickly merged with the conclusions. The only link to current literature one is expecting here merely consists in a enumeration of possible post-processing of the forecasts with currently available methods. Here some more effort has to be shown to make also this section a valuable part of the manuscript.

26 – 2: You address here the issue of implementation in operational systems. Again, declare how you deal the PET, and then re-evaluate the potential for real-time operations.

Minor comments (Page(s) – Line(s)):

2 – 11: I guess here you should give one or two references for the statistical models, too. Eg. Some approaches relating winter snowpack to summer-flows (e.g.: Godsey et al., 2014; Jenicek et al., 2016).

5 – 4: Please support the "one-year-leave-out cross-validation method" with a reference.

6 -2: Please support "Precipitation and streamflow forecasts are evaluated with deterministic and probabilistic scores commonly used in ensemble forecasting" with a reference, e.g. Brown et al EVS paper.

8-15: Nice idea to use the ensemble of past-streamflow observations as a reference. If you would "sort-out" some past years by means on analogues techniques you might get a very challenging set of members for your ensemble forecast. Have you tried this?

8-22: Another interesting feature here. This definition of gain is very elucidative. Can you maybe elaborate on pro and contra of this kind of "gain" definition with respect to scores based on cost-loss considerations?. Why choosing such a large gap of day between the classes? Have you tried to make a 30-day moving window? Or a 15-days moving window?

9 – 2 & 9 -19: Both in Figure 2 & 4 CRPSS is showing increasing skill at weeks 5 and 9. We are also used to "struggle" in interpreting such cycles. Do you have some ideas on your particular case here?

11 – Figure 4: How would look like this figure if you use the "ensemble based on past streamflow" as a reference?

13 – Figure 6: Right margin is cropped. Additionally, the "too wet"=red is not really intuitive.

13 – 2: "the 2-month" or the "month-2" ? If you mean the one for the second month of the forecast I would find more adequate to use "month-2".

17 – Figure 8 (and later 9): I like such Figures because the train my brain. Tell me if I am reading it wrong:

> If a look at a certain score in a certain season than for a particular bias correction method a percentage of the basins is showing improvement in lead time. Of this percentage a distinct portion shows improvement of let's say 60 to 90 days.

> So largest improvement is in the PIT-Skill in summer and Winter for the EDMD methods.

Right?

22 – 8: This would be the only heading with a question mark. Maybe replace this with a sentence

23 – Figure 15: is there any special reason (beside readability) for having different scales in the three panels?

Final considerations:

I find this manuscript is a very solid communication for the growing community dealing with seasonal forecasting in hydrology. It uses a strong set of data and robust statistics and comes to valuable conclusions. I indicated some weakness that let me recommend to the editors to ask for moderate revisions for this manuscript.

Best regards

Massimiliano Zappa

Birmensdorf, 23. March 2016

References:

Brown, James D., et al. "The Ensemble Verification System (EVS): A software tool for verifying ensemble forecasts of hydrometeorological and hydrologic variables at discrete locations." Environmental Modelling & Software 25.7 (2010): 854-872.

Godsey, S. E., Kirchner, J. W., and Tague, C. L.: Effects of changes in winter snowpacks on summer low flows: case studies in the Sierra Nevada, California, USA, Hydrol. Process., 28, 5048–5064, doi:10.1002/hyp.9943, 2014.

Jenicek, M., Seibert, J., Zappa, M., Staudinger, M., and Jonas, T.: Importance of maximum snow accumulation for summer low flows in humid catchments, Hydrol. Earth Syst. Sci., 20, 859-874, doi:10.5194/hess-20-859-2016, 2016.

Joerg-Hess S, Kempf SB, Fundel F, Zappa M. 2015. The benefit of climatological and calibrated reforecast data for simulating hydrological droughts in Switzerland. Met. Apps. 22: 444–458. doi:10.1002/met.1474

---

## Referee Comment (RC3) · Anonymous Referee #2 · 26 Mar 2016

**Summary**

The study analyses the skill of ECMWF's System 4 seasonal forecasting system for precipitation and streamflow forecasts in 16 French catchments, using the hydrological model GR6J for transferring the meteorological forecasts into streamflow forecasts. In particular, the study focusses on the effect of bias-correcting precipitation in different ways. Main conclusions are that linear scaling and EDMD bias-correction with monthly calibration windows perform better than other methods. In general, bias-correction was found to improve the skills, but the result varies for the different skill scores. Often, a trade-off between decreasing sharpeness and increasing reliability was found when

applying bias-correction. When comparing the differences of the bias-correction effect on precipitation and streamflow, it was found that in cases when sharpness and overall performance increased by bias-correction, it increased more strongly for streamflow than for precipitation. The opposite was found to be true for reliability.

**General comments**

The study addresses a relevant scientific question and the methods used are sound. A few results need some further clarification and/or discussion and I have listed the open issues in the detailed comments. Additionally to the detailed comments, I would like to add two general comments.

- The study uses mainly modelled streamflow as a reference. Nevertheless, I miss some indication of the hydrological model performance in the 16 basins. This is particularly relevant since also the observed streamflow is used as a reference forecast in one part of the manuscript, and this analysis would critically depend on systematic biases of the hydrological model.

- The manuscript covers a large body of results and is therefore lengthy. I think that it could be streamlined without losing too much information.

Over all, I suggest acceptance of the manuscript after my comments have been taken into account. I'm looking forward to the revised manuscript.

**Detailed comments**

**Page 3, line 1:** Some reference needed to support the statement that linear scaling and distribution mapping are widely used methods in seasonal forecasting.

**Page 4, line 13:** Which parametrization was used to derive potential evapotranspiration?

**Page 4, line 23:** What is meant by interannual potential evapotranspiration? I would have understood the manuscript in such a way that potential evapotranspiration is derived from raw, i.e. non-bias-corrected, forecasts, but in this case, the term interannual potential evapotranspiration does not make sense. I probably misunderstood something and would like that the authors clarify the manuscript in that respective.

**Page 5, lines 3-4:** Just a comment, nothing to change: leave-one-year-out might result in the validation years not being really independent, as interannual serial correlation might be quite high. Maybe it would be interesting to test larger block sizes in future studies.

**Page 6, lines 21-25:** In the case of EDM-m and GDM-m, only 29 data points are used to derive a cumulative distribution function for the reference data. This is a rather low number of data points, potentially leading to estimated cumulative distributions that are non-robust. Maybe, and this is of course rather speculative without analyzing the data, this could be a reason for the worse bias validation of EDM-m and GDM-m in Fig. 6.

**Page 6, lines 21-25:** I'm not aware of a study that applied gamma distribution fitting for monthly precipitation data. Could you please cite a study to support the method GDM-m? I'm a bit worried that the gamma distribution might not be a good choice for monthly mean precipitation values.

**Page 6, lines 25:** It is unclearly written how exactly the EDM and GDM correction is applied to daily values. I assume it is done as such that the monthly values are corrected following the quantile mapping procedure. After that, a correction factor is estimated between the corrected and the uncorrected monthly mean value and this correction factor is applied to all daily values. The text on line 25 is though misleading as the actual correction in a quantile-mapping framework is the mapping of the uncorrected values to the cumulative probability space, from which a corrected value is derived following an inverse mapping based on the reference data. As the mapping is calibrated for monthly values, it cannot be used for daily values directly. Please clarify the text.

**Page 8, lines 23-24:** The first sentence in this paragraph is redundant. Consider removing it.

**Page 8, lines 27-28:** According to section 3.3, all data was first converted to weekly means, thus a seven-day moving average cannot be derived. Please clarify the contradictions.

**Page 9, line 12:** Why is the value +0.1 and -0.1 for the deviations from the diagonal chosen?

**Page 9, line 19:** Unclear use of the word "translate".

**Sections 4-6:** The presentation of the results could be improved and shortened. When I read the manuscript, I would have liked to have the comparison of the raw and bias-corrected forecasts closer together and I suggest combining the discussion of the raw and the EMDD corrected forecasts. It would be much easier for the reader to follow the discussion if, for e.g., figure 2 and 10 are to be combined into one figure. Similarly for all other seasonal skill score figures in sections 4 and 6.

**Page 11, line 8-9:** I thought that the reference forecast is the streamflow simulated using the reference precipitation. Thus, any model deficiencies regarding low flows should not affect the skill score as also the reference forecast would suffer from those deficiencies. Also, similarly as for the low flows, the PIT diagram reports difficulties to forecast the high flow. What could be the reason for this issue? The explanations give in the manuscript so far are not fully convincing.

**Page 14, lines 19-23:** The reasoning is unclear to me, probably due to an unclear explanation how the bias-correction works. If it is done in the way I described in the comment regarding EDM and GDM, I don't think that the reasoning is correct. Everything stated for the monthly correction would also apply for the daily correction. Also on the daily time scale, the rank structure (see comment below) of the forecast is not the same as for the reference data. In both cases (monthly and daily correction), the distribution mapping should be able to correct differing rank structures and remove biases in the monthly mean effectively. In fact, I would have expected the daily correction to perform worse than monthly correction when evaluated on the monthly scale since it is not

targeted to the monthly scale but the daily scale. I rather think it has to do with a higher sensitivity of monthly corrections to overfitting as evaluated within the cross-validation framework. Admittedly, distribution mapping can lead to unforeseen effects and it might very well be that I'm wrong. If the authors are convinced that their reasoning is correct, I would like them to describe in the reply a case where the distribution mapping fails in more detail, for e.g. by showing how the reference and forecast distribution look like and how the mapping fails to come up with a correct monthly mean value.

**Page 14, lines 19 and 21:** Usage of the term "time structure" seems to be misleading. I understand this term in a way that it refers to the temporal sequence of values, i.e. that the day n in the reference corresponds to day n in the forecasts. However, distribution mapping does not have this requirement. It is rather the rank structure as I would call it: Rank n in the reference has to correspond to the rank n in the forecasts. Please correct the terminology or explain in more detail what "time structure" means.

**Section 5.2:** In my opinion, this section does not give new information which is not already present in figure 6 (time varying bias-correction factors can be inferred from the panel "Before bias correction") and I suggest removing it for the sake of shortening the result section. The only new aspect is that the correction factors for EDMD vary more than for LS, but this comparison is not valid in my point of view as one should not compare a mean correction factor with a correction factor for a quantile level. I'm pretty sure that if you would calculate the correction factor for the mean in the case of EDMD, it would be very similar to the LS factor.

**Section 5.3 and figures 8 and 9:** I very much like this analysis. I'm not sure though if I really understand the analysis completely. MAE is partly related to the bias analysis in figure 6, i.e. if biases in figure 6 are substantial, then MAE should be even larger since MAE does not allow for a compensation of errors. EDM-y and GDM-y have large biases throughout the year in figure 6, and in some cases and particularly in summer, the bias is even larger than in the uncorrected data. However, in figure 8 the two methods stick out for MAE and IQR in summer lead to skill improvements in all catchments up to a lead time >60 days. To me, this seems to be contradicting. Could you please

explain this particularity?

**Page 19, line 1:** If I read the figure 10 correctly, there are negative skill score values and therefore, the statement that the skill scores are always larger than zero does not hold.

**Page 20, line 3:** If I read the figure 12 correctly, there are negative skill score values and therefore, the forecast performs sometimes worse than ESP, which is the opposite of what is stated on this line.

**Page 20, lines 12-13:** It is not clear to me why this is expected. I would expect that comparison to streamflow climatology is a harder check and therefore the skillfull lead-time should be smaller than in the comparison to the baseline reference run since also the hydrological model bias deteriorates the skill. I surely misunderstand something but I think it would be good to add a bit more explanation in the manuscript.

**Page 22, line 4:** As for the uncorrected forecast discussion, I do not understand why it is the hydrological model that causes the problems with low-flow overestimation. The reference data is also output of the same hydrological model driven by the reference precipitation data. I would therefore rather think that it is some characteristics in the input data which the bias-correction cannot correct for that causes the problem (for e.g. dry-spell lengths). If the authors still think their statement holds, I would like to have a bit more explanations why this can be the case.

**Page 27, lines 13-15:** References needed

**Section 6.3:** Just a comment: I very much like this analysis.

**Section 6.4:** Although I like the illustrative character of this section, it stands a bit loose within the rest of the manuscript. I suggest to either motivate the section better or, for the sake of brevity, to remove it. In my opinion, the main statements of this sections have already been made, i.e. increased sharpeness after bias-correction compared to ESP.

**Figures**

**Figure 2:** "... and all seasons." The figure only shows two seasons, please correct the caption.

**Figures 3, 5, 11, 14:** The dashed lines should be explained in the figure as well, and not just in the text describing figure 3.

**Figure, 6:** Although certainly correct, I do not see a reason why to transform the simple relative bias into 1-bias. I understand that this transformation turns the bias into a skill score. However, in my opinion, the interpretation is not following the one for skill scores anyway. The perfect bias-correction would not yield 1 but 0. I suggest plotting the relative bias without transformation. The scale would be much easier interpretable as it directly refers to a percentage over- or underestimation.

**Figures 8 and 9:** Why are there different color scales for the different seasons?

**Figure 15:** What are the colours standing for? There is probably also an error in the caption where it reads "shown for all seasons".

**Technical comments**

**Page 19, line 10:** precipitation instead of precipitations

---

## Referee Comment (RC4) · Anonymous Referee #3 · 1 Apr 2016

This paper deals with an interesting topic, the effect of bias corrections on the quality of seasonal streamflow forecasts. It is mostly well written and the overall structure of the paper is good. However, I found the following issues that need revision before I could recommend the paper to be published.

Main points:

1) My most important point is that the paper is too long. I suggest to set a hard (!) reduction requirement of at least 25% (number of words). It is up to authors to decide which parts they remove or shorten. Just a few suggestions from my side: discuss fewer bias correction methods, remove almost completely page 13 line 3 – page 14 line 7, remove third and fourth sentence of section 3.2.1.

[Figure]

2) In the paper sharpness is discussed with the assumption that quality increases with sharpness. Mason and Stephenson (2008) write that "in the extreme case of no predictability, the forecast probability should always be equal to the climatological probability". So, forecasts can be too sharp, which should be a conclusion from e.g. Figure 2, where sharpness for longer lead times is larger than that of the reference. So, the sharpness results and conclusions should be reconsidered.

3) A better (and longer) introduction to PIT diagrams is needed. Since these diagrams are not well explained in the paper, I was not able to understand the PIT results. I suggest at least to write much more clearly how these diagrams are constructed, to show a figure like Figure 2 from Laio and Tamea, to clarify what PIT values (vertical axis of figures in paper) are and to add a text to the horizontal axis of the PIT diagrams displayed in the paper. How does the area in the diagrams measure reliability? Is the area also sensitive to bias? Is that acceptable? In Section 3.3.1. the text mentions "concentration of points" but only lines are shown in the diagrams. So, what do you mean by "concentration of points"?

4) PIT area, MAE and CRPS are all sensitive to bias, as far as I can see. This should be mentioned in Sections 3 and 7 and discussed in Section 7.

5) Section 2.2 mentions that observations are used to initialize streamflow. What about the initialization of snow and soil moisture? These form important contributions to predictability.

6) Sections 3.2.1. and 3.2.2. about the bias correction methods need references. EDM and GDM seem to be have strange effects: a specific amount of daily precipitation is corrected differently for different years, depending on the monthly amount of precipitation. What is the motivation to possibly employ these two methods? Perhaps some of the investigated methods should not be considered at all, see point 1 about shortening the paper. I found LS-m and EDMD-m the most interesting methods.

Minor points:

page 1, line 16: "contributes" instead of "contribute".

page 2, line 7: "widespread use of" instead of "the widespread of"

page 2, line 21: remove "rather than by initial conditions"

page 3, line 13: "varied between" instead of "derived from"

The hydrological model also needs temperature as input to compute potential evapo-transpiration. Write clearly how this input is constructed.

page 3, line 18: add "heavily" before "influenced"

page 3, line 23: replace "interannual" by "long-term mean". Over which years? On a monthly basis? Also for hindcasts?

page 3, section 2.2: motivate why the focus is solely on the influence of precipitation input .

page 6, section 3.3: So, do the evaluations for lead week 1 for the winter include all the hindcasts made on December 1, January 1 and February 1? These are then 15 members issued in December and January and 52 members issued in February. How do you deal with this inequality? And do the evaluations for lead week 6 for the winter include all the hindcasts made on November 1, December 1 and January 1? Explain this clearly.

page 7, line 8: "coinciding with" instead of "superposed with"

page 7, line 24: "Ranked" instead of "Rank"

page 8, line 6: What is the observation period?

page 8, line 14: From which period are the observations?

page 8, line 23: "caused" instead of "brought"

page 8, line 28: "becomes negative". What is done if there is more than one transition

from a positive to a negative score?

page 9, line 28: "this is observed in the majority of catchments". This does not seem to be the case. There is roughly an equal number of curves below and above zero.

page 13, figure 6: I would expect no bias at all in the lower right and left panel. What is the cause of these biases? Are the remaining biases caused by the one-year-leave-out method? If so, I would expect them to vary randomly around zero.

page 13, line 13: "in the easternmost part" instead of "at the most eastern part"

page 14, line 30: add "cumulative" before "probability"

page 17, figure 8: "Fraction of catchments" instead of "Number of catchments"

page 18, last line: As far as I can see the CRPS in not lower after bias correction.

page 19, line 3: replace "in regards to" by "with respect to"

I recommend to combine figure 2 with figure 10 into one figure, and figures 3 with figure 11 into one figure, etc. The reader now has to turn over pages to compare the figures.

Figure 15: how are seasons represented?
* * *

---

## Editor Comment (EC1) · I.G. Pechlivanidis (Editor) · 6 Apr 2016

Dear authors,

Your manuscript was reviewed by three experts on seasonal hydro-meteorological forecasting. I found their comments crucial and addressing them would improve the quality of your article. I encourage you to submit your response to allow an open discussion and also a preliminary evaluation of the potentially revised manuscript, which will eventually result in a publication in this special issue.

I would kindly highlight (as two referees also pointed) the need to reduce the length of this manuscript. Please focus on highlighting the key messages from your analysis.

I am looking forward to your reply.

Best regards, Ilias Pechlivanidis

---

## Author Comment (AC1) · 18 Apr 2016

**Response to Reviewer#1**

The authors want to thank Dr. Zappa for his valuable review which will help to enhance the document significantly. We provide below our answers.

**Reviewer 1**

The authors are presenting a generally well-conceived study focussed on possible improvement of seasonal streamflow forecasts by applying bias correction to the forcing precipitation forecasts. I found the manuscript well written, with appealing and adequate artwork and well discussed results. I found that the whole manuscript well supports the results the authors are enumerating. The scores the chose are adequate and the combination of scores allows drawing conclusions that are not blended by the scores that show maximal improvement with respect to the defined references. It is not clear which PET forcing is used in forecasts mode. If the authors use the observation-based reanalysis of PET in combination with the precipitation forecasts, then the appeal of this study would be quite reduced, in my "operational-minded" opinion.
Furthermore, the discussion and conclusions section is not adequately considering previous studies.

Authors' reply (AR): We thank the reviewer for his comments and we provide a detailed reply to the use of PET and to the final session of the manuscript below.
* * *
**Reviewer's comment (RC):** While the Introduction is well balanced and gives useful insight on previous work on the topic and also references supporting the envisaged methodology, I found that the final paragraphs should possibly include more information on the novelty of the present manuscript. Also in the methodological section some more referencing is needed. See minor comments for this.

Authors' reply (AR): Thank you for pointing this out. We propose to change the following in the Introduction to better emphasize the novelty of our study (end of line 29, beginning of line 30, page 3): "Despite these recent works and to the knowledge of the authors, no previous study has compared bias correction methods and their impact on streamflow forecasting in a systematic way, with a focus on understanding how the main attributes of forecast performance are impacted by bias correction.
This paper aims to provide insights into the way bias correcting seasonal precipitation forecasts can contribute to the skill of seasonal streamflow predictions, notably in terms of overall performance, reliability, sharpness and skilful lead time. It investigates the potential of bias corrected ECMWF System 4 forecasts to improve streamflow forecasts at extended lead times over 16 catchments in France. An in-depth comparison of eight variants of linear scaling and distribution mapping methods applied over the 1981-2010 period is presented. Section 2 presents…"
* * *
**RC:** 4-4-15: We learn here about the meteorological forcing. It is clear to me how you use precipitation, but as a forecast and as SAFRAN product. Concerning Potential Evapotranspiration (PET), only SAFRAN is declared. I'd like you to declare which PET is used in retrospective forecasts forced by the ECMWF products. If it is from ECMWF, you should state why you are not post-processing it. If you use SAFRAN, you should be able to assess how much uncertainty are you neglecting by using the best observed estimates of PET instead of using a forecasted value (which you need to do as soon as you will deploy the system in real-time). In our experience, for basins not affected by snow-melt, the post-processing of relative-humidity data (an important proxy the evaporation demand by the atmosphere) helps improving the estimation of hydrological droughts (Jörg-Hess et al., 2015).

AR: The potential evapotranspiration (PET) used to force the hydrological model is, in fact, the mean interannual PET. For a given day of the year, the estimated PET on this day is assumed to be the mean of all PET computed for this day of the year, in all available years.

Here, the mean interannual PET is the average of the PET calculated for each year from 1958 to 2010. PET for each year is calculated using SAFRAN. Regardless of the precipitation scenario fed to the model (historical precipitations or System 4), the PET scenario used as input to the model is always the same: the series of mean interannual PET corresponding to the forecast period. With this setup, we can focus on the changes in skill that can solely be attributed to the bias correction of precipitations, which is in the aim of our study. Adding the uncertainty of temperature forecasts in the analysis would in fact require a different framework. For instance, we would need to set up multi-variable bias corrections to take into account the dependencies between precipitation and temperature, or we would need to consider the impact of observed trends in time series of observed temperatures in some regions in France prior to post-processing and ESP forecasting. This is beyond the scope of this study, although interesting for further investigations and specific operational setups. We will clarify the way PET is considered and our reasons for doing so in the revised version.

RC: 4-25: I just reviewed another paper on seasonal forecasting where authors did not show any score concerning their calibration/validation and I amended it. Same here. I am happy with a table as supplementary material.

AR: The following table summarizes some scores on the calibration and the validation of the GR6J model. Since other reviewers and the editor recommended decreasing the length of the paper, we propose to summarize this information in a sentence on line 25 in the revised version: "…applied to root-squared flows. We obtained an average KGE of 0.95 in calibration and 0.94 in validation over the sixteen catchments. The bias obtained in simulation ranges from -0.02 to 0.05."

| Catchment | Calibration KGERQ | Validation KGERQ | Validation C2MQ | Validation 1-Bias |
|---|---|---|---|---|
| 1 | 0.93 | 0.92 | 0.75 | 0.01 |
| 2 | 0.93 | 0.92 | 0.65 | 0.03 |
| 3 | 0.94 | 0.94 | 0.64 | 0.05 |
| 4 | 0.94 | 0.94 | 0.72 | 0.02 |
| 5 | 0.94 | 0.94 | 0.69 | 0.00 |
| 6 | 0.95 | 0.95 | 0.77 | -0.02 |
| 7 | 0.95 | 0.95 | 0.79 | 0.03 |
| 8 | 0.97 | 0.97 | 0.87 | 0.02 |
| 9 | 0.97 | 0.97 | 0.84 | -0.01 |
| 10 | 0.89 | 0.88 | 0.58 | 0.00 |
| 11 | 0.95 | 0.95 | 0.81 | 0.04 |
| 12 | 0.95 | 0.95 | 0.82 | 0.04 |
| 13 | 0.93 | 0.93 | 0.86 | 0.05 |
| 14 | 0.96 | 0.96 | 0.88 | 0.03 |
| 15 | 0.97 | 0.97 | 0.84 | 0.02 |
| 16 | 0.95 | 0.94 | 0.81 | 0.04 |

RC: 24-18: I like this evaluation very much, just, I miss some quantification supporting the description based on visual inspection you are giving. Be pragmatic.

AR: Thank you very much for this suggestion. We propose to include in the plots of Figure 16 a quantification of the performance of the systems over the period presented (April 2004 to April 2007). Notably, the coverage probability provides a good quantification to support the description. Here below, we summarize the values of MAE, coverage probability 90% (COV 5-95) and 50% (COV 25-75) obtained by each forecasting system over the displayed period. From these values, we observe that the ensembles based on past precipitations and past streamflow (HistQ and ESP) are more accurate over the chosen period (lower MAE values).

We also observe that the coverage probability of EDMD-m is the best over the chosen period. We propose to add these values to the plots of Figure 16 and to include a sentence on it in the interpretation of the hydrographs.

|  | HistQ | ESP | LS-m | EDMD-m |
|---|---|---|---|---|
| MAE ($m^3$/s) | 3.81 | 4.06 | 4.26 | 4.26 |
| COV 90 % (5-95) | 97 % | 92 % | 85 % | 89 % |
| COV 50 % (25-75) | 66 % | 60 % | 46 % | 51 % |

**RC:** 25-3: The discussion section is here quickly merged with the conclusions. The only link to current literature one is expecting here merely consists in a enumeration of possible post-processing of the forecasts with currently available methods. Here some more effort has to be shown to make also this section a valuable part of the manuscript.

AR: We will revise this section. For instance, we think we can discuss our results in the light of those of Gudmundsson et al. (2012) and Teutschbein and Seibert (2012). However, we will not be able to add too much text, since the length of the paper was an issue for the other two reviewers and the editor.

**RC**: 26-2: You address here the issue of implementation in operational systems. Again, declare how you deal the PET, and then re-evaluate the potential for real-time operations.

AR: See previous reply for PET. We will consider deleting this sentence or clarifying the limitations of the framework we adopted for real-time operations.

**RC:** 2-11: I guess here you should give one or two references for the statistical models, too. Eg. Some approaches relating winter snowpack to summer-flows (e.g.: Godsey et al., 2014; Jenicek et al., 2016).

AR: Thank you, we will consider adding these two references in the revised version.

**RC:** 5-4: Please support the "one-year-leave-out cross-validation method" with a reference.

AR: We propose to add the following reference:
Arlot, Sylvain; Celisse, Alain. A survey of cross-validation procedures for model selection. Statist. Surv. 4 (2010), 40--79. doi:10.1214/09-SS054. http://projecteuclid.org/euclid.ssu/1268143839.

**RC:** 6 -2: Please support "Precipitation and streamflow forecasts are evaluated with deterministic and probabilistic scores commonly used in ensemble forecasting" with a reference, e.g. Brown et al EVS paper.

AR: We will add these references:

Brown, J.D., Demargne, J., Seo, D.-J., Liu, Y., 2010. The Ensemble Verification System (EVS): A software tool for verifying ensemble forecasts of hydrometeorological and hydrologic variables at discrete locations. Environmental Modelling & Software 25, 854 – 872. doi:http://dx.doi.org/10.1016/j.envsoft.2010.01.009

Casati, B., Wilson, L. J., Stephenson, D. B., Nurmi, P., Ghelli, A., Pocernich, M., Damrath, U., Ebert, E. E., Brown, B. G. and Mason, S. (2008), Forecast verification: current status and future directions. Met. Apps, 15: 3–18. doi: 10.1002/met.52

Jolliffe, I.T., Stephenson, D.B., 2003. Forecast Verification: A Practicioner's Guide in Atmospheric Science. John Wiley.
* * *
**RC:** 8-15: Nice idea to use the ensemble of past-streamflow observations as a reference. If you would "sort-out" some past years by means on analogues techniques you might get a very challenging set of members for your ensemble forecast. Have you tried this?

AR: This is precisely the topic of a second paper that is in preparation, and that should be submitted to this special issue (adding it here would result in a very long and unreadable paper). Past years were selected based on precipitation indices derived from seasonal forecasts. The resulting ensemble based on past-streamflow observations was compared with the ensemble of all past streamflow observations, the ESP and the streamflow forecasts obtained from precipitation forecasts and EDMD-m. We do not want to spoil the conclusions of this second paper here, so we invite the reviewer to check this special issue for our next submission.
* * *
**RC:** 8-22: Another interesting feature here. This definition of gain is very elucidative. Can you maybe elaborate on pro and contra of this kind of "gain" definition with respect to scores based on cost-loss considerations?. Why choosing such a large gap of day between the classes? Have you tried to make a 30-day moving window? Or a 15-days moving window?

AR: Thank you for the comment. We chose to evaluate the gain in terms of anticipation in response time, rather than in terms of relative economic value (REV), for instance, since cost-loss considerations would need an evaluation of (or additional assumptions on) mitigation costs, avoidable losses, as well as unavoidable losses for each studied catchment. Here, we may assume that increasing the anticipation response time could increase time for preparedness, which would decrease costs and losses related to missed events or actions taken with no or little anticipation to a critical situation. The cost-loss approach would need to be applied considering this evolution of forecasting with time since in a seasonal forecasting system one has several forecasts or months ahead to detect a potential critical situation and act accordingly. Actions and consequences would need to be stratified according to the time available for action in order to have this aspect reflected in an evaluation score.

The gap was chosen to help represent the improvements due to bias correction at a monthly time scale of reference. It seemed to us that a month ahead could be a good minimum of time necessary to adapt any mitigation actions once a critical situation is forecasted by a seasonal forecasting system. As shown in Figures 8 and 9, this choice seems to be appropriate to a joint representation in a plot, while differentiating situations for a useful analysis.

We did not try to use windows larger than 7 days. The objective of the rolling mean was to smooth the skill curves and remove the high frequency variations of the skill at the daily time step. Seven days appeared to be enough to smooth the curves, while keeping the moving mean as a good estimate of the gain in lead time.
* * *
**RC:** 9 – 2 & 9 -19: Both in Figure 2 & 4 CRPSS is showing increasing skill at weeks 5 and 9. We are also used to "struggle" in interpreting such cycles. Do you have some ideas on your particular case here?

AR: We have also spent some time trying to interpret these cycles. Despite a closer look at the data and the scores, under different angles, we could not see any systematic reasons for these cycles. We think it may be related to several correlated aspects, such as the type of forecasting model/system, the forcings, the behaviour of the catchment, etc. This remark is interesting and we would be glad to have more insights from other researchers on this too.

**RC:** 11 – Figure 4: How would look like this figure if you use the "ensemble based on past streamflow" as a reference?

AR: The following figure represents the IQRSS and the CRPSS of the streamflow forecasts without bias correction, when the ensemble based on past streamflow is used as reference. It can be compared to Figure 4 to see that the skill is higher with this reference, and to Figure 13 to see that bias correction has also increased the skill of forecasts with regard to past streamflow.

[Figure]

**RC:** 13 – Figure 6: Right margin is cropped. Additionally, the "too wet"=red is not really intuitive.

AR: The reviewer is right. Exchanging the blue and red colours in the scale, and increasing the right margin lead to the following figure. We propose to replace the original figure with this one:

[Figure]

**RC:** 13 – 2: "the 2-month" or the "month-2"? If you mean the one for the second month of the forecast I would find more adequate to use "month-2".

AR: We agree with the reviewer, the occurrences in 12 – 9, 13 – 2, 14 – 10, 14 – 27, 14 – 28 of "the 2-month" will be changed to "month-2" in the revised version.
* * *
**RC:** 17 – Figure 8 (and later 9): I like such Figures because the train my brain. Tell me if I am reading it wrong:
If a look at a certain score in a certain season than for a particular bias correction method a percentage of the basins is showing improvement in lead time. Of this percentage a distinct portion shows improvement of let's say 60 to 90 days.
So largest improvement is in the PIT-Skill in summer and Winter for the EDMD methods. Right?

AR: Thank you. Your reading of the figure is absolutely correct. We will revise our description of the results to make sure it is clear to the readers.
* * *
**RC:** 22 – 8: This would be the only heading with a question mark. Maybe replace this with a sentence

AR: The question mark was a typo and we will remove it.
* * *
**RC:** 23 – Figure 15: is there any special reason (beside readability) for having different scales in the three panels?

AR: No, there is no special reason apart to zoom in on the case of the CRPSS. Keeping the same scales gives the figure shown here below. With the same scales, we do not see clearly the impacts on CRPS, but we can better see the relative improvements between the different forecast attributes. For instance, we can see that the impact of bias correction is more seen in sharpness and reliability, rather than in overall performance. We will consider which could be the best figure to show when preparing the revised version of the paper.

[Figure]

[Figure]

Final considerations:
I find this manuscript is a very solid communication for the growing community dealing with seasonal forecasting in hydrology. It uses a strong set of data and robust statistics and comes to valuable conclusions. I indicated some weakness that let me recommend to the editors to ask for moderate revisions for this manuscript.
Best regards
Massimiliano Zappa
Birmensdorf, 23. March 2016

[Figure]

AR: Thank you again for your comments that greatly contribute to improving our paper.

---

## Author Comment (AC2) · 9 May 2016

**Response to Reviewer#3**

The authors want to thank Reviewer#3 for the valuable comments, which will help us to enhance our paper. We provide below our answers to the comments.

**Reviewer 3**

This paper deals with an interesting topic, the effect of bias corrections on the quality of seasonal streamflow forecasts. It is mostly well written and the overall structure of the paper is good. However, I found the following issues that need revision before I could recommend the paper to be published.
* * *
**Reviewer's comment (RC):**
Main points:
1) My most important point is that the paper is too long. I suggest to set a hard (!) reduction requirement of at least 25% (number of words). It is up to authors to decide which parts they remove or shorten. Just a few suggestions from my side: discuss fewer bias correction methods, remove almost completely page 13 line 3 – page 14 line 7, remove third and fourth sentence of section 3.2.1.

Authors' reply (AR): Reviewer 2 and the editor also recommended shortening the paper. This issue will be addressed when producing the revised version.
* * *
**RC:** 2) In the paper sharpness is discussed with the assumption that quality increases with sharpness. Mason and Stephenson (2008) write that "in the extreme case of no predictability, the forecast probability should always be equal to the climatological probability". So, forecasts can be too sharp, which should be a conclusion from e.g. Figure 2, where sharpness for longer lead times is larger than that of the reference. So, the sharpness results and conclusions should be reconsidered.

AR: We agree with the reviewer that sharpness in itself, as any other forecast quality attribute, is not necessarily an indicator of a perfect forecast. In our study we adopted the paradigm of Gneiting et al. (2007): « maximizing the sharpness of the predictive distributions subject to calibration ». This means that for two systems with equal levels of reliability, the best one is the sharper one (i.e., lower IQR score in our study). The evaluation of sharpness is thus complementary to the evaluation of reliability. That is the reason why we adopted the scores PIT diagram and IQR in our study. We will make sure that this is clear throughout the text in the revised version.
* * *
**RC:** 3) A better (and longer) introduction to PIT diagrams is needed. Since these diagrams are not well explained in the paper, I was not able to understand the PIT results. I suggest at least to write much more clearly how these diagrams are constructed, to show a figure like Figure 2 from Laio and Tamea, to clarify what PIT values (vertical axis of figures in paper) are and to add a text to the horizontal axis of the PIT diagrams displayed in the paper. How does the area in the diagrams measure reliability? Is the area also sensitive to bias? Is that acceptable? In Section 3.3.1. the text mentions "concentration of points" but only lines are shown in the diagrams. So, what do you mean by "concentration of points"?

AR: The probability integral transform (PIT) histogram is used in forecast verification to evaluate if the empirical time series of PIT values (the PIT value is the value that the predictive cumulative distribution function associates with the observation at a given time step) has a uniform distribution (see also, Gneiting et al., 2005 [1], where it is also explained that "uniformity is usually evaluated in an exploratory sense, and one way of doing this is by plotting the empirical cumulative distribution function of the PIT values"). This is what we have done in our paper (Page 7, lines 6-8). In order to compare systems, we also evaluated the score defined as the "PIT area", as proposed in the reference cited in the paper (Renard et al., 2010). The further the PIT curve is from the 1:1 diagonal, the less reliable the ensemble is. Therefore, the smaller the area between the curve and the 1:1 diagonal, the more reliable the ensemble is. The rank histogram or Talagrand diagram, proposed independently in

the literature, is a similar measure. Gneiting et al. (2005) indicate that "If we identify the predictive distribution with the empirical cumulative distribution function of the ensemble values, this technique is seen to be equivalent to plotting a PIT histogram". The visual inspection of the PIT diagram can be a useful assessment (on systematic biases or spread deficiencies, as we mention on page 7, lines 8-13), but forecast deficiencies may still be hidden behind the assessment (deficiencies in sharpness, for instance). That's why we use (and recommend) the joint evaluation of other scores. We hope this clarifies our approach.

The paragraph on page 7, lines 5-15, as well as the x and y axes of the PIT diagrams in Figures 3, 5, 11 and 14 will be revised to clarify the construction and representation of the PIT diagram. We have linked our points with lines for a better visualization of the results of the 16 catchments in a unique PIT diagram, so we will also clarify the term "concentration of points" in the text. We would like, however, to avoid adding a figure that is already presented in another easy-to-access paper that we are referencing (Laio and Tamea, 2007). This is especially important since we also need to reduce the length of the paper.

*[1] Probabilistic Forecasts, Calibration and Sharpness, Tilmann Gneiting Fadoua Balabdaoui and Adrian E. Raftery. Available here: https://www.stat.washington.edu/research/reports/2005/tr483.pdf*
* * *
**RC:** 4) PIT area, MAE and CRPS are all sensitive to bias, as far as I can see. This should be mentioned in Sections 3 and 7 and discussed in Section 7.

AR: These scores may indeed inform on biases, and we will make sure it is clearly mentioned in Sections 3 and 7.
* * *
**RC:** 5) Section 2.2 mentions that observations are used to initialize streamflow. What about the initialization of snow and soil moisture? These form important contributions to predictability.

AR: The GR6J model is a conceptual, reservoir-based hydrological model (Page 4, lines 20-22). Its inputs are daily precipitation and potential evapotranspiration. These data are used to run the model and initialize its states, including the state of its reservoirs, prior to the forecast date. The upper reservoir of the model can be assimilated (although it is not equivalent to, as it is not a physically-based model) to a "soil moisture accounting" reservoir. Therefore, in a sense, this is also initialized. As for snow modelling, it is not represented in the version of the model used in our study. The catchments chosen have little or no snow-influenced runoff (Page 4, lines 17-19). On lines 25-27, page 4, we mention the forecast updating of the model, which is a different procedure from the initialization. After initialization, the model goes through an "updating procedure", common in hydrological forecasting, which, in our case, is based on the last observed discharge. We will check section 2.2 to make the difference clearer in the revised version.
* * *
**RC:** 6) Sections 3.2.1. and 3.2.2. about the bias correction methods need references. EDM and GDM seem to be have strange effects: a specific amount of daily precipitation is corrected differently for different years, depending on the monthly amount of precipitation. What is the motivation to possibly employ these two methods? Perhaps some of the investigated methods should not be considered at all, see point 1 about shortening the paper. I found LS-m and EDMD-m the most interesting methods.

AR: Our motivation is to evaluate if EDM brings additional value regarding LS, notably in correcting bias for extreme precipitation, and whether the use of a fitted distribution (here, GDM) enhances performance or not. We also found LS-m and EDMD-m more interesting, but this comes from the progressive analysis of all the other methods too. We think it is important to show all the methods as they have different levels of complexity. When shortening the length of the paper, however, we will pay attention to check if we can cut some text from this part of the paper.
* * *
**RC:** Minor points:
page 1, line 16: "contributes" instead of "contribute".

page 2, line 7: "widespread use of" instead of "the widespread of"
page 2, line 21: remove "rather than by initial conditions"
page 3, line 13: "varied between" instead of "derived from"

AR: All minor points will be considered in the revised version.
* * *
**RC:** The hydrological model also needs temperature as input to compute potential evapotranspiration. Write clearly how this input is constructed.

AR: The calculation of the evapotranspiration was done following the Oudin formulation. This formulation can be found in Equation (3) of Oudin et al. (2005). It was computed based on the daily temperature from the SAFRAN reanalysis. The reference is cited in the paper. We will make sure that this is clear in the text.
* * *
**RC:** page 3, line 18: add "heavily" before "influenced"

AR: This will be corrected in the revised version.
* * *
**RC:** page 3, line 23: replace "interannual" by "long-term mean". Over which years? On a monthly basis? Also for hindcasts?

AR: For a given day of the year, the estimated PET on this day is the mean of all PET computed for this day of the year, over all available years (with exception for the targeted year). Reviewer 1 and 2 also pointed out that the PET used in the article should be better explained (please, refer to the answers to their reviews). This will be clarified in the revised version.
* * *
**RC:** page 3, section 2.2: motivate why the focus is solely on the influence of precipitation input.

AR: This is a choice we made as we were focusing on catchments with a pluvial-dominated hydrological regime.
* * *
**RC:** page 6, section 3.3: So, do the evaluations for lead week 1 for the winter include all the hindcasts made on December 1, January 1 and February 1? These are then 15 members issued in December and January and 52 members issued in February. How do you deal with this inequality? And do the evaluations for lead week 6 for the winter include all the hindcasts made on November 1, December 1 and January 1? Explain this clearly.

AR: The reviewer's understanding is correct on the way we aggregated the forecast values to compute the evaluation scores for the four seasons. We can thus have seasonal-based scores that involve forecasts with 15 or 51 members. This comes from the data setup of ECMWF. We only handled inequality when comparing ensemble of different sizes with the CRPS (as explained on page 8, line 16-21). Despite the inequality in the seasonal aggregation of scores, we note that this should not impact comparisons between seasons (since all seasons have a month with 51 members), and comparisons between raw and bias corrected forecasts (since aggregation is considered equally in both systems).
* * *
**RC:** page 7, line 8: "coinciding with" instead of "superposed with"
page 7, line 24: "Ranked" instead of "Rank"

AR: These will be corrected in the revised version.
* * *
**RC:** page 8, line 6: What is the observation period?
**RC:** page 8, line 14: From which period are the observations?

AR: Observed precipitation data were available for the period running from August 1958 to July 2010. Observed streamflow data were available for different time periods, ranging from 51 years to 35 years, according to the catchment. This will be clarified in the revised version.
* * *
**RC:** page 8, line 23: "caused" instead of "brought"

AR: This will be corrected in the revised version.
* * *
**RC:** page 8, line 28: "becomes negative". What is done if there is more than one transition from a positive to a negative score?
AR: If there are several transitions, the lead time of the first transition is considered. We will add "first" before "lead time beyond which" in line 27, page 8, to make this clearer.
* * *
**RC:** page 9, line 28: "this is observed in the majority of catchments". This does not seem to be the case. There is roughly an equal number of curves below and above zero.
AR: We will revise the sentence in the revised version.
* * *
**RC:** page 13, figure 6: I would expect no bias at all in the lower right and left panel. What is the cause of these biases? Are the remaining biases caused by the one-year-leave-out method? If so, I would expect them to vary randomly around zero.
AR: We also believe that they may be mainly due to the one-year-leave-out approach, especially when differences among the validation (target) year and the calibration period exist (e.g. for the wettest or driest years of the data period, which may not be of equal intensity). Depending on the "distance" between the target year and the calibration period this may cause a divergence from zero.
* * *
**RC:** page 13, line 13: "in the easternmost part" instead of "at the most eastern part"
page 14, line 30: add "cumulative" before "probability"
AR: These will be corrected in the revised version.
* * *
**RC:** page 17, figure 8: "Fraction of catchments" instead of "Number of catchments"
AR: This will be changed in Figure 8 and in Figure 9.
* * *
**RC:** page 18, last line: As far as I can see the CRPS in not lower after bias correction.
AR: We will review the sentence and separate IQR and CRPS analysis in the revised version.
* * *
**RC:** page 19, line 3: replace "in regards to" by "with respect to"
AR: This will be changed in the revised version.
* * *
**RC:** I recommend to combine figure 2 with figure 10 into one figure, and figures 3 with figure 11 into one figure, etc. The reader now has to turn over pages to compare the figures.
AR: We will consider if we can have the figures closer together in the revised version.
* * *
**RC:** Figure 15: how are seasons represented?
AR: Strong blue is used for winter, lighter blue for autumn, red for summer and lighter red for spring. We realized that the legend for the four seasons was missing in the figure and we will add it in the revised version.

---

## Author Comment (AC3) · 9 May 2016

**Response to Reviewer#2**

The authors thank Reviewer#2 for his valuable review, which help us to enhance our paper. We provide below our answers to the reviewer's comments.

**Reviewer 2**

**Summary**
The study analyses the skill of ECMWF's System 4 seasonal forecasting system for precipitation and streamflow forecasts in 16 French catchments, using the hydrological model GR6J for transferring the meteorological forecasts into streamflow forecasts. In particular, the study focusses on the effect of bias-correcting precipitation in different ways. Main conclusions are that linear scaling and EDMD bias-correction with monthly calibration windows perform better than other methods. In general, bias-correction was found to improve the skills, but the result varies for the different skill scores. Often, a trade-off between decreasing sharpeness and increasing reliability was found when applying bias-correction. When comparing the differences of the bias-correction effect on precipitation and streamflow, it was found that in cases when sharpeness and overall performance increased by bias-correction, it increased more strongly for streamflow than for precipitation. The opposite was found to be true for reliability.

**General comments**

The study addresses a relevant scientific question and the methods used are sound. A few results need some further clarification and/or discussion and I have listed the open issues in the detailed comments. Additionally to the detailed comments, I would like to add two general comments.
* * *
**Reviewer's comment (RC):**
• The study uses mainly modelled streamflow as a reference. Nevertheless, I miss some indication of the hydrological model performance in the 16 basins. This is particularly relevant since also the observed streamflow is used as a reference forecast in one part of the manuscript, and this analysis would critically depend on systematic biases of the hydrological model.

**Authors' reply (AR):** Reviewer 1 also recommended indicating the performance of the hydrological model. Please refer also to our answer to his review. We proposed to add the following sentence on line 25: "…applied to root-squared flows. We obtained an average KGE of 0.95 in calibration and 0.94 in validation over the sixteen catchments. The bias obtained in simulation ranges from -0.02 to 0.05." Both KGE values and bias values show good performance of the model. We provide the table of all values hereafter:

| Catchment | Calibration KGERQ | Validation KGERQ | Validation C2MQ | Validation 1-Bias |
|---|---|---|---|---|
| 1 | 0.93 | 0.92 | 0.75 | 0.01 |
| 2 | 0.93 | 0.92 | 0.65 | 0.03 |
| 3 | 0.94 | 0.94 | 0.64 | 0.05 |
| 4 | 0.94 | 0.94 | 0.72 | 0.02 |
| 5 | 0.94 | 0.94 | 0.69 | 0.00 |
| 6 | 0.95 | 0.95 | 0.77 | -0.02 |
| 7 | 0.95 | 0.95 | 0.79 | 0.03 |
| 8 | 0.97 | 0.97 | 0.87 | 0.02 |
| 9 | 0.97 | 0.97 | 0.84 | -0.01 |
| 10 | 0.89 | 0.88 | 0.58 | 0.00 |
| 11 | 0.95 | 0.95 | 0.81 | 0.04 |
| 12 | 0.95 | 0.95 | 0.82 | 0.04 |
| 13 | 0.93 | 0.93 | 0.86 | 0.05 |
| 14 | 0.96 | 0.96 | 0.88 | 0.03 |
| 15 | 0.97 | 0.97 | 0.84 | 0.02 |
| 16 | 0.95 | 0.94 | 0.81 | 0.04 |

**RC:** • The manuscript covers a large body of results and is therefore lengthy. I think that it could be streamlined without losing too much information.

AR: The third reviewer and the editor also recommended shortening the article. This will be our priority when producing the revised version.
* * *
**RC:** Over all, I suggest acceptance of the manuscript after my comments have been taken into account. I'm looking forward to the revised manuscript.

AR: We thank the reviewer for this positive appreciation of our paper.
* * *
**RC: Page 3, line 1:** Some reference needed to support the statement that linear scaling and distribution mapping are widely used methods in seasonal forecasting.

AR: We will explicitly add specific references showing the application of these bias corrections methods in seasonal forecasting.
* * *
**RC: Page 4, line 13:** Which parametrization was used to derive potential evapotranspiration?

AR: The calculation of the evapotranspiration was done prior to this study and is embedded in the database we use in the Catchment hydrology team at IRSTEA. It follows the Oudin formula, which can be found in Equation (2) of the reference Oudin et al. (2005). $K_1$ is set to 100 and $K_2$ to 5, as shown in Equation (3) of this same paper. This reference is cited in our paper.
* * *
**RC: Page 4, line 23:** What is meant by interannual potential evapotranspiration? I would have understood the manuscript in such a way that potential evapotranspiration is derived from raw, i.e. non-bias-corrected, forecasts, but in this case, the term interannual potential evapotranspiration does not make sense. I probably misunderstood something and would like that the authors clarify the manuscript in that respective.

AR: Reviewer 1 also requested that the PET used in the paper should be better explained. For a given day of the year, the estimated PET on this day is assumed to be the mean of all PET computed for this day of the year, in all available years. Here, the mean interannual PET is the average of the PET calculated from observed temperatures for each year from 1958 to 2010. This will be clarified in the revised version (please refer also to our answer to his review).
* * *
**RC: Page 5, lines 3-4:** Just a comment, nothing to change: leave-one-year-out might result in the validation years not being really independent, as interannual serial correlation might be quite high. Maybe it would be interesting to test larger block sizes in future studies.

AR: Definitely. This point was also recently raised in a HEPEX bog post by colleagues from CSIRO (http://hepex.irstea.fr/how-good-is-my-forecasting-method-some-thoughts-on-forecast-evaluation-using-cross-validation-based-on-australian-experiences/). We think that a more-than-one-year-leave-out procedure could potentially fit better for one of our catchments, which has a high base-flow index. We believe that its impact on the other catchments would be lower, given the length of our calibration periods. Also, the impact is expected to be lower when calibrating the hydrological model than when implementing the bias corrections. In any case, it would certainly be interesting to test it in a future study, where more catchments could also be included and focus could be put on this aspect.
* * *
**RC: Page 6, lines 21-25:** In the case of EDM-m and GDM-m, only 29 data points are used to derive a cumulative distribution function for the reference data. This is a rather low number of data points, potentially leading to estimated cumulative distributions that are non-robust. Maybe, and this is of course rather speculative without analyzing the data, this could be a reason for the worse bias validation of EDM-m and GDM-m in Fig. 6.

AR: This is an interesting point and could, as suggested, partly explain the poorer performance of EDM-m and GDM-m. Nevertheless, it is also worth noting that it is difficult to have much more years available for the calibration of these correction methods, since the meteorological reforecast archive

needs to be homogeneous (i.e., based on the same model) over the period. The fact that bias correction methods require long time series of forecasts is a well-known limitation in the field. The point raised by the reviewer is worth mentioning in the revised version and we will include a comment on it.
* * *
**RC: Page 6, lines 21-25:** `I'm not aware of a study that applied gamma distribution fitting for monthly precipitation data. Could you please cite a study to support the method GDM-m? I'm a bit worried that the gamma distribution might not be a good choice for monthly mean precipitation values.`

AR: The choice of a cumulative distribution function for precipitation (or streamflow) data is always a challenging one. The Gamma distribution is often assumed to be suitable and fitted to precipitation sums. Some examples of the gamma distribution fitted to monthly precipitations are:

*Zekai S. and A. G. Eljadid (1999) Rainfall distribution function for Libya and rainfall prediction, Hydrological Sciences Journal, 44:5, 665-680, DOI:10.1080/02626669909492266,*

*Husak, G. J., Michaelsen, J. and Funk, C. (2007), Use of the gamma distribution to represent monthly rainfall in Africa for drought monitoring applications. Int. J. Climatol., 27: 935–944. doi:10.1002/joc.1441*

The gamma distribution is also often used when computing the SPI. Examples are:

*Lavaysse, C., Vogt, J., and Pappenberger, F.: Early warning of drought in Europe using the monthly ensemble system from ECMWF, Hydrol. Earth Syst. Sci., 19, 3273-3286, doi:10.5194/hess-19-3273-2015, 2015.*

*X. Lana, A. Burgueño, M. D. Martínez and C. Serra: A review of statistical analyses on monthly and daily rainfall in Catalonia. 2009. Tethys (Journal of Weather & Climate of the Western Mediterranean), 6, 15–29, 2009, doi:10.3369/tethys.2009.6.02.*

In the preliminary steps of our study, we visually compared several distributions to fit to monthly precipitations in the selected catchments. The gamma distribution showed the best fit to the empirical distributions.
* * *
**RC: Page 6, lines 25:** `It is unclearly written how exactly the EDM and GDM correction is applied to daily values. I assume it is done as such that the monthly values are corrected following the quantile mapping procedure. After that, a correction factor is estimated between the corrected and the uncorrected monthly mean value and this correction factor is applied to all daily values. The text on line 25 is though misleading as the actual correction in a quantile-mapping framework is the mapping of the uncorrected values to the cumulative probability space, from which a corrected value is derived following an inverse mapping based on the reference data. As the mapping is calibrated for monthly values, it cannot be used for daily values directly. Please clarify the text.`

AR: The reviewer has understood correctly how the EDM and GDM methods are applied. To clarify the text, we propose to add on line 24, page 6: "In the case of EDM and GDM, the monthly values are first corrected based on the distribution mapping procedure. Then, for a given month, the ratio of the corrected monthly value and the non-corrected ones is used to correct all daily values within this month."
* * *
**RC: Page 8, lines 23-24:** `The first sentence in this paragraph is redundant. Consider removing it.`

AR: Following the reviewer's suggestion, we propose to merge the sentences on lines 23-24, page 8 into a single sentence: "To investigate the gain in performance brought by bias correction methods, we use the raw (uncorrected) forecasts as reference in the computation of the skill scores."
* * *
**RC: Page 8, lines 27-28:** `According to section 3.3, all data was first converted to weekly means, thus a seven-day moving average cannot be derived. Please clarify the contradictions.`

AR: When computing skill scores with reference to the ESP or historical streamflow, we computed scores based on weekly-averaged precipitation or streamflow. But when we computed the skill scores with reference to the raw System 4 forecasts (to calculate the UFL), scores were computed for daily values. In this case, the moving average allowed us to remove the high frequency variations in the skill

scores while looking at the impact of bias corrections on daily forecast values. We will clarify this in the revised version.
* * *
**RC: Page 9, line 12:** `Why is the value +0.1 and -0.1 for the deviations from the diagonal chosen?`

AR: Laio and Tamea (2007) propose to calculate the position of these "tolerance" lines to correspond to a significance test: *« The Kolmogorov bands are two straight lines, parallel to the bisector and at a distance q(α)/sqrt(n) from it, where q(α) is a coefficient, dependent upon the significance level of the test α (e.g., q(α = 0.05) = 1.358, see D'Agostino and Stephens, 1986). The test is passed when the curves remain inside these confidence bands. ».* In our case, the Kolmogorov significance bands should be approximately at 0.15 to correspond to a 5 % significance test. The 0.1 bands we use are thus a good conservative choice to test deviations from the diagonal.

*Laio F. and Tamea S.: Verificatin tools for probabilistic forecasts of continuous hydrological variables. HESS, 11, 1267-1277, doi: 10.5194/hess-11-1267-2007,2007.*
* * *
**RC: Page 9, line 19:** `Unclear use of the word "translate".`

AR: We propose to replace "translate" by "indicate" in the revised version.
* * *
**RC: Sections 4-6:** `The presentation of the results could be improved and shortened. When I read the manuscript, I would have liked to have the comparison of the raw and bias corrected forecasts closer together and I suggest combining the discussion of the raw and the EMDD corrected forecasts. It would be much easier for the reader to follow the discussion if, for e.g., figure 2 and 10 are to be combined into one figure. Similarly for all other seasonal skill score figures in sections 4 and 6.`

AR: Reviewer 3 also proposed to have the figures of Sections 4 and 6 closer together for an easier visual comparison of the performances before and after bias correction. We will consider these suggestions in the revised version of the paper.
* * *
**RC: Page 11, line 8-9:** `I thought that the reference forecast is the streamflow simulated using the reference precipitation. Thus, any model deficiencies regarding low flows should not affect the skill score as also the reference forecast would suffer from those deficiencies. Also, similarly as for the low flows, the PIT diagram reports difficulties to forecast the high flow. What could be the reason for this issue? The explanations give in the manuscript so far are not fully convincing.`

AR: The reference forecast in the computation of the skill scores uses the hydrological model (in all figures but Figure 13), and model deficiencies cannot be detected based on the graphs of skill scores, as well noted by the reviewer. Here, however, the explanations proposed on lines 8-9 refer to the PIT diagrams of Figure 5 (which are not expressed as skill scores) and, more specifically, to the lack of reliability observed in the summer season (JJA). The tendency to have observations below the forecast range is obtained with both the streamflow simulated with System 4 precipitations (in red) and the streamflow simulated with the reference precipitation (in grey). This is why we make the assumption that this lack of reliability is due to the hydrological model rather than the precipitation forcings. We also should note that it is hard to distinguish between high and low flows based on the PIT diagram solely. In summer, we have observed an under-dispersion of forecasts, but also a strong tendency to have observations falling below the forecast range. From the hydrographs, we also observed that a large part of the observations falling in the lowest forecast range in summer can be associated with low flows. The PIT diagram thus needs to be analysed together with the hydrographs to better separate the effects of under-dispersion on high or low flows. We will clarify this in the revised version of the paper.
* * *
**RC: Page 14, lines 19-23:** `The reasoning is unclear to me, probably due to an unclear explanation how the bias-correction works. If it is done in the way I described in the comment regarding EDM and GDM, I don't think that the reasoning is`

correct. Everything stated for the monthly correction would also apply for the daily correction. Also on the daily time scale, the rank structure (see comment below) of the forecast is not the same as for the reference data. In both cases (monthly and daily correction), the distribution mapping should be able to correct differing rank structures and remove biases in the monthly mean effectively. In fact, I would have expected the daily correction to perform worse than monthly correction when evaluated on the monthly scale since it is not targeted to the monthly scale but the daily scale. I rather think it has to do with a higher sensitivity of monthly corrections to overfitting as evaluated within the cross-validation framework. Admittedly, distribution mapping can lead to unforeseen effects and it might very well be that I'm wrong. If the authors are convinced that their reasoning is correct, I would like them to describe in the reply a case where the distribution mapping fails in more detail, for e.g. by showing how the reference and forecast distribution look like and how the mapping fails to come up with a correct monthly mean value.

**Page 14, lines 19 and 21:** Usage of the term "time structure" seems to be misleading. I understand this term in a way that it refers to the temporal sequence of values, i.e. that the day n in the reference corresponds to day n in the forecasts. However, distribution mapping does not have this requirement. It is rather the rank structure as I would call it: Rank n in the reference has to correspond to the rank n in the forecasts. Please correct the terminology or explain in more detail what "time structure" means.

AR: We believe that compensation effects (linked to data aggregation) may occur when evaluating monthly values with bias corrected daily values. Since daily corrections are more numerous than monthly corrections, this can result in more flexibility and daily correction performing better than monthly correction when evaluated at the monthly scale. This is more or less similar to monthly correction performing well when evaluated at the yearly scale. However, as mentioned by the reviewer, this may also be linked to a "higher sensitivity of monthly corrections to overfitting". Further studies would be necessary to conclude more firmly on this issue. We will revise the text in order to be more careful on the comment we made.

Concerning "time structures", we agree that the terminology may not be very clear. We meant that, for the DM methods to be efficient, the uncorrected and corrected values with the same rank (in their respective cumulative distributions) should also occur at the same time (e.g. as observed in the hyetograph). Therefore, the "time evolution" of values should be consistent and DM methods will be more efficient if they are applied on forecast hyetograph that are not too discrepant. We believe that this can go unnoticed in performance evaluation if daily values are corrected and aggregated at the monthly scale for evaluation. However, it will be harder to cover up if we apply and evaluate the corrections at the same time scale. We will be more precise about this in the revised version in order to clarify our idea.
* * *
**RC: Section 5.2:** In my opinion, this section does not give new information which is not already present in figure 6 (time varying bias-correction factors can be inferred from the panel "Before bias correction") and I suggest removing it for the sake of shortening the result section. The only new aspect is that the correction factors for EDMD vary more than for LS, but this comparison is not valid in my point of view as one should not compare a mean correction factor with a correction factor for a quantile level. I'm pretty sure that if you would calculate the correction factor for the mean in the case of EDMD, it would be very similar to the LS factor.

AR: The idea behind this plot was not to compare LS and EDMD average correction factors (we fully agree with the reviewer that this comparison is not valid). In fact, we wanted to give an additional element to understand the different features behind the LS and the EDMD methods. The main conclusions from this figure are that (1) EDMD can correct the frequency of null precipitations, whereas LS cannot, and (2) correction coefficients do not vary much from one application year to the other (especially with LS) and, therefore, in operational contexts, one can choose a more parsimonious calibration of the bias correction method applied. We will consider the ways to shorten this section in the shortened revised version of the paper.
* * *
**RC: Section 5.3 and figures 8 and 9:** I very much like this analysis. I'm not sure though if I really understand the analysis completely. MAE is partly related to the

bias analysis in figure 6, i.e. if biases in figure 6 are substantial, then MAE
should be even larger since MAE does not allow for a compensation of errors. EDM-y
and GDM-y have large biases throughout the year in figure 6, and in some cases and
particularly in summer, the bias is even larger than in the uncorrected data.
However, in figure 8 the two methods stick out for MAE and IQR in summer lead to
skill improvements in all catchments up to a lead time >60 days. To me, this seems
to be contradicting. Could you please explain this particularity?

AR: Thank you. Fig. 8 reflects, somehow, the ability to bring skill to the (corrected) forecasts in terms
of lead time and expressed as a percentage of catchments where improvements (comparatively to the
raw forecasts used as reference) were seen. Even if EDM-y and GDM-y methods result in forecasts
that still present some strong biases (as seen in, and commented from, Fig. 6), these may result in
MAE values smaller than MAE values computed from the raw forecast. This is enough to characterize
a relative gain in skill and, if this is observed over all lead times, the UFL will be >60 days and count
in the percentage represented in Fig. 8. It seems contradictory at first sight, as well observed by the
reviewer, but can, computationally, happen (e.g. due to the different aggregations: MAE is computed
with daily values over a season, the bias is computed with monthly values over a month, or when
biases change from under to over prediction or vice versa after correction). Overall, it is interesting to
note that forecast skill is definitely hard to evaluate as there are many facets that one can look at. We
tried to explore this in this paper and shed lights into the different aspects that can better inform
forecast users.
* * *
RC: Page 19, line 1: If I read the figure 10 correctly, there are negative skill
score values and therefore, the statement that the skill scores are always larger
than zero does not hold.

AR: You are right. We propose to replace the sentence with: "Nevertheless, bias corrected forecasts
remain sharper than the reference (i.e., skill scores are mostly greater than zero)."
* * *
RC: Page 20, line 3: If I read the figure 12 correctly, there are negative skill
score values and therefore, the forecast performs sometimes worse than ESP, which
is the opposite of what is stated on this line.

AR: By "overall" we meant to indicate that. We propose to clarify it by changing to: "Overall, after
bias correction, streamflow forecasts are sharper than ESP in all catchments and seasons (only the
winter and summer seasons are shown but results are similar for the spring and autumn seasons)".
* * *
RC: Page 20, lines 12-13: It is not clear to me why this is expected. I would
expect that comparison to streamflow climatology is a harder check and therefore
the skillfull lead time should be smaller than in the comparison to the baseline
reference run since also the hydrological model bias deteriorates the skill. I
surely misunderstand something but I think it would be good to add a bit more
explanation in the manuscript.

AR: It is usually expected that ensembles based on streamflow climatology have less skill than
ensembles based on hydrological modelling, at least in the first lead times, because ensembles based
on hydrological modelling benefit from knowledge of initial hydrologic conditions. For instance, here,
the states of the GR6J model are first initialized by running the model with observed inputs for a year
prior to the forecast date. Therefore, ensembles based on streamflow climatology are supposed to be
less skilful for forecast lead times that are impacted by initial hydrologic conditions. In France, studies
have shown that these lead times extend to a month, on average. We will make sure it is clearer in the
revised version.
* * *
RC: Page 22, line 4: As for the uncorrected forecast discussion, I do not
understand why it is the hydrological model that causes the problems with low-flow
overestimation. The reference data is also output of the same hydrological model
driven by the reference precipitation data. I would therefore rather think that it
is some characteristics in the input data which the bias-correction cannot correct
for that causes the problem (for e.g. dry-spell lengths). If the authors still
think their statement holds, I would like to have a bit more explanations why this
can be the case.

AR: The main discussion here is about reliability, which can clearly still be improved for streamflows. Comments on the model performance are linked to the analysis of the simulated and observed hydrographs, which complement the PIT analysis. The lack of reliability in streamflow forecasts may come from the input data, but not solely (as shown in Fig. 3, which analyses the reliability of the precipitation forcing; please also refer to the answer given above referring to Fig. 5, `RC: Page 11, line 8-9`). A lack of spread in hydrologic initial conditions may also play a role in the reliability of streamflow forecasts. That is why we referred to the needs of accounting for other sources of uncertainty, with, for instance, additional post-processing (lines 4-6). This may not be clearly stated and we will clarify it in the revised version.
* * *
`RC: Page 27, lines 13-15:` References needed

AR: We propose to add the following references:

*Hamlet, A. F. and Lettenmaier, D. P.: Columbia River Streamflow Forecasting Based on ENSO and PDO Climate Signals, J. Water Resour. Plan. Manag., 125(6), 333–341, doi:10.1061/(ASCE)0733-9496(1999)125:6(333), 1999.*

*van Dijk, A. I. J. M., Peña-Arancibia, J. L., Wood, E. F., Sheffield, J. and Beck, H. E.: Global analysis of seasonal streamflow predictability using an ensemble prediction system and observations from 6192 small catchments worldwide, Water Resour. Res., 49(5), 2729–2746, doi:10.1002/wrcr.20251, 2013.*

*Werner, K., Brandon, D., Clark, M. and Gangopadhyay, S.: Climate index weighting schemes for NWS ESP-based seasonal volume forecasts., J. Hydrometeorol., 5(6), 1076–1090, 2004.*

In addition, the last paragraph will include the following reference:

*Ionita, M., Boroneant, C., Chelcea, S., 2015. Seasonal modes of dryness and wetness variability over Europe and their connections with large scale atmospheric circulation and global sea surface temperature. Climate Dynamics 45, 2803–2829. doi:10.1007/s00382-015-2508-2*
* * *
`RC: Section 6.3:` Just a comment: I very much like this analysis.

AR: Thank you!
* * *
`RC: Section 6.4:` Although I like the illustrative character of this section, it stands a bit loose within the rest of the manuscript. I suggest to either motivate the section better or, for the sake of brevity, to remove it. In my opinion, the main statements of this sections have already been made, i.e. increased sharpeness after bias-correction compared to ESP.

AR: Reviewer 1 also appreciated this figure, so we prefer to keep it. He proposed some improvements that we think will better motivate the section: adding a quantification of what is shown in this figure. Notably, we show that the coverage probability of the streamflow forecasts is improved after bias correction compared to ESP (see our answers to Reviewer 1 for details). Additionally, studies have shown the need to combine statistical evaluations with visual evaluations. Even though this is hard to achieve in probabilistic forecasting, we wanted to propose a visual appreciation of the ensembles to have a better overview of how bias corrections affect streamflow forecasts.
* * *
`RC: Figure 2:` "… and all seasons." The figure only shows two seasons, please correct the caption.

AR: Correct: "… all seasons" will be replaced by "and the winter (DJF) and summer (JJA) seasons".
* * *
`RC: Figures 3, 5, 11, 14:` The dashed lines should be explained in the figure as well, and not just in the text describing figure 3.

AR: The explanation will be added in the captions.
* * *
`RC: Figure, 6:` Although certainly correct, I do not see a reason why to transform the simple relative bias into 1-bias. I understand that this transformation turns the bias into a skill score. However, in my opinion, the interpretation is not following the one for skill scores anyway. The perfect bias-correction would not

yield 1 but 0. I suggest plotting the relative bias without transformation. The
scale would be much easier interpretable as it directly refers to a percentage
over- or underestimation.

AR: We used this transformation so that "no bias" corresponded to the null value, over-prediction corresponded to positive values and under-prediction corresponded to negative values. This representation of the scale seemed more intuitive, but the reviewer is right that the interpretation in terms of percentage is easier without this transformation. We will test and consider the reviewer suggestion when preparing the revised version.
* * *
RC: **Figures 8 and 9:** Why are there different color scales for the different
seasons?

AR: The four colours are supposed to help the reader identify the four seasons throughout the article. These colours include the blue and red colours used throughout the paper: blue for winter, lighter blue for autumn, red for summer and lighter red for spring. In these figures, the four colour scales in the legend are needed to clarify the colour shades related to the percentage of catchments in each category (e.g. to avoid light blue (autumn) being mistaken for a shade of bright blue (winter)).
* * *
RC: **Figure 15:** What are the colours standing for? There is probably also an error
in the caption where it reads "shown for all seasons".

AR: The colours represent the four seasons as mentioned in the reply above. We will add an explanation in the figure.
* * *
RC: **Page 19, line 10:** precipitation instead of precipitations

AR: This will be modified in the revised version.

---

## Author Response (AR1)

**Main changes in the revised version and detailed answers to the reviewers**

We thank the reviewers for their comments and suggestions, which helped us to improve our paper. The main changes in the revised version concern the following issues:

- We added some sentences in order to: i) better explain how PET was considered, ii) present the KGE results of the calibration and validation of the hydrological model, as requested by two reviewers, iii) add pragmatic results on the interpretation of the hydrographs, as requested by one reviewer, which clarifies the role of the bias correction methods in improving forecast quality.
- We shortened considerably the paper, as suggested by two reviewers and by the Editor, by cutting several words and shortening several sentences, which we thought could be deleted or rephrased without affecting the message the paper conveys. We also deleted an entire section on the corrective factors, which we believe was not essential for the comprehension of the main findings of the paper.

All other minor changes and answers to the reviewers are detailed below.

**Reviewer 1**
* * *
**Reviewer's comment (RC):** While the Introduction is well balanced and gives useful insight on previous work on the topic and also references supporting the envisaged methodology, I found that the final paragraphs should possibly include more information on the novelty of the present manuscript. Also in the methodological section some more referencing is needed. See minor comments for this.

Authors' reply (AR): We changed the following in the Introduction to better emphasize the novelty of our study (end of line 29, beginning of line 30, page 3): "Despite these recent works, and to the knowledge of the authors, no previous study has compared bias correction methods and their impact on streamflow forecasting in a systematic way, with a focus on understanding how the main attributes of forecast performance are impacted by bias correction.

This paper aims to provide insights into the way bias correcting seasonal precipitation forecasts can contribute to the skill of seasonal streamflow predictions, notably in terms of overall performance, reliability, sharpness and skilful lead time. It investigates the potential of bias corrected ECMWF System 4 forecasts to improve streamflow forecasts at extended lead times over 16 catchments in France. An in-depth comparison of eight variants of linear scaling and distribution mapping methods applied over the 1981-2010 period is presented. Section 2 presents…"
* * *
**RC:** 4-4-15: We learn here about the meteorological forcing. It is clear to me how you use precipitation, but as a forecast and as SAFRAN product. Concerning Potential Evapotranspiration (PET), only SAFRAN is declared. I'd like you to declare which PET is used in retrospective forecasts forced by the ECMWF products. If it is from ECMWF, you should state why you are not post-processing it. If you use SAFRAN, you should be able to assess how much uncertainty are you neglecting by using the best observed estimates of PET instead of using a forecasted value (which you need to do as soon as you will deploy the system in real-time). In our experience, for basins not affected by snow-melt, the post-processing of relative-humidity data (an important proxy the evaporation demand by the atmosphere) helps improving the estimation of hydrological droughts (Jörg-Hess et al., 2015).

AR: The potential evapotranspiration (PET) used to force the hydrological model is, in fact, the mean interannual PET. For a given day of the year, the estimated PET on this day is assumed to be the mean of all PET computed for this day of the year, in all available years. Here, the mean interannual PET is the average of the PET calculated for each year from 1958 to 2010. PET for each year is calculated using SAFRAN. Regardless of the precipitation scenario fed to the model (historical precipitations or System 4), the PET scenario used as input to the model is always the same: the series of mean interannual PET

corresponding to the forecast period. With this setup, we can focus on the changes in skill that can solely be attributed to the bias correction of precipitations, which is in the aim of our study. Adding the uncertainty of temperature forecasts in the analysis would in fact require a different framework. For instance, we would need to set up multi-variable bias corrections to take into account the dependencies between precipitation and temperature, or we would need to consider the impact of observed trends in time series of observed temperatures in some regions in France prior to post-processing and ESP forecasting. This is beyond the scope of this study, although interesting for further investigations and specific operational setups.

In the revised version, we clarified the way PET is considered by adding this sentence in Section 2.1: "The interannual potential evapotranspiration was then computed in each catchment, i.e. for a given day of the year, we computed the average potential evapotranspiration for this day over all available years (1958 to 2010)", and the following in Section 2.2: "Here, the series of interannual potential evapotranspiration corresponding to the forecast period was systematically used as input to the hydrological model. With this setup, we aimed to isolate the influence of precipitation forecast inputs on the quality of streamflow forecasts."
* * *
**RC:** 4-25: I just reviewed another paper on seasonal forecasting where authors did not show any score concerning their calibration/validation and I amended it. Same here. I am happy with a table as supplementary material.

AR: The table below summarizes the scores obtained in calibration and validation of the GR6J model. In the revised version, we added the following sentence in Section 2.2: "We obtained an average KGE of 0.95 in calibration and 0.94 in validation over the sixteen catchments. The bias obtained in simulation ranges from 0.95 to 1.02."

| Catchment | Calibration KGERQ | Validation KGERQ | Validation C2MQ | Validation Bias |
|---|---|---|---|---|
| 1 | 0.93 | 0.92 | 0.75 | 0.99 |
| 2 | 0.93 | 0.92 | 0.65 | 0.97 |
| 3 | 0.94 | 0.94 | 0.64 | 0.95 |
| 4 | 0.94 | 0.94 | 0.72 | 0.98 |
| 5 | 0.94 | 0.94 | 0.69 | 1.00 |
| 6 | 0.95 | 0.95 | 0.77 | 1.02 |
| 7 | 0.95 | 0.95 | 0.79 | 0.97 |
| 8 | 0.97 | 0.97 | 0.87 | 0.98 |
| 9 | 0.97 | 0.97 | 0.84 | 1.01 |
| 10 | 0.89 | 0.88 | 0.58 | 1.00 |
| 11 | 0.95 | 0.95 | 0.81 | 0.96 |
| 12 | 0.95 | 0.95 | 0.82 | 0.96 |
| 13 | 0.93 | 0.93 | 0.86 | 0.95 |
| 14 | 0.96 | 0.96 | 0.88 | 0.97 |
| 15 | 0.97 | 0.97 | 0.84 | 0.98 |
| 16 | 0.95 | 0.94 | 0.81 | 0.96 |
* * *
**RC:** 24-18: I like this evaluation very much, just, I miss some quantification supporting the description based on visual inspection you are giving. Be pragmatic.

AR: Thank you very much for this suggestion. The MAE and coverage probability provide a good quantification to support the description. The values of MAE, coverage probability 90% (COV 5-95) and 50% (COV 25-75) obtained by each forecasting system over the displayed period (April 2004 to April 2007) are shown below. They show that the ensembles based on past precipitations and past streamflow (HistQ and ESP) are more accurate over the chosen period (lower MAE values), but that EDMD-m performs better in terms of coverage probability. We included this quantitative analysis in the interpretation of the hydrographs, as suggested by the reviewer.

| | HistQ | ESP | LS-m | EDMD-m |
|---|---|---|---|---|
| MAE ($m^3$/s) | 3.81 | 4.06 | 4.26 | 4.26 |
| COV 90 % (5-95) | 97 % | 92 % | 85 % | 89 % |
| COV 50 % (25-75) | 66 % | 60 % | 46 % | 51 % |

**RC:** 25-3: The discussion section is here quickly merged with the conclusions. The only link to current literature one is expecting here merely consists in a enumeration of possible post-processing of the forecasts with currently available methods. Here some more effort has to be shown to make also this section a valuable part of the manuscript.

AR: The reviewer is right that this section reflects more our conclusions. A posteriori, we think that sections 6.3 and 6.4 reflect the discussions, putting the results into a broader perspective. We thus renamed the last section "Conclusions".
* * *
**RC**: 26-2: You address here the issue of implementation in operational systems. Again, declare how you deal the PET, and then re-evaluate the potential for real-time operations.

AR: We added some sentences to better explain how PET was considered (see reply above) and we deleted the sentence on operational issues to avoid introducing a discussion that is not the focus of this paper.
* * *
**RC:** 2-11: I guess here you should give one or two references for the statistical models, too. Eg. Some approaches relating winter snowpack to summer-flows (e.g.: Godsey et al., 2014; Jenicek et al., 2016).

AR: Thank you for pointing out to these interesting references. We added Jenicek et al. (2016) (mentioning references therein) as suggested.
* * *
**RC:** 5-4: Please support the "one-year-leave-out cross-validation method" with a reference.

AR: We added the following reference and changed the "one-year-leave-out" denomination to "leave-one-year-out" for consistency:

Arlot, S. and C. Alain. A survey of cross-validation procedures for model selection. Statist. Surv. 4 (2010), 40-79. doi:10.1214/09-SS054. http://projecteuclid.org/euclid.ssu/1268143839.
* * *
**RC:** 6 -2: Please support "Precipitation and streamflow forecasts are evaluated with deterministic and probabilistic scores commonly used in ensemble forecasting" with a reference, e.g. Brown et al EVS paper.

AR: In the process of reducing the length of the paper, this sentence was removed. Yet, references supporting the evaluation criteria can be found in Section 3.3.
* * *
**RC:** 8-15: Nice idea to use the ensemble of past-streamflow observations as a reference. If you would "sort-out" some past years by means on analogues techniques you might get a very challenging set of members for your ensemble forecast. Have you tried this?

AR: This is precisely the topic of another paper that we have just submitted to this special issue. It is available here: http://www.hydrol-earth-syst-sci-discuss.net/hess-2016-285/
* * *
**RC:** 8-22: Another interesting feature here. This definition of gain is very elucidative. Can you maybe elaborate on pro and contra of this kind of "gain" definition with respect to scores based on cost-loss considerations?. Why choosing such a large gap of day between the classes? Have you tried to make a 30-day moving window? Or a 15-days moving window?

AR: Thank you for the comment. We chose to evaluate the gain in terms of anticipation in response time, rather than in terms of relative economic value (REV), for instance, since cost-loss considerations would need an evaluation of (or additional assumptions on) mitigation costs, avoidable losses, as well as unavoidable losses for each studied catchment. Here, we may assume that increasing the anticipation response time could increase time for preparedness, which would decrease costs and losses related to missed events or actions taken with no or little anticipation to a critical situation. The cost-loss approach would need to be applied considering this evolution of forecasting with time since in a seasonal forecasting system one has several forecasts or months ahead to detect a potential critical situation and

act accordingly. Actions and consequences would need to be stratified according to the time available for action in order to have this aspect reflected in an evaluation score.

The gap was chosen to help represent the improvements due to bias correction at a monthly time scale of reference. It seemed to us that a month ahead could be a good minimum of time necessary to adapt any mitigation actions once a critical situation is forecasted by a seasonal forecasting system. As shown in Figures 8 and 9, this choice seems to be appropriate to a joint representation in a plot, while differentiating situations for a useful analysis.

We did not try to use windows larger than 7 days. The objective of the rolling mean was to smooth the skill curves and remove the high frequency variations of the skill at the daily time step. Seven days appeared to be enough to smooth the curves, while keeping the moving mean as a good estimate of the gain in lead time.
* * *
**RC:** 9 – 2 & 9 -19: Both in Figure 2 & 4 CRPSS is showing increasing skill at weeks 5 and 9. We are also used to "struggle" in interpreting such cycles. Do you have some ideas on your particular case here?

AR: We have also spent some time trying to interpret these cycles. Despite a closer look at the data and the scores, under different angles, we could not see any systematic reasons for these cycles. We think it may be related to several correlated aspects, such as the type of forecasting model/system, the forcings, the behaviour of the catchment, etc.
* * *
**RC:** 11 – Figure 4: How would look like this figure if you use the "ensemble based on past streamflow" as a reference?

AR: The following figure represents the IQRSS and the CRPSS of the streamflow forecasts without bias correction, when the ensemble based on past streamflow is used as reference. It can be compared to Figure 4 to see that the skill is higher with this reference, and to Figure 13 to see that bias correction has also increased the skill of forecasts with regard to past streamflow.

[Figure]
* * *
**RC:** 13 – Figure 6: Right margin is cropped. Additionally, the "too wet"=red is not really intuitive.

AR: We exchanged the blue and red colours in the scale so that blue corresponds to overestimation and red corresponds to underestimation. We also increased the right margin.
* * *
**RC:** 13 – 2: "the 2-month" or the "month-2"? If you mean the one for the second month of the forecast I would find more adequate to use "month-2".

AR: We agree with the reviewer and changed the occurrences of "the 2-month" to "month-2" in the revised version.

**RC:** 17 – Figure 8 (and later 9): I like such Figures because the train my brain. Tell me if I am reading it wrong: If a look at a certain score in a certain season than for a particular bias correction method a percentage of the basins is showing improvement in lead time. Of this percentage a distinct portion shows improvement of let's say 60 to 90 days. So largest improvement is in the PIT-Skill in summer and Winter for the EDMD methods.
Right?
AR: Thank you. Your reading of the figure is absolutely correct.
* * *
**RC:** 22 – 8: This would be the only heading with a question mark. Maybe replace this with a sentence
AR: In fact, the question mark was a typo and we removed it.
* * *
**RC:** 23 – Figure 15: is there any special reason (beside readability) for having different scales in the three panels?
AR: No, there is no special reason, apart to zoom in on the case of the CRPSS. Following this comment, we changed the figures and used the same scales in the axes.

**Reviewer 2**
* * *
**Reviewer's comment (RC):**
 • The study uses mainly modelled streamflow as a reference. Nevertheless, I miss some indication of the hydrological model performance in the 16 basins. This is particularly relevant since also the observed streamflow is used as a reference forecast in one part of the manuscript, and this analysis would critically depend on systematic biases of the hydrological model.

**Authors' reply (AR):** Reviewer 1 also recommended indicating the performance of the hydrological model (see our reply above). We added the following sentence in Section 2.2: "We obtained an average KGE of 0.95 in calibration and 0.94 in validation over the sixteen catchments. The bias obtained in simulation ranges from 0.95 to 1.02."
* * *
**RC:**  • The manuscript covers a large body of results and is therefore lengthy. I think that it could be streamlined without losing too much information.
AR: We cut several words and sentences along the text and removed an entire sub-section concerning the corrective factors (Section "Comparison of bias correction factors for LS and EDMD methods"), which reduced the length of the paper.
* * *
**RC:** Over all, I suggest acceptance of the manuscript after my comments have been taken into account. I'm looking forward to the revised manuscript.
AR: We thank the reviewer for this positive appreciation of our paper.
* * *
**RC: Page 3, line 1:** Some reference needed to support the statement that linear scaling and distribution mapping are widely used methods in seasonal forecasting.
AR: We added a reference to the review paper of Yuan et al. (2015).
* * *
**RC: Page 4, line 13:** Which parametrization was used to derive potential evapotranspiration?
AR: The calculation of the evapotranspiration was done prior to this study and is embedded in the database we used. It follows the Oudin formula, which can be found in Equation (2) of the reference Oudin et al. (2005). $K_1$ is set to 100 and $K_2$ to 5, as shown in Equation (3) of this same paper. This reference is cited in the paper.

**RC: Page 4, line 23:** What is meant by interannual potential evapotranspiration? I would have understood the manuscript in such a way that potential evapotranspiration is derived from raw, i.e. non-bias-corrected, forecasts, but in this case, the term interannual potential evapotranspiration does not make sense. I probably misunderstood something and would like that the authors clarify the manuscript in that respective.

AR: For a given day of the year, the estimated PET on this day is assumed to be the mean of all PET computed for this day of the year, in all available years. Here, the mean interannual PET is the average of the PET calculated from observed temperatures for each year from 1958 to 2010.

In the revised version, we clarified the way PET is considered by adding one sentence in Section 2.1 and one sentence in Section 2.2 (see also our reply to Reviewer 1 above).

**RC: Page 5, lines 3-4:** Just a comment, nothing to change: leave-one-year-out might result in the validation years not being really independent, as interannual serial correlation might be quite high. Maybe it would be interesting to test larger block sizes in future studies.

AR: Definitely. This point was also recently raised in a HEPEX bog post by colleagues from CSIRO (http://hepex.irstea.fr/how-good-is-my-forecasting-method-some-thoughts-on-forecast-evaluation-using-cross-validation-based-on-australian-experiences/). We think that a more-than-one-year-leave-out procedure could potentially fit better for one of our catchments, which has a high base-flow index. We believe that its impact on the other catchments would be lower, given the length of our calibration periods. Also, the impact is expected to be lower when calibrating the hydrological model than when implementing the bias corrections. In any case, it would certainly be interesting to test it in a future study, where more catchments could also be included and focus could be put on this aspect.

**RC: Page 6, lines 21-25:** In the case of EDM-m and GDM-m, only 29 data points are used to derive a cumulative distribution function for the reference data. This is a rather low number of data points, potentially leading to estimated cumulative distributions that are non-robust. Maybe, and this is of course rather speculative without analyzing the data, this could be a reason for the worse bias validation of EDM-m and GDM-m in Fig. 6.

AR: This is also an interesting point and could, as suggested, partly explain the poorer performance of EDM-m and GDM-m. Nevertheless, it is also worth noting that it is difficult to have much more years available for the calibration of these correction methods, since the meteorological reforecast archive needs to be homogeneous (i.e., based on the same model) over the period. The fact that bias correction methods require long time series of forecasts is a well-known limitation in the field.

**RC: Page 6, lines 21-25:** I'm not aware of a study that applied gamma distribution fitting for monthly precipitation data. Could you please cite a study to support the method GDM-m? I'm a bit worried that the gamma distribution might not be a good choice for monthly mean precipitation values.

AR: The choice of a cumulative distribution function for precipitation (or streamflow) data is always a challenging one. The Gamma distribution is often assumed to be suitable and fitted to precipitation sums. Some examples of the gamma distribution fitted to monthly precipitations are:

*Zekai S. and A. G. Eljadid (1999) Rainfall distribution function for Libya and rainfall prediction, Hydrological Sciences Journal, 44:5, 665-680, DOI:10.1080/02626669909492266,*

*Husak, G. J., Michaelsen, J. and Funk, C. (2007), Use of the gamma distribution to represent monthly rainfall in Africa for drought monitoring applications. Int. J. Climatol., 27: 935–944. doi:10.1002/joc.1441*

The gamma distribution is also often used when computing the SPI. Examples are:

*Lavaysse, C., Vogt, J., and Pappenberger, F.: Early warning of drought in Europe using the monthly ensemble system from ECMWF, Hydrol. Earth Syst. Sci., 19, 3273-3286, doi:10.5194/hess-19-3273-2015, 2015.*

*X. Lana, A. Burgueño, M. D. Martínez and C. Serra: A review of statistical analyses on monthly and daily rainfall in Catalonia. 2009. Tethys (Journal of Weather & Climate of the Western Mediterreanean), 6, 15–29, 2009, doi:10.3369/tethys.2009.6.02.*

In the preliminary steps of our study, we visually compared several distributions to fit to monthly precipitations in the selected catchments. The gamma distribution showed the best fit to the empirical distributions.
* * *
**RC: Page 6, lines 25:** It is unclearly written how exactly the EDM and GDM correction is applied to daily values. I assume it is done as such that the monthly values are corrected following the quantile mapping procedure. After that, a correction factor is estimated between the corrected and the uncorrected monthly mean value and this correction factor is applied to all daily values. The text on line 25 is though misleading as the actual correction in a quantile-mapping framework is the mapping of the uncorrected values to the cumulative probability space, from which a corrected value is derived following an inverse mapping based on the reference data. As the mapping is calibrated for monthly values, it cannot be used for daily values directly. Please clarify the text.

AR: The reviewer has understood correctly how the EDM and GDM methods are applied. In order to clarify it in the revised version, we added the following sentence: "In the case of EDM and GDM, the monthly values are first corrected based on the distribution mapping procedure. Then, for a given month, the ratio of the corrected monthly value and the non-corrected ones is used to correct all daily values within this month."
* * *
**RC: Page 8, lines 23-24:** The first sentence in this paragraph is redundant. Consider removing it.

AR: This was done and only one sentence appears in the revised version: "To investigate the gain in performance brought by bias correction methods, we use the raw (uncorrected) forecasts as reference in the computation of the skill scores."
* * *
**RC: Page 8, lines 27-28:** According to section 3.3, all data was first converted to weekly means, thus a seven-day moving average cannot be derived. Please clarify the contradictions.

AR: When computing skill scores with reference to the ESP or historical streamflow, we computed scores based on weekly-averaged precipitation or streamflow. But when we computed the skill scores with reference to the raw System 4 forecasts (to calculate the UFL), scores were computed for daily values. In this case, the moving average allowed us to remove the high frequency variations in the skill scores while looking at the impact of bias corrections on daily forecast values.

In the revised version, we clarified this point. We changed the first sentence of Section 3.3 which was too general to the following: "The quality of the forecasts was evaluated as a function of lead time and for the winter (December-January-February), the spring (March-April-May), the summer (June-July-August) and the autumn (September-October-November) seasons".
* * *
**RC: Page 9, line 12:** Why is the value +0.1 and -0.1 for the deviations from the diagonal chosen?

AR: Laio and Tamea (2007) propose to calculate the position of these "tolerance" lines to correspond to a significance test: *« The Kolmogorov bands are two straight lines, parallel to the bisector and at a distance q(α)/sqrt(n) from it, where q(α) is a coefficient, dependent upon the significance level of the test α (e.g., q(α = 0.05) = 1.358, see D'Agostino and Stephens, 1986). The test is passed when the curves remain inside these confidence bands. ».* In our case, the Kolmogorov significance bands should be approximately at 0.15 to correspond to a 5 % significance test. The 0.1 bands we use are thus a good conservative choice to test deviations from the diagonal.
* * *
**RC: Page 9, line 19:** Unclear use of the word "translate".

AR: We replaced "translate" by "indicate" in the revised version.
* * *
**RC: Sections 4-6:** The presentation of the results could be improved and shortened. When I read the manuscript, I would have liked to have the comparison of the raw and bias corrected forecasts closer together and I suggest combining the discussion of

the raw and the EMDD corrected forecasts. It would be much easier for the reader to follow the discussion if, for e.g., figure 2 and 10 are to be combined into one figure. Similarly for all other seasonal skill score figures in sections 4 and 6.

AR: The manuscript was shortened by removing sentences, repetitions and by removing a whole sub-section. Concerning combining figures with and without bias correction from EDMD-m, this would in fact disturb the logic of the paper since, in between these figures, we propose an in-depth analysis of the impact of the bias corrections per month and for each tested method. We then preferred to keep the organization as it was first proposed.
* * *
RC: Page 11, line 8-9: I thought that the reference forecast is the streamflow simulated using the reference precipitation. Thus, any model deficiencies regarding low flows should not affect the skill score as also the reference forecast would suffer from those deficiencies. Also, similarly as for the low flows, the PIT diagram reports difficulties to forecast the high flow. What could be the reason for this issue? The explanations give in the manuscript so far are not fully convincing.

AR: The reference forecast in the computation of the skill scores uses the hydrological model (in all figures but Figure 13), and model deficiencies cannot be detected based on the graphs of skill scores, as well noted by the reviewer. Here, however, the explanations proposed on lines 8-9 refer to the PIT diagrams of Figure 5 (which are not expressed as skill scores) and, more specifically, to the lack of reliability observed in the summer season (JJA). The tendency to have observations below the forecast range is obtained with both the streamflow simulated with System 4 precipitations (in red) and the streamflow simulated with the reference precipitation (in grey). This is why we make the assumption that this lack of reliability is due to the hydrological model rather than the precipitation forcings. We also should note that it is hard to distinguish between high and low flows based on the PIT diagram solely. In summer, we have observed an under-dispersion of forecasts, but also a strong tendency to have observations falling below the forecast range. From the hydrographs, we also observed that a large part of the observations falling in the lowest forecast range in summer can be associated with low flows. The PIT diagram thus needs to be analysed together with the hydrographs to better separate the effects of under-dispersion on high or low flows.
* * *
RC: Page 14, lines 19-23: The reasoning is unclear to me, probably due to an unclear explanation how the bias-correction works. If it is done in the way I described in the comment regarding EDM and GDM, I don't think that the reasoning is correct. Everything stated for the monthly correction would also apply for the daily correction. Also on the daily time scale, the rank structure (see comment below) of the forecast is not the same as for the reference data. In both cases (monthly and daily correction), the distribution mapping should be able to correct differing rank structures and remove biases in the monthly mean effectively. In fact, I would have expected the daily correction to perform worse than monthly correction when evaluated on the monthly scale since it is not targeted to the monthly scale but the daily scale. I rather think it has to do with a higher sensitivity of monthly corrections to overfitting as evaluated within the cross-validation framework. Admittedly, distribution mapping can lead to unforeseen effects and it might very well be that I'm wrong. If the authors are convinced that their reasoning is correct, I would like them to describe in the reply a case where the distribution mapping fails in more detail, for e.g. by showing how the reference and forecast distribution look like and how the mapping fails to come up with a correct monthly mean value.

Page 14, lines 19 and 21: Usage of the term "time structure" seems to be misleading. I understand this term in a way that it refers to the temporal sequence of values, i.e. that the day n in the reference corresponds to day n in the forecasts. However, distribution mapping does not have this requirement. It is rather the rank structure as I would call it: Rank n in the reference has to correspond to the rank n in the forecasts. Please correct the terminology or explain in more detail what "time structure" means.

AR: We believe that compensation effects (linked to data aggregation) may occur when evaluating monthly values with bias corrected daily values. Since daily corrections are more numerous than monthly corrections, this can result in more flexibility and daily correction performing better than monthly correction when evaluated at the monthly scale. This is more or less similar to monthly correction performing well when evaluated at the yearly scale. However, as mentioned by the reviewer,

this may also be linked to a "higher sensitivity of monthly corrections to overfitting". Further studies would be necessary to conclude more firmly on this issue.

Concerning "time structures", we agree that the terminology may not be very clear. We meant that, for the DM methods to be efficient, the uncorrected and corrected values with the same rank (in their respective cumulative distributions) should also occur at the same time (e.g. as observed in the hyetograph). Therefore, the "time evolution" of values should be consistent and DM methods will be more efficient if they are applied on forecast hyetograph that are not too discrepant. We believe that this can go unnoticed in performance evaluation if daily values are corrected and aggregated at the monthly scale for evaluation. However, it will be harder to cover up if we apply and evaluate the corrections at the same time scale.

In the process of reducing the length of the paper, this paragraph was removed in the revised version.
* * *
RC: Section 5.2: In my opinion, this section does not give new information which is not already present in figure 6 (time varying bias-correction factors can be inferred from the panel "Before bias correction") and I suggest removing it for the sake of shortening the result section. The only new aspect is that the correction factors for EDMD vary more than for LS, but this comparison is not valid in my point of view as one should not compare a mean correction factor with a correction factor for a quantile level. I'm pretty sure that if you would calculate the correction factor for the mean in the case of EDMD, it would be very similar to the LS factor.

AR: The idea behind this plot was not to compare LS and EDMD average correction factors, but to give an additional element to understand the different features behind the LS and the EDMD methods. The main conclusions from this figure are that (1) EDMD can correct the frequency of null precipitations, whereas LS cannot, and (2) correction coefficients do not vary much from one application year to the other (especially with LS) and, therefore, in operational contexts, one can choose a more parsimonious calibration of the bias correction method applied.

In the revised version, we followed the reviewer's suggestion and removed this section.
* * *
RC: Section 5.3 and figures 8 and 9: I very much like this analysis. I'm not sure though if I really understand the analysis completely. MAE is partly related to the bias analysis in figure 6, i.e. if biases in figure 6 are substantial, then MAE should be even larger since MAE does not allow for a compensation of errors. EDM-y and GDM-y have large biases throughout the year in figure 6, and in some cases and particularly in summer, the bias is even larger than in the uncorrected data. However, in figure 8 the two methods stick out for MAE and IQR in summer lead to skill improvements in all catchments up to a lead time >60 days. To me, this seems to be contradicting. Could you please explain this particularity?

AR: Thank you. Fig. 8 reflects, somehow, the ability to bring skill to the (corrected) forecasts in terms of lead time and expressed as a percentage of catchments where improvements (comparatively to the raw forecasts used as reference) were seen. Even if EDM-y and GDM-y methods result in forecasts that still present some strong biases (as seen in, and commented from, Fig. 6), these may result in MAE values smaller than MAE values computed from the raw forecast. This is enough to characterize a relative gain in skill and, if this is observed over all lead times, the UFL will be >60 days and count in the percentage represented in Fig. 8. It seems contradictory at first sight, as well observed by the reviewer, but can, computationally, happen (e.g. due to the different aggregations: MAE is computed with daily values over a season, the bias is computed with monthly values over a month, or when biases change from under to over prediction or vice versa after correction). Overall, it is interesting to note that forecast skill is definitely hard to evaluate as there are many facets that one can look at. We tried to explore this in this paper and shed light on the different aspects that can better inform forecast users.
* * *
RC: Page 19, line 1: If I read the figure 10 correctly, there are negative skill score values and therefore, the statement that the skill scores are always larger than zero does not hold.

AR: You are right. We replaced the sentence with: "Nevertheless, bias corrected forecasts remain sharper than the reference (i.e., skill scores are mostly greater than zero)."
* * *
**RC: Page 20, line 3:** If I read the figure 12 correctly, there are negative skill score values and therefore, the forecast performs sometimes worse than ESP, which is the opposite of what is stated on this line.

AR: We changed the sentence to: "Overall, after bias correction, streamflow forecasts are sharper than ESP in most catchments and for most lead times".
* * *
**RC: Page 20, lines 12-13:** It is not clear to me why this is expected. I would expect that comparison to streamflow climatology is a harder check and therefore the skillfull lead time should be smaller than in the comparison to the baseline reference run since also the hydrological model bias deteriorates the skill. I surely misunderstand something but I think it would be good to add a bit more explanation in the manuscript.

AR: It is usually expected that ensembles based on streamflow climatology have less skill than ensembles based on hydrological modelling, at least in the first lead times, because ensembles based on hydrological modelling benefit from knowledge of initial hydrologic conditions. For instance, here, the states of the GR6J model are first initialized by running the model with observed inputs for a year prior to the forecast date. Therefore, ensembles based on streamflow climatology are supposed to be less skilful for forecast lead times that are impacted by initial hydrologic conditions. In the revised version, in order to clarify this issue, the sentence was changed to: "Streamflow forecasts generated from…up to twelve weeks in some catchments. This was expected because ensembles based on hydrological modelling benefit from knowledge of initial hydrologic conditions."
* * *
**RC: Page 22, line 4:** As for the uncorrected forecast discussion, I do not understand why it is the hydrological model that causes the problems with low-flow overestimation. The reference data is also output of the same hydrological model driven by the reference precipitation data. I would therefore rather think that it is some characteristics in the input data which the bias-correction cannot correct for that causes the problem (for e.g. dry-spell lengths). If the authors still think their statement holds, I would like to have a bit more explanations why this can be the case.

AR: The main discussion here is about reliability, which can clearly still be improved for streamflows. Comments on the model performance are linked to the analysis of the simulated and observed hydrographs, which complement the PIT analysis. The lack of reliability in streamflow forecasts may come from the input data, but not solely (as shown in Fig. 3, which analyses the reliability of the precipitation forcing). A lack of spread in hydrologic initial conditions may also play a role in the reliability of streamflow forecasts. That is why we referred to the needs of accounting for other sources of uncertainty, with, for instance, additional post-processing.
* * *
**RC: Page 27, lines 13-15:** References needed

AR: We added the following references:

*Hamlet, A. F. and Lettenmaier, D. P.: Columbia River Streamflow Forecasting Based on ENSO and PDO Climate Signals, J. Water Resour. Plan. Manag., 125(6), 333–341, doi:10.1061/(ASCE)0733-9496(1999)125:6(333), 1999.*

*van Dijk, A. I. J. M., Peña-Arancibia, J. L., Wood, E. F., Sheffield, J. and Beck, H. E.: Global analysis of seasonal streamflow predictability using an ensemble prediction system and observations from 6192 small catchments worldwide, Water Resour. Res., 49(5), 2729–2746, doi:10.1002/wrcr.20251, 2013.*

*Werner, K., Brandon, D., Clark, M. and Gangopadhyay, S.: Climate index weighting schemes for NWS ESP-based seasonal volume forecasts., J. Hydrometeorol., 5(6), 1076–1090, 2004.*
* * *
**RC: Section 6.4:** Although I like the illustrative character of this section, it stands a bit loose within the rest of the manuscript. I suggest to either motivate the section better or, for the sake of brevity, to remove it. In my opinion, the main statements of this sections have already been made, i.e. increased sharpeness after bias-correction compared to ESP.

AR: Thank you. Reviewer 1 also appreciated this figure and suggested some improvements, which we implemented in the revised version. We therefore added a quantification of what is shown in this figure. Notably, we show that the coverage probability of the streamflow forecasts is improved after bias

correction compared to ESP (see our answers to Reviewer 1 for details). Additionally, studies have shown the need to combine statistical evaluations with visual evaluations. Even though this is hard to achieve in probabilistic forecasting, we wanted to propose a visual appreciation of the ensembles to have a better overview of how bias corrections affect streamflow forecasts.
* * *
RC: **Figure 2:** "… and all seasons." The figure only shows two seasons, please correct the caption.

AR: Thank you for pointing this out. The caption was corrected in all occurrences of this problem.
* * *
RC: **Figures 3, 5, 11, 14:** The dashed lines should be explained in the figure as well, and not just in the text describing figure 3.

AR: The explanation for the dotted lines was added in the captions of the four figures.
* * *
RC: **Figure, 6:** Although certainly correct, I do not see a reason why to transform the simple relative bias into 1-bias. I understand that this transformation turns the bias into a skill score. However, in my opinion, the interpretation is not following the one for skill scores anyway. The perfect bias-correction would not yield 1 but 0. I suggest plotting the relative bias without transformation. The scale would be much easier interpretable as it directly refers to a percentage over- or underestimation.

AR: We used this transformation so that "no bias" corresponded to the null value, over-prediction corresponded to positive values and under-prediction corresponded to negative values. This representation of the scale seemed more intuitive, but the reviewer is right that the interpretation in terms of percentage is easier without this transformation. Figure 6 was modified in the revised version to take into account this reviewer's comment.
* * *
RC: **Figures 8 and 9:** Why are there different color scales for the different seasons?

AR: The four colours are supposed to help the reader identify the four seasons throughout the article. These colours include the blue and red colours used throughout the paper: blue for winter, lighter blue for autumn, red for summer and lighter red for spring. In these figures, the four colour scales in the legend are needed to clarify the colour shades related to the percentage of catchments in each category (e.g. to avoid light blue (autumn) being mistaken for a shade of bright blue (winter)).
* * *
RC: **Figure 15:** What are the colours standing for? There is probably also an error in the caption where it reads "shown for all seasons".

AR: The colours represent the four seasons as mentioned in the reply above. We added a legend for the colours in the figure.
* * *
RC: **Page 19, line 10:** precipitation instead of precipitations

AR: This was corrected in the revised version.

**Reviewer 3**
* * *
**Reviewer's comment (RC):**
Main points:
1) My most important point is that the paper is too long. I suggest to set a hard (!) reduction requirement of at least 25% (number of words). It is up to authors to decide which parts they remove or shorten. Just a few suggestions from my side: discuss fewer bias correction methods, remove almost completely page 13 line 3 – page 14 line 7, remove third and fourth sentence of section 3.2.1.

Authors' reply (AR): We have cut several sentences along the text and removed an entire analyses concerning the corrective factors (Section Comparison of bias correction factors for LS and EDMD methods), which reduced the length of the paper.
* * *
**RC:** 2) In the paper sharpness is discussed with the assumption that quality increases with sharpness. Mason and Stephenson (2008) write that "in the extreme case of no predictability, the forecast probability should always be equal to the climatological probability". So, forecasts can be too sharp, which should be a conclusion from e.g. Figure 2, where sharpness for longer lead times is larger than that of the reference. So, the sharpness results and conclusions should be reconsidered.

AR: We agree with the reviewer that sharpness in itself, as any other forecast quality attribute, is not necessarily an indicator of a perfect forecast. In our study we adopted the paradigm of Gneiting et al. (2007): « maximizing the sharpness of the predictive distributions subject to calibration ». This means that for two systems with equal levels of reliability, the best one is the sharper one (i.e., lower IQR score in our study). The evaluation of sharpness is thus complementary to the evaluation of reliability. That is the reason why we adopted the scores based on the PIT diagram and the IQR. In order to clarify this issue, we added the following sentence to Section 3.3.1: "In this study, we considered that for two reliable systems, the sharpest one is the best (Gneiting et al., 2007)."
* * *
**RC:** 3) A better (and longer) introduction to PIT diagrams is needed. Since these diagrams are not well explained in the paper, I was not able to understand the PIT results. I suggest at least to write much more clearly how these diagrams are constructed, to show a figure like Figure 2 from Laio and Tamea, to clarify what PIT values (vertical axis of figures in paper) are and to add a text to the horizontal axis of the PIT diagrams displayed in the paper. How does the area in the diagrams measure reliability? Is the area also sensitive to bias? Is that acceptable? In Section 3.3.1. the text mentions "concentration of points" but only lines are shown in the diagrams. So, what do you mean by "concentration of points"?

AR: The probability integral transform (PIT) histogram is used in forecast verification to evaluate if the empirical time series of PIT values (the PIT value is the value that the predictive cumulative distribution function associates with the observation at a given time step) has a uniform distribution (see also, Gneiting et al., 2005 [1], where it is also explained that "uniformity is usually evaluated in an exploratory sense, and one way of doing this is by plotting the empirical cumulative distribution function of the PIT values"). This is what we have done in our paper. In order to compare systems, we also evaluated the score defined as the "PIT area", as proposed in the reference cited in the paper (Renard et al., 2010). The further the PIT curve is from the 1:1 diagonal, the less reliable the ensemble is. Therefore, the smaller the area between the curve and the 1:1 diagonal, the more reliable the ensemble is. The rank histogram or Talagrand diagram, proposed independently in the literature, is a similar measure. Gneiting et al. (2005) indicate that "If we identify the predictive distribution with the empirical cumulative distribution function of the ensemble values, this technique is seen to be equivalent to plotting a PIT histogram". The visual inspection of the PIT diagram can be a useful assessment (on systematic biases or spread deficiencies), but forecast deficiencies may still be hidden behind the assessment (deficiencies in sharpness, for instance). That's why we use (and recommend) the joint evaluation of other scores. We hope this clarifies our approach. We would like to avoid adding a figure that is already presented in another easy-to-access paper that we are referencing (Laio and Tamea, 2007), especially since we are asked to shorten the paper. However, to make the PIT interpretation clearer, we modified some sentences in the description of this score in Section 3.3.1, and we added a more explicit title to the x axes of the PIT diagrams in our figures. The terms describing and explaining the shapes of the PIT diagram do not refer to "points" anymore (we have linked our points with lines for a better visualization of the results of the 16 catchments in a unique PIT diagram, and we think that deleting references to "points" will make the PIT diagram easier to understand).

[1] Probabilistic Forecasts, Calibration and Sharpness, Tilmann Gneiting Fadoua Balabdaoui and Adrian E. Raftery. Available here: https://www.stat.washington.edu/research/reports/2005/tr483.pdf
* * *
**RC:** 4) PIT area, MAE and CRPS are all sensitive to bias, as far as I can see. This should be mentioned in Sections 3 and 7 and discussed in Section 7.

AR: The scores are described in Section 3.3.1, and details on their characteristics can be found in the references provided. The way they are impacted by bias correction is illustrated throughout the results and discussed in Section 7.
* * *
RC: 5) Section 2.2 mentions that observations are used to initialize streamflow. What about the initialization of snow and soil moisture? These form important contributions to predictability.

AR: The GR6J model is a conceptual, reservoir-based hydrological model. Its inputs are daily precipitation and potential evapotranspiration. These data are used to run the model and initialize its states, including the state of its reservoirs, prior to the forecast date. The upper reservoir of the model can be assimilated (although it is not equivalent to, as it is not a physically-based model) to a "soil moisture accounting" reservoir. Therefore, in a sense, this is also initialized. As for snow modelling, it is not represented in the version of the model used in our study. The catchments studied have a dominant pluvial regime and are not strongly influenced by snow. In Section 2.2, we mention the forecast updating of the model, which is a different procedure from the initialization. After initialization, the model goes through an "updating procedure", common in hydrological forecasting, which, in our case, is based on the last observed discharge. We slightly changed the description of the model in the revised version (Section 2.2), which we hope will make this part of the paper clearer.
* * *
RC: 6) Sections 3.2.1. and 3.2.2. about the bias correction methods need references. EDM and GDM seem to be have strange effects: a specific amount of daily precipitation is corrected differently for different years, depending on the monthly amount of precipitation. What is the motivation to possibly employ these two methods? Perhaps some of the investigated methods should not be considered at all, see point 1 about shortening the paper. I found LS-m and EDMD-m the most interesting methods.

AR: Our motivation is to evaluate if EDM brings additional value regarding LS, notably in correcting bias for extreme precipitation, and whether the use of a fitted distribution (here, GDM) enhances performance or not. We also found LS-m and EDMD-m more interesting, but this comes from the progressive analysis of all the other methods too. We think it is important to show all the methods as they have different levels of complexity. References on the bias correction methods are already provided in the Introduction. We also added a reference to Yuan et al. (2015), which gives an extensive review of methods and a list of additional references.
* * *
RC: Minor points:
page 1, line 16: "contributes" instead of "contribute".
page 2, line 7: "widespread use of" instead of "the widespread of"
page 2, line 21: remove "rather than by initial conditions"
page 3, line 13: "varied between" instead of "derived from"

AR: These points were corrected in the revised version.

RC: The hydrological model also needs temperature as input to compute potential evapotranspiration. Write clearly how this input is constructed.

AR: The calculation of the evapotranspiration was done following the Oudin formulation. This formulation can be found in Equation (3) of Oudin et al. (2005). It was computed based on the daily temperature from the SAFRAN reanalysis. We rephrased it in the revised version to make it clearer.
* * *
RC: page 3, line 18: add "heavily" before "influenced"

AR: This was corrected in the revised version.
* * *
RC: page 3, line 23: replace "interannual" by "long-term mean". Over which years? On a monthly basis? Also for hindcasts?

AR: For a given day of the year, the estimated PET on this day is the mean of all PET computed for this day of the year, over all available years (with exception for the targeted year). Reviewer 1 and 2 also pointed out that the PET used in the article should be better explained (please, refer to the answers to their reviews). The following text was added in Section 2.1: "The interannual potential

evapotranspiration was then computed in each catchment, i.e. for a given day of the year, we computed the average potential evapotranspiration for this day over all available years (1958 to 2010)."
* * *
RC: page 3, section 2.2: motivate why the focus is solely on the influence of precipitation input.

AR: This is a choice we made as we were focusing on catchments with a pluvial-dominated hydrological regime. We added the following sentence in Section 2.2: "This setup is also consistent with the fact that our catchment set is dominated by a pluvial regime".
* * *
RC: page 6, section 3.3: So, do the evaluations for lead week 1 for the winter include all the hindcasts made on December 1, January 1 and February 1? These are then 15 members issued in December and January and 52 members issued in February. How do you deal with this inequality? And do the evaluations for lead week 6 for the winter include all the hindcasts made on November 1, December 1 and January 1? Explain this clearly.

AR: The reviewer's understanding is correct. We can thus have seasonal-based scores that involve forecasts with 15 or 51 members. This comes from the data setup of ECMWF. We only handled inequality when comparing ensemble of different sizes with the CRPS (as explained in the paper). Despite the inequality in the seasonal aggregation of scores, we note that this should not impact comparisons between seasons (since all seasons have a month with 51 members), and comparisons between raw and bias corrected forecasts (since aggregation is considered equally in both systems).
* * *
RC: page 7, line 8: "coinciding with" instead of "superposed with"
page 7, line 24: "Ranked" instead of "Rank"

AR: These were corrected in the revised version.
* * *
RC: page 8, line 6: What is the observation period?
RC: page 8, line 14: From which period are the observations?

AR: Observed precipitation data were available for the period running from 1958 to 2010. Observed streamflow data were available for different time periods, ranging from 36 years to 52 years depending on the catchment, and up to 2010. This was specified in Section 3.3.2 of the revised version.
* * *
RC: page 8, line 23: "caused" instead of "brought"

AR: This was corrected in the revised version.
* * *
RC: page 8, line 28: "becomes negative". What is done if there is more than one transition from a positive to a negative score?

AR: If there are several transitions, the lead time of the first transition is considered. In the revised version, we added "first" before "lead time beyond which" to make this clearer.
* * *
RC: page 9, line 28: "this is observed in the majority of catchments". This does not seem to be the case. There is roughly an equal number of curves below and above zero.

AR: The reviewer is right. In the process of shortening the paper, this sentence was removed in the revised version.
* * *
RC: page 13, figure 6: I would expect no bias at all in the lower right and left panel. What is the cause of these biases? Are the remaining biases caused by the one-year-leave-out method? If so, I would expect them to vary randomly around zero.

AR: We also believe that they may be mainly due to the one-year-leave-out approach, especially when differences among the validation (target) year and the calibration period exist (e.g. for the wettest or driest years of the data period, which may not be of equal intensity). Depending on the "distance" between the target year and the calibration period this may cause a divergence from zero.

**RC:** page 13, line 13: "in the easternmost part" instead of "at the most eastern part"

AR: This was corrected in the new version.

**RC:** page 14, line 30: add "cumulative" before "probability"

AR: This section was deleted in the revised version in order to shorten the length of the paper.
* * *
**RC:** page 17, figure 8: "Fraction of catchments" instead of "Number of catchments"

AR: This was changed in Figure 8 and in Figure 9.
* * *
**RC:** page 18, last line: As far as I can see the CRPS in not lower after bias correction.

AR: The reviewer is right. We changed the sentence to "In some catchments, the values of IQR are lower, but bias corrected forecasts remain sharper than the reference (i.e., skill scores are mostly greater than zero)" to clarify this point.
* * *
**RC:** page 19, line 3: replace "in regards to" by "with respect to"

AR: This was corrected in the revised version.
* * *
**RC:** I recommend to combine figure 2 with figure 10 into one figure, and figures 3 with figure 11 into one figure, etc. The reader now has to turn over pages to compare the figures.

AR: Combining figures with and without bias correction from EDMD-m, would in fact disturb the logic of the paper since, in between these figures, we propose an in-depth analysis of the impact of the bias corrections on forecast quality for each month and each tested method. We thus preferred to keep the organization as it was originally proposed.
* * *
**RC:** Figure 15: how are seasons represented?

AR: Strong blue is used for winter, lighter blue for autumn, red for summer and lighter red for spring. We added a legend for the four seasons represented in the figure.

[revised manuscript text omitted]

---

## Referee Report (RR1)

**Bias correcting precipitation forecasts to improve the skill of seasonal streamflow forecasts**

by Louise Crochemore, Maria-Helena Ramos, Florian Pappenberger

**General comments**

This is a revised version of the manuscript that I already reviewed in the first round (I therefor refrain from writing a summary). The authors have carefully revised most reviewer comments and did so in a very transparent way which is easy to follow for a reviewer. Thanks for that. I went through the revised manuscript and mainly checked if the various reviewer comments were addressed or not. Only in a very few minor issues, I made new comments.

Please find my comments below. I used the same comment labels as they were written in the reply. If anything is not listed below, this means that I accepted the authors reply. From my viewpoint, the main issue that needs a bit further discussion is the MAE results of GDM-y and EDM-y for JJA and its discrepancy with respect to the bias-results in Fig. 6. This is mainly due to my first comment not being concise enough. My apologies for that. I'm happy to discuss this in the open discussion and could reply to a posted comment if the authors posted a reply to that specific comment before July 15, as I'm out-of-office after that. Otherwise, I leave it to the editor to decide whether or not the reply is satisfactory.

Overall, I think the manuscript has much improved and is much better readable than in the first version. I suggest that the article is ready for publication after a few minor comments have been taken into account.

**Detailed comments**

**New comments to the revised version:**
**Page 6, line 8-10:** The sentence "A scaling factor higher (lower) than 1 indicates that the mean ensemble forecast underpredicts (overpredicts) the mean observed value. A value of 1 indicates no bias in the forecasts." could be left out since the analysis of the scaling factors has been removed from the manuscript. Instead, a reference for the method could be given (for e.g. Lenderink et al. (2007), or any other paper that gives a more detailed explanation of the approach).

**Page 7, line 19:** "equally reliable" instead of "reliable"

**Page 24, lines 6-7:** This sentence about the challenges for the bias-correction methods would seem to fit better into a later paragraph, since the current paragraph is describing the raw forecasts. Consider moving this sentence to a different paragraph.

**Reviewer comments that were not addressed entirely satisfactorily:**
**Reviewer 1, comment 1:** I think the rephrased introduction section about novelty and the main goals of the study is fine. However, the statement about "… no previous study has compared bias correction methods and their impact on streamflow forecasting in a systematic way, with a focus on understanding how the main attributes of forecast performance are impacted by bias correction." does not hold. First, it is contradictory to the statement on line page 3, line 7 where you say that such studies are rare – which implies that there are at least some studies -, and second, after a short search on the internet, I found the paper by Hashino et al. (2007). It seems that seasonal forecasts in combination with bias-correction methods has been studied before. This said, I do not think this impedes on the quality of the paper. It is just the statement above that needs to be removed or adjusted.

**Reviewer 2, Page 4, line 23:** I understand now what the authors mean by interannual potential evapotranspiration, but I still think that the term "interannual" is misleading. It is not the potential evapotranspiration between the years, but a multi-year climatological mean year. I suggest rephrasing the term, for e.g. to something alike "climatological potential evapotranspiration".

**Reviewer 2, Page 9, line 12:** In the revised version, you refer to the lines corresponding to the 5% significance test. In your reply, you mention though that the value of 0.1 for the bands is not exactly correct for the 5% significance level. I'm a bit confused by the different information given. I do not doubt that the value of 0.1 is a good choice, but if you show the lines at 0.1, you should not write that they represent the 5% significance level. Please correct the manuscript accordingly.

**Reviewer 2, Page 11, line 8-9:** Thanks a lot for the explanations. I indeed misinterpreted the figures in the first place.

**Reviewer 2, Section 5.3 and figures 8 and 9:** I agree with the authors that even in case of strong biases in GDM-y and EDM-y, the MAE might still, in principle, show some skill improvement with respect to uncorrected forecasts. For me, this would be logical in case the strong biases after bias-correction are still smaller than before bias-correction.

Related to the last condition stated above, it appears now to me that I have formulated my comment unclearly. I should have written that I'm particularly surprised to see such a clear gain in MAE skill for summer since the bias after bias-correction is larger than before bias-correction. In Fig. 6 and for the months JJA, the bias skill after bias-correction shows a clear underprediction whereas for the data before bias-correction, the bias-skill is very close to 1 or slightly on the overprediction side for the catchment 4. It is this before bias-correction vs after bias-correction discrepancy as well as the very clear skill gain in figure 7 for JJA that make me doubting the results for GDM-y and EDM-y in figure 7. Or to look at the same thing from a different perspective, I would expect the methods with low bias in Fig. 6 to also show better skill in MAE than methods with high bias in Fig. 6. (as long as no strong non-linearities are to be expected, which I doubt here). How could one for example explain that EDMD-m that has a bias of very close to 1 in Fig. 6 for all the months in JJA only leads to skill gain in MAE in about 60% of the catchments, but EDM-y which has a clear underprediction in Fig. 6 and is clearly worse than EDMD-m still leads to an extremely high gain increase?

Could the authors please comment a bit more on this issue? I'm happy to follow this on the interactive discussion. However, I'm out-of-office from July 15[th] until August 15[th].

**Reviewer 2, figure 6:** I appreciate that the authors took into account my comment. It seems to me that in the revised manuscript on page 11, line 13 and following, the text explaining the bias values shown in figure 6 has not been changed accordingly. Also, if you are plotting the relative bias in figure 6 now, why is the interval from 1 to 4 so much larger than the interval from 0.25 to 1? Is there still some transformation involved? If there is no reason for the unequal interval size, I suggest to use a linear colorscale for easier interpretation (i.e. equal distance on the colorscale means equal factor difference to the no-bias point).

**Reviewer 3, RC: page 3, line 23:** I agree with the reviewer that the term interannual PET is confusing, and the suggestion "long-term mean" sounds fine to me. The authors should change the wording since two of three reviewers observed the same issue.

**RC: 3)** I noticed that in the result section, the authors still refer to points when discussing the PIT diagram (for e.g. on page 9, line 8). For consistency reason, this should be changed according to the changes in section 3.

**References**

Hashino, T., Bradley, a. a., & Schwartz, S. S. (2007). Evaluation of bias-correction methods for ensemble streamflow volume forecasts. *Hydrology and Earth System Sciences*, *11*, 939–950. doi:10.5194/hess-11-939-2007

Lenderink, G., Buishand, a., & van Deursen, W. (2007). Estimates of future discharges of the river Rhine using two scenario methodologies: direct versus delta approach. *Hydrology and Earth System Sciences*, *11*(3), 1145–1159. doi:10.5194/hess-11-1145-2007

---

## Author Response (AR2)

**Changes in the revised version and answers to Reviewer 2**

We thank the reviewer for his/her very relevant additional comments and suggestions. All changes and answers to the reviewer are detailed below.

**Reviewer 2**

**General comments**

This is a revised version of the manuscript that I already reviewed in the first round (I therefor refrain from writing a summary). The authors have carefully revised most reviewer comments and did so in a very transparent way which is easy to follow for a reviewer. Thanks for that. I went through the revised manuscript and mainly checked if the various reviewer comments were addressed or not. Only in a very few minor issues, I made new comments.

Please find my comments below. I used the same comment labels as they were written in the reply. If anything is not listed below, this means that I accepted the authors reply. From my viewpoint, the main issue that needs a bit further discussion is the MAE results of GDM-y and EDM-y for JJA and its discrepancy with respect to the bias results in Fig. 6. This is mainly due to my first comment not being concise enough. My apologies for that. I'm happy to discuss this in the open discussion and could reply to a posted comment if the authors posted a reply to that specific comment before July 15, as I'm out-of-office after that. Otherwise, I leave it to the editor to decide whether or not the reply is satisfactory.

Overall, I think the manuscript has much improved and is much better readable than in the first version. I suggest that the article is ready for publication after a few minor comments have been taken into account.

**Detailed comments**

**New comments to the revised version:**

**Reviewer's comment (RC):** Page 6, line 8-10: The sentence "A scaling factor higher (lower) than 1 indicates that the mean ensemble forecast underpredicts (overpredicts) the mean observed value. A value of 1 indicates no bias in the forecasts." could be left out since the analysis of the scaling factors has been removed from the manuscript. Instead, a reference for the method could be given (for e.g. Lenderink et al. (2007), or any other paper that gives a more detailed explanation of the approach).

**Authors' reply (AR):** These sentences were removed. We also removed the sub-sections since they are not needed anymore. A reference to Teutschbein and Seibert (2013), were the method is explained with details, was added..

**RC:** Page 7, line 19: "equally reliable" instead of "reliable"

**AR:** We think it is tricky to use the word "equally" because here we refer to the feature of reliability qualitatively, as an attribute of the quality of a forecast. The word "equally" may imply that the two systems have identical values of a score measuring their reliability, which is not the meaning here. We prefer to keep the sentence unchanged in the revised version.

**RC:** Page 24, lines 6-7: This sentence about the challenges for the bias-correction methods would seem to fit better into a later paragraph, since the current paragraph is describing the raw forecasts. Consider moving this sentence to a different paragraph.

**AR:** We moved the sentence to the third paragraph of the conclusion.
* * *
**Reviewer comments that were not addressed entirely satisfactorily:**
* * *
**RC:** Reviewer 1, comment 1: I think the rephrased introduction section about novelty and the main goals of the study is fine. However, the statement about "… no previous study has compared bias correction methods and their impact on streamflow forecasting in a systematic way, with a focus on understanding how the main attributes of forecast performance are impacted by bias correction." does not hold. First, it is contradictory to the statement on line page 3, line 7 where you say that such studies are rare – which implies that there are at least some studies -, and second, after a short search on the internet, I found the paper by Hashino et al. (2007). It seems that seasonal forecasts in combination with bias-correction methods has been studied before. This said, I do not think this impedes on the quality of the paper. It is just the statement above that needs to be removed or adjusted.

**AR:** We agree with the reviewer that the paper mentioned (Hashino et al., 2007) is a reference to be included. In our original text, we were separating between "studies evaluating bias correction methods" and "studies evaluating bias correction methods in a systematic way" in order to go beyond comparing quality score values and to additionally provide clarifications on the impacts and trade-offs between the different attributes of forecast quality. We agree however that this difference is very subtle and we changed the statement on page 3 to "only few studies have compared" and added the proposed reference, as well as Wood and Schaake (2008).
* * *
**RC:** Reviewer 2, Page 4, line 23: I understand now what the authors mean by interannual potential evapotranspiration, but I still think that the term "interannual" is misleading. It is not the potential evapotranspiration between the years, but a multi-year climatological mean year. I suggest rephrasing the term, for e.g. to something alike "climatological potential evapotranspiration".

**AR:** The reviewer is right. We replaced "interannual" by "multi-annual mean", following the UNESCO glossary.
* * *
**RC:** Reviewer 2, Page 9, line 12: In the revised version, you refer to the lines corresponding to the 5% significance test. In your reply, you mention though that the value of 0.1 for the bands is not exactly correct for the 5% significance level. I'm a bit confused by the different information given. I do not doubt that the value of 0.1 is a good choice, but if you show the lines at 0.1, you should not write that they represent the 5% significance level. Please correct the manuscript accordingly.

**AR:** Thanks for pointing this out. We deleted all references to the 5% level of Kolmogorov significance test since this level would in fact represent +/- 0.15 bands and not +/- 0.1 bands.
* * *
**RC:** Reviewer 2, Page 11, line 8-9: Thanks a lot for the explanations. I indeed misinterpreted the figures in the first place.

**AR:** We are glad to learn that it is clearer now.
* * *
**RC:** Reviewer 2, Section 5.3 and figures 8 and 9: I agree with the authors that even in case of strong biases in GDM-y and EDM-y, the MAE might still, in principle, show some skill improvement with respect to uncorrected forecasts. For me, this would be logical in case the strong biases after bias-correction are still smaller than before bias-correction. Related to the last condition stated above, it appears now to me that I have formulated my comment unclearly. I should have written that I'm particularly surprised to see such a clear gain in MAE skill for summer since the bias after bias-correction is larger than before bias-correction. In Fig. 6 and for the months JJA, the bias skill after bias-correction shows a clear underprediction whereas for the data before bias-correction, the bias-skill is very close to 1 or slightly on the overprediction side for the catchment 4. It is this before bias-correction vs after bias-correction discrepancy as well as the very clear skill gain in figure 7 for JJA that make me doubting the results for GDM-y and EDM-y in figure 7. Or to look at the same thing from a different perspective, I would expect the methods with low bias in Fig. 6 to also show better skill in MAE than methods with high bias in Fig. 6. (as long as no strong non-linearities are to be expected, which I doubt here). How could one for example explain that EDMD-m that has a bias of very close to 1 in Fig. 6 for all the months in JJA only leads to skill gain in MAE in about 60% of the catchments, but EDM-y which has a clear underprediction in Fig. 6 and is clearly worse than EDMD-m still leads to an extremely high gain increase? Could the authors please comment a bit more on this issue? I'm happy to follow this on the interactive discussion. However, I'm out-of-office from July 15 th until August 15 th.

**AR:** If we understand correctly, the reviewer would expect that a stronger bias after bias correction should be associated with worse MAE values after bias correction. The results in Figure 6 and Figure 7 would thus be contradicting. In fact, it is possible to improve one while deteriorating the other and we try to explain why hereafter.

The following two figures show the skill scores vs. lead time that led to the results in Figure 7. The raw forecasts are used as reference. Results are shown for JJA and for EDM-y and EDMD-m. With EDM-y, all curves are above the zero line. In other words, all catchments have a smaller (i.e. better) MAE after bias correction, which is not the case with EDMD-m. This is consistent with Figure 7.

[Figure]

A first obvious difference between the bias and the MAE is that the first one focuses on monthly-averaged values, while the second one focuses on daily errors. The improvement (or deterioration) of skill indicated by these scores is thus hard to compare because they act over different variables.

The following figure illustrates what happens during the summer season. It represents the observed daily precipitation values (in black), and the corresponding mean forecast obtained with raw System 4

precipitations (in red) and with System 4 precipitations bias corrected with EDM-y (in blue). The forecast is issued for Catchment 2 on June 1st 1989. Only the lead times with a month-2 lead are represented (for consistency with Figure 6). We computed summed precipitations (Sum) and the MAE for each curve.

[Figure]

In accordance with Figure 6, the raw forecast (29 mm) overestimates the observed precipitation (22 mm) for the period. Because EDM-y is calibrated over the year, the scaling factor smaller than 1 drastically reduces the forecast sum precipitations (7 mm). This phenomenon generalized to most forecast dates in June, July and August, leads to strong biases with EDM-y, as observed in Figure 6. In the figure, we observe that bias corrected daily precipitations are thus strongly reduced, which, in fact, leads to better forecasts for null and low precipitations. Since null and low precipitations are predominant in JJA time series, and because the MAE gives an equal weight to all errors in the time series, improving null or low precipitations improves the MAE.

RC: Reviewer 2, figure 6: I appreciate that the authors took into account my comment. It seems to me that in the revised manuscript on page 11, line 13 and following, the text explaining the bias values shown in figure 6 has not been changed accordingly. Also, if you are plotting the relative bias in figure 6 now, why is the interval from 1 to 4 so much larger than the interval from 0.25 to 1? Is there still some transformation involved? If there is no reason for the unequal interval size, I suggest to use a linear colorscale for easier interpretation (i.e. equal distance on the colorscale means equal factor difference to the no-bias point).

AR: Thank you for pointing this out. The section 5.1 was adjusted accordingly and now matches the figure. Concerning the scales, the bias represented in Figure 6 is not a relative bias [i.e., (Obs – For)/Obs], but it is computed by taking the ratio between the observed and the forecast values (as stated in the Section "Bias correction methods", page 6, and reminded in Section 5.1, page 11). This is why the scale is non-symmetrical.

RC: Reviewer 3, page 3, line 23: I agree with the reviewer that the term interannual PET is confusing, and the suggestion "long-term mean" sounds fine to me. The authors should change the wording since two of three reviewers observed the same issue.

**AR:** We replaced "interannual" by "multi-annual mean", following the UNESCO glossary (see also answer above).

RC: 3) I noticed that in the result section, the authors still refer to points when discussing the PIT diagram (for e.g. on page 9, line 8). For consistency reason, this should be changed according to the changes in section 3.

**AR:** Thank you for pointing this out. This was modified:

[revised manuscript text omitted]